# DOUBLE/DEBIASED MACHINE LEARNING FOR TIME-TO-EVENT OUTCOMES UNDER POOR OVERLAP

[1]

## ABSTRACT

In empirical studies with time-to-event outcomes, investigators often leverage observational data to conduct causal inference on the effect of exposure when randomized controlled trial data is unavailable. Model misspecification and lack of overlap are common issues in observational studies, and they often lead to inconsistent and inefficient estimators of the average treatment effect. Estimators targeting overlap weighted effects have been proposed to address the challenge of poor overlap, and methods enabling flexible machine learning for nuisance models address model misspecification. However, the approaches that allow machine learning for nuisance models have not been extended to the setting of weighted average treatment effects for time-to-event outcomes. In this work, we propose a class of one-step cross-fitted double/debiased machine learning estimators for the weighted cumulative causal effect as a function of restriction time. We prove that the proposed estimators are consistent, asymptotically linear, and reach semiparametric efficiency bounds under regularity conditions. Our simulations show that the proposed estimators using nonparametric machine learning nuisance models perform as well as established methods given access to correctly-specified parametric nuisance models, illustrating that our estimators mitigate the need for oracle parametric nuisance models. We apply the proposed methods to real-world observational data from a UK primary care database to compare the effects of anti-diabetic drugs on cancer clinical outcomes.

## 1 INTRODUCTION

Two well documented challenges in estimating average treatment effects (ATEs) from observational data—even when the standard causal assumptions of consistency, positivity, and exchangeability hold—are poor overlap between treatment groups and so-called nuisance model misspecification. Poor overlap occurs when one (binary) treatment assignment is much likelier than the other at certain values of covariates that need to be adjusted for in the analysis. If a treatment value cannot occur at all at certain covariate values, then the positivity assumption is violated and the quantity of interest is not identified. But even if positivity is not violated and overlap is merely poor, ATE estimators may have undesirable statistical properties. Methods have been proposed to address the issue of poor overlap by "moving the goalposts" (Crump et al., 2006) towards the population with equipoise and targeting a weighted average treatment effect (WATE) in which subjects with propensity scores near zero or one are down-weighted. Some authors have even argued that these alternative estimands are more policy relevant (in addition to having more stable estimators) than the ATE. We will consider WATE estimators in this work.

For time to event outcomes with censoring, multiple measures have been defined to quantify the effect of exposure, such as cumulative hazards (Vansteelandt et al., 2022), number-needed-to-treat, survival functions, median survival times, and restricted mean survival time (RMST) (Gao et al., 2025). In this work, we focus on the RMST which is the expected event-free survival time through the follow-up time of interest $\tau$, the 'restriction time'. This quantity has the appealing properties that it both summarizes survival outcomes in a single number and retains a causally and clinically mean-

---

[1]Code: `https://github.com/xushenbo/dml`

ingful interpretation. Past work has proposed estimators for WATEs on survival functions (Cheng et al., 2022). However, these methods depend on correctly specified parametric nuisance models.

Approaches have been developed to solve the issue of potential misspecification of parametric nuisance regression models for the outcome, treatment, and censoring process required to compute causal effect estimators. Nonparametric machine learning models are inherently correctly specified. However, simply plugging machine learning estimated nuisance parameters into standard effect estimators can lead to plug-in bias and incorrect confidence intervals due to slow convergence rates (Ausset et al., 2022). Estimators based on influence functions (IFs), such as debiased machine learning (DML) (Chernozhukov et al., 2018; 2022; Mackey et al., 2018; Jung et al., 2021) and targeted minimum loss based estimation (TMLE) (Van Der Laan & Rubin, 2006), have been proposed to ensure $n^{1/2}$ consistency and asymptotic normality without imposing proportional hazards or other parametric restrictions on survival, treatment, or censoring distributions, if the outcome model and the treatment model are both consistent (e.g. if they are both estimated with nonparametric machine learning), and the *product* of their convergence rates is at least $n^{1/2}$. This property is known as rate double-robustness (Smucler et al., 2019). However, DML for WATEs in a survival setting has not yet been developed. That is a primary gap we fill here.

We also extend our estimators to the competing risks setting. When there are multiple causes of failure but only a subset is of interest, the other causes of failure are referred to as competing events because they can prevent the event of interest from occurring. In such scenarios, researchers often consider marginal, cause-specific, subdistribution, and composite cumulative incidence functions (CIFs) (Young et al., 2020). Here, we consider cause-specific restricted mean time lost (RMTLs) as a cumulative measure that summarizes CIFs which can roughly be interpreted as expected time lost from the cause or causes of interest until restriction time $\tau$ (Andersen, 2013). We are not aware of any estimator for the weighted cause-specific RMTL or the weighted cumulative treatment effect that is defined as the contrast between weighted treatment-specific RMTLs.

The organization of the paper is as follows. In Section 2, we introduce notation and formally define our weighted estimands of interest. In section 3, we derive a class of cross-fitted DML estimators for these estimands from their influence functions. In Section 4, we derive the asymptotic linearity of the proposed estimators and show the proposed estimators achieve semi-parametric efficiency bounds. We also provide methods for constructing pointwise confidence intervals and simultaneous confidence bands across a range of restriction times. In Section 5, we present a simulation study illustrating that our DML estimators with nuisance functions estimated via machine learning perform as well as estimators using correctly specified parametric nuisance models. In Section 6, we apply the proposed estimators to compare the effects of two first-line anti-hyperglycemia treatments on cancer clinical outcomes among patients with type II diabetes using real-world observational primary care data from the UK. In Section 7, we conclude.

## 2 DATA, ESTIMANDS, AND IDENTIFICATION

In this section, we introduce notation, define causal estimands, and state identification assumptions.

### 2.1 NOTATIONS

We observe pretreatment confounders $X \subseteq \mathcal{R}^d$, binary exposure $A \in \{0, 1\}$. Let $T_j^a > 0$ denote the potential failure time of cause $j$ had, possibly contrary to fact, treatment been set to $a$ and $T_j^a = \infty$ if failure from cause $j$ would not occur under treatment assignment $a$. Define $T^a = \min_j\{T_j^a\}$ as the uncensored potential survival time from 0 to the first event $j$ and $J^a = \operatorname{argmin}_j\{T_j^a\}$ as the corresponding potential cause of failure. We define $T = T^A$ and $J = J^A$ as the failure time and the cause of failure had $A = a$, respectively. We consider censoring as an external coarsening process at time $C$, the censored failure time $\widetilde{T} = \min\{T, C\}$, an event indicator as $\Delta = I(T \leq C)$ of whether failure is censored, and $\widetilde{J} = J\Delta$ as the observed cause of failure where $\widetilde{J} \in \{0, 1, \ldots, j^*\}$ and $j^*$ is the number of competing causes of failure. The structure of observed data unit is $O = (X, A, \widetilde{T}, \widetilde{J}) \sim P_0$, where $P_0$ is the unknown true data-generating distribution. When there is only one type of failure, i.e. $j^* = 1$, the observed data become $O = (X, A, \widetilde{T}, \Delta)$. We assume that we observe $n$ independent and identically distributed (i.i.d.) observations $O_1, \ldots, O_n$ sampled from

$P_0$ with probability $1/n$ and denote $P_n$ as the empirical probability distribution. We denote the observed cause-specific counting process as $N_j(t) = I(\widetilde{T} \leq t, \widetilde{J} = j)$, all-cause counting process as $N(t) = I(\widetilde{T} \leq t)$, censoring process as $N^C(t) = I(\widetilde{T} \leq t, \Delta = 0) = (1 - \Delta)N(t)$, and at-risk process as $R(t) = I(\widetilde{T} \geq t)$.

## 2.2 ESTIMANDS AND IDENTIFICATION

In the absence of competing risks, we take our causal estimand to be the weighted average difference in restricted mean survival time (RMST) under alternative treatments, i.e.

$$\psi_0^{\text{RMST},h}(\tau) = \psi_0^{\text{RMST},h}(\tau, a = 1) - \psi_0^{\text{RMST},h}(\tau, a = 0)$$
$$= \frac{E_0[h_0(X)\{\min(T^{a=1}, \tau) - \min(T^{a=0}, \tau)\}]}{E_0\{h_0(X)\}}$$

where $h_0(X)$ is a prespecified 'tilting function'. $h_0(X)$ is typically chosen to downweight subjects with propensity scores near 0 or 1 (Li et al., 2018). If there are multiple competing causes of failure ($j* > 1$), our estimand is the weighted average cause-specific restricted mean time lost (RMTL)

$$\psi_{j,0}^{\text{RMTL},h}(\tau) = \psi_{j,0}^{\text{RMTL},h}(\tau, a = 1) - \psi_{j,0}^{\text{RMTL},h}(\tau, a = 0)$$
$$= \frac{E_0[h_0(X)\{(\tau - \min(T^{a=1}, \tau))I(J^{a=1} = j) - (\tau - \min(T^{a=0}, \tau))I(J^{a=0} = j)\}]}{E_0\{h_0(X)\}}$$

The following assumptions are required to identify causal effects, for $a \in \{0, 1\}$:

**Assumption 1** *(Positivity)* $P_0(A = a \mid X) \geq 1/\epsilon$ *and* $P_0(C \geq t \mid A = a, X) \geq 1/\epsilon$ *almost surely for* $t \in [0, \tau]$ *and* $\epsilon \in (1, \infty)$;

**Assumption 2** *(SUTVA)* $T = T^A$, $T_i^{(A_1,...,A_n)} = T_i^{(A_1',...,A_n')}$, *if* $A_i = A_i'$, $\forall i$;

**Assumption 3** *(Conditional ignorability)* $T^a \perp\!\!\!\perp A \mid X$;

**Assumption 4** *(Conditionally independent censoring)* $T^a \perp\!\!\!\perp C \mid A, X$.

**Assumption 5** *(Competing risks SUTVA)* $(T^a, J^a) = A(T^{a=1}, J^{a=1}) + (1 - A)(T^{a=0}, J^{a=0})$, $(T^{(A_1,...,A_n)}, J^{(A_1,...,A_n)}) = (T^{(A_1',...,A_n')}, J^{(A_1',...,A_n')})$, *if* $A_i = A_i'$, $\forall i$;

**Assumption 6** *(Competing risks conditional ignorability)* $(T^a, J^a) \perp\!\!\!\perp A \mid X$;

**Assumption 7** *(Competing risks conditionally independent censoring)* $(T^a, J^a) \perp\!\!\!\perp C \mid A, X$.

Assumptions 1—4 are required for identification of (weighted) RMST estimands in the absence of competing risks, and assumptions 1 and 5—7 are required for identification of (weighted) RMTL estimands in the presence of competing risks. Assumption 1 requires that every unit has positive probability of receiving each treatment and remaining uncensored through follow up. Assumption 2 and 5 are called stable unit treatment value (SUTVA) assumptions. They state that observed outcomes are equal to the potential outcomes corresponding to observed treatments (consistency) and that one unit's treatment does not impact another's outcomes (no interference). The conditional ignorability and conditionally independent censoring Assumptions 3—4 and 6—7 state that observed baseline covariates are sufficient to adjust for confounding.

Under Assumptions 1-4, we have

$$\psi_0^{\text{RMST},h}(\tau, a) = \frac{E_0[h_0(X)\{\text{RMST}_0(\tau \mid a, X)\}]}{E_0\{h_0(X)\}}$$

and under Assumptions 1 and 5-7

$$\psi_{j,0}^{\text{RMTL},h}(\tau, a) = \frac{E_0[h_0(X)\{\text{RMTL}_{j,0}(\tau \mid a, X)\}]}{E_0\{h_0(X)\}}$$

where $\text{RMST}_0(\tau \mid a, X) = E_0\{\min(T, \tau) \mid A = a, X\}$ and $\text{RMTL}_{j,0}(\tau \mid a, X) = E_0[\{\tau - \min(T, \tau)\}I(J = j) \mid A = a, X]$.

## 3 PROPOSED METHODS

In this section, we consider influence function based estimation of the causal estimands defined in the previous section. We propose a double/debiased machine learning algorithm.

### 3.1 INFLUENCE FUNCTION

We first derive the IFs for our estimands Equation 1 and Equation 2. Then, we construct cross-fitted double debiased estimators based on these IFs, followed by a simple precedure to incorporate shape constraints. We provide sandwich estimators for variances and ensure that variance monotonically increases with respect to restriction time $\tau$.

We introduce several quantities before we proceed. Define the propensity score as $\pi(a \mid X) = P(A = a \mid X)$, the balancing weight as $w^h(a, X) = h(X)I(A = a)/\pi(a \mid X)$, the conditional survival function $S(t \mid a, X) = P(T > t \mid A = a, X)$, the conditional cumulative incidence function $F_j(t \mid a, X) = P(T \leq t, J = j \mid A = a, X)$, and the conditional censoring survival function $G(t \mid a, X) = P(C > t \mid A = a, X)$. Additional definitions for cumulative hazards $\Lambda$, martingales $M$, and nuisance parameter collections $\eta_0^\Lambda$, $\eta_0^{\Lambda,h}$, $\eta_{j,0}^\Lambda$, $\eta_{j,0}^{\Lambda,h}$ are provided in Appendix A.1.

In Theorem 1, we provide IFs for $\psi_0^{\mathrm{RMST},h}(\tau, a)$ and $\psi_{j,0}^{\mathrm{RMTL},h}(\tau, a)$.

**Theorem 1** *Assume $S_0(t \mid a, X) > 1/\epsilon$, $0 \leq h_{\min} \leq h_0(X) \leq h_{\max} \leq \infty$, where $\epsilon \in (1, \infty)$ and let $\Delta(\tau) = I(C > \tau \wedge T-)$, then $\psi_0^{\mathrm{RMST},h}(\tau, a)$ is a pathwise differentiable quantity with weighted uncentered IF as $\phi_0^{\mathrm{RMST},h}(\tau, a; \eta_0^{\Lambda,h}) = [E_0\{h_0(X)\}]^{-1}\{h_0(X)\phi_0^{\mathrm{RMST}}(\tau, a; \eta_0^\Lambda)\}$ and weighted centered IF*

$$\varphi_0^{\mathrm{RMST},h}(\tau, a; \eta_0^{\Lambda,h}) = [E_0\{h_0(X)\}]^{-1}[h_0(X)\{\phi_0^{\mathrm{RMST}}(\tau, a; \eta_0^\Lambda) - \psi_0^{\mathrm{RMST},h}(\tau, a)\}] \quad (1)$$

*where the unweighted nonparametric uncentered IF is given in Eq. equation 5 in Appendix D.*

$\psi_{j,0}^{\mathrm{RMTL},h}(\tau, a)$ *is a pathwise differentiable quantity with weighted uncentered IF $\phi_{j,0}^{\mathrm{RMTL},h}(\tau, a; \eta_{j,0}^{\Lambda,h}) = [E_0\{h_0(X)\}]^{-1}[h_0(X)\phi_{j,0}^{\mathrm{RMTL}}(\tau, a; \eta_{j,0}^\Lambda)]$ and weighted centered IF*

$$\varphi_{j,0}^{\mathrm{RMTL},h}(\tau, a; \eta_{j,0}^{\Lambda,h}) = [E_0\{h_0(X)\}]^{-1}[h_0(X)\{\phi_{j,0}^{\mathrm{RMTL}}(\tau, a; \eta_{j,0}^\Lambda) - \psi_{j,0}^{\mathrm{RMTL},h}(\tau, a)\}] \quad (2)$$

*where the unweighted nonparametric uncentered IF is given in Eq. equation 6 in Appendix D.*

The proof of Theorem 1 can be found in Appendix D.2. The first terms in Equation 5 and Equation 6 are the inverse probability weighted complete case terms, representing individuals who take the treatment of interest $A = a$ with observed event of interest before restriction time, $\Delta(\tau) = 1$ or $\widetilde{J} = j$. The empirical mean of the first terms are often called IPCW estimators (Robins & Rotnitzky, 1992). As the IPCW estimators only consider complete cases, there is a loss in efficiency of these estimators, which becomes severe when censoring is heavy and/or treatment assignment is highly unbalanced. To improve efficiency, the second terms in Equation 5 and Equation 6, similar to the augmentation term in doubly-robust estimators for continuous outcomes (Robins et al., 1994), capture information for all participants through conditional expectations of outcomes, regardless of exposure group or censoring status. The remaining Lebesgue-Stieltjes censoring martingale integral terms further restore information from censored observations $\Delta = 0$ in the exposure group of interest $A = a$.

### 3.2 DEBIASED MACHINE LEARNING ESTIMATORS

We describe a short version of the estimation procedure with four steps:

(i) partition the data with size $n$ into $K = 2$ disjoint sets $\mathcal{V}_1$ and $\mathcal{V}_2$ with sizes $n_1$ and $n_2$;

(ii) construct $\widehat{\eta}_1^{\Lambda,h}$ or $\widehat{\eta}_{j,1}^{\Lambda,h}$ trained on $\mathcal{V}_2$ and evaluated on $\mathcal{V}_1$. Reverse the training and estimation set to obtain $\widehat{\eta}_2^{\Lambda,h}$ or $\widehat{\eta}_{j,2}^h$;

(iii) For $k = 1, 2$, compute $\widehat{\psi}_k^{\mathrm{RMST},h}(\tau, a) = P_{n,k}\{\widehat{\phi}_k^{\mathrm{RMST},h}(\tau, a; \widehat{\eta}_k^{\Lambda,h})\}$ or $\widehat{\psi}_{j,k}^{\mathrm{RMTL},h}(\tau, a) = P_{n,k}\{\widehat{\phi}_{j,k}^{\mathrm{RMTL},h}(\tau, a; \widehat{\eta}_{j,k}^{\Lambda,h})\}$ by plugging in $\widehat{\eta}_k^{\Lambda,h}$ or $\widehat{\eta}_{j,k}^{\Lambda,h}$, where $P_{n,k} = 1/n_k \sum_{i \in \mathcal{V}_k}$ is the empirical probability distribution of subset $\mathcal{V}_k$ and the full expression for $\widehat{\phi}_k^{\mathrm{RMST},h}(\tau, a; \widehat{\eta}_k^{\Lambda,h})$ and $\widehat{\phi}_{j,k}^{\mathrm{RMTL},h}(\tau, a; \widehat{\eta}_{j,k}^{\Lambda,h})$ are given as Equation 3 and Equation 4 in Appendix A.2;

(iv) compute $\widehat{\psi}^{\mathrm{RMST},h}(\tau, a) = 1/2\{\widehat{\psi}_{k=1}^{\mathrm{RMST},h}(\tau, a) + \widehat{\psi}_{k=2}^{\mathrm{RMST},h}(\tau, a)\}$ or $\widehat{\psi}_j^{\mathrm{RMTL},h}(\tau, a) = 1/2\{\widehat{\psi}_{j,k=1}^{\mathrm{RMTL},h}(\tau, a) + \widehat{\psi}_{j,k=2}^{\mathrm{RMTL},h}(\tau, a)\}$ and correct for shape constraints thereafter following 4 in the Appendix.

The details of the implementation procedure of the proposed cross-fitted DML estimators are described in Appendix A.2. Note that estimators $\widehat{\psi}_k^{\mathrm{RMST},h}(\tau, a)$ and $\widehat{\psi}_{j,k}^{\mathrm{RMTL},h}(\tau, a)$ solve estimating equations corresponding to the influence functions for the target parameters.

The detailed implementation procedures, including handling of numerical stability, shape constraints, and computational considerations, are provided in Appendix A.2. Extensions to multiple treatments and mixed survival times are discussed in Appendix A.3.

# 4 ASYMPTOTIC PROPERTIES

In this section, we study the asymptotic behavior of the proposed estimators. We establish the asymptotic properties of our estimators and defer the detailed construction of confidence intervals and bands to Appendix D.1.

We consider large-sample properties of the proposed estimators here. We begin with conditions for pointwise and uniform consistency.

**Theorem 2** *(Consistency): Under Conditions 1-3,*

$$\widehat{\psi}^{\mathrm{RMST},h}(\tau, a) \xrightarrow{P} \psi_0^{\mathrm{RMST},h}(\tau, a), \quad \widehat{\psi}^{\mathrm{RMST},h}(\tau) \xrightarrow{P} \psi_0^{\mathrm{RMST},h}(\tau).$$

*If Condition 4 also holds, then convergence is uniform, i.e.*

$$\sup_{t \in [0,\tau]} \left| \widehat{\psi}^{\mathrm{RMST},h}(\tau, a) - \psi_0^{\mathrm{RMST},h}(\tau, a) \right| \xrightarrow{P} 0, \quad \sup_{t \in [0,\tau]} \left| \widehat{\psi}^{\mathrm{RMST},h}(\tau) - \psi_0^{\mathrm{RMST},h}(\tau) \right| \xrightarrow{P} 0.$$

*Under Conditions 2 and 5- 6,*

$$\widehat{\psi}_j^{\mathrm{RMTL},h}(\tau, a) \xrightarrow{P} \psi_{j,0}^{\mathrm{RMTL},h}(\tau, a), \quad \widehat{\psi}_j^{\mathrm{RMTL},h}(\tau) \xrightarrow{P} \psi_{j,0}^{\mathrm{RMTL},h}(\tau).$$

*If Condition 7 also holds, then convergence is uniform, i.e.*

$$\sup_{t \in [0,\tau]} \left| \widehat{\psi}_j^{\mathrm{RMTL},h}(\tau, a) - \psi_{j,0}^{\mathrm{RMTL},h}(\tau, a) \right| \xrightarrow{P} 0, \quad \sup_{t \in [0,\tau]} \left| \widehat{\psi}_j^{\mathrm{RMTL},h}(\tau) - \psi_{j,0}^{\mathrm{RMTL},h}(\tau) \right| \xrightarrow{P} 0.$$

The proof of Theorem 2 can be found in Appendix D.7. The following theorem shows that the proposed estimators are asymptotically linear both pointwise and uniformly, which enables construction of both pointwise confidence intervals and simultaneous confidence bands.

**Theorem 3** *(Asymptotic linearity) Under regularity conditions given in Appendix D, the proposed estimators $\widehat{\psi}^{\mathrm{RMST},h}(\tau, a)$ and $\widehat{\psi}_j^{\mathrm{RMTL},h}(\tau, a)$ are asymptotically linear for given $\tau$. Under slightly stronger regularity conditions, also in Appendix D, the asymptotic linearity is uniform over $\tau$.*

*(Efficiency bounds) Furthermore, under $\mathrm{var}\{\phi_0^{\mathrm{RMST}}(\tau, 1; \eta_0^\Lambda) \mid X\} = \mathrm{var}\{\phi_0^{\mathrm{RMST}}(\tau, 0; \eta_0^\Lambda) \mid X\}$ and $\mathrm{var}\{\phi_{j,0}^{\mathrm{RMTL}}(\tau, 1; \eta_{j,0}^\Lambda) \mid X\} = \mathrm{var}\{\phi_{j,0}^{\mathrm{RMTL}}(\tau, 0; \eta_{j,0}^\Lambda) \mid X\}$, the variance is minimized across the class of balancing IPTW estimators when $h_0(X) \propto \mathrm{var}_0(a \mid X)$.*

The proof of Theorem 3 can be found in Appendix D.8. Based on these asymptotic properties, we can construct pointwise confidence intervals and simultaneous confidence bands for the proposed

estimators. While our approach provides inference for population-level treatment effects, Davidov et al. (2025); Sesia & Svetnik (2025) offer lower prediction bounds (LPB) for survival times with distribution-free finite-sample guarantees at the individual level. The detailed inference procedures for our population-level estimators are provided in Appendix D.1.

Theorem 3 calls for the product of the nuisance functions in Equation 3 and Equation 4 (Appendix A.2) to converge at $n^{-1/2}$ rate. When not all nuisance functions are estimated consistently, the resulting variance estimators will be biased in unknown directions with unknown magnitude despite the point estimates being consistent owing to the double robustness property. In such scenarios, the bootstrap procedure may be able to reclaim proper coverage at the cost of significant computing time (Bai et al., 2013). Practical implementation of these theoretical results for constructing confidence intervals and bands is detailed in Appendix D.1.

## 5 SIMULATION STUDY

In this section, we conduct simulation studies to compare the proposed method with other estimators under good and poor overlap.

We evaluate finite-sample performance using four data generating processes with $n = 4000$ and 1000 replications each, crossing good/poor overlap conditions with absent/present competing risks (detailed specifications in Appendix B.1). The propensity score distributions in Figure 1 demonstrate the overlap conditions across settings.

The distributions of the true propensity scores by exposure arm are shown in Figure 1. We can observe that *Setting* 1 and *Setting* 3 share sufficient overlap while *Setting* 2 and *Setting* 4 have poor overlap.

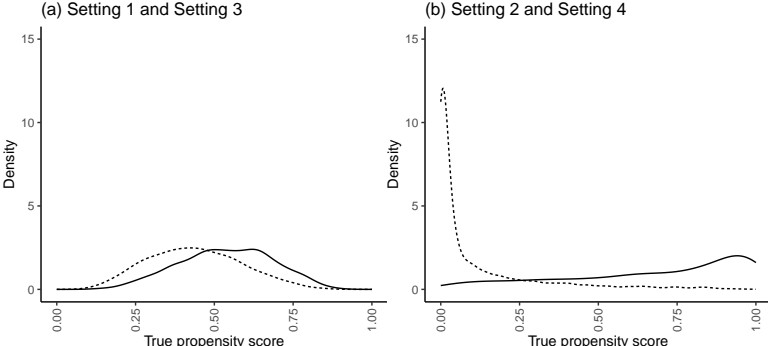

Figure 1: Density of true propensity score, note that the treatment models for the good overlap settings (*Setting* 1 and *Setting* 3) are identical, and the treatment models for the poor overlap settings (*Setting* 2 and *Setting* 4) are identical; dotted: treatment; solid: control.

We evaluate estimands for four types of balancing weights: ATE, ATO (overlap population), ATM (matched population), and ATEN (entropy weighted population) - see Table 2 in Appendix B for specifications. We compare outcome regression, IPCW, and doubly robust estimators with correctly specified parametric models against our proposed DML estimators. Though many deep neural network architectures have been developed to estimate censoring distributions (Zhong et al., 2021; Alaa & van der Schaar, 2017; Lee et al., 2018; Wu et al., 2023; Hu et al., 2021), we use SuperLearner (Van der Laan et al., 2007) ensembles to mitigate model misspecification (detailed in Appendix B).

Simulation results from the survival settings (*Setting* 1 and *Setting* 2) are shown here in Figure 2 and results from the competing risks setting (*Setting* 3 and *Setting* 4) can be found in Figure 6 in the Appendix B. The survival settings and the competing risks settings have similar interpretations and conclusions.

Figure 2 shows finite sample performance at restriction time $\tau = 4$. Our proposed estimators demonstrate competitive performance with low bias across all settings, while maintaining proper coverage probabilities (detailed analysis in Appendix B).

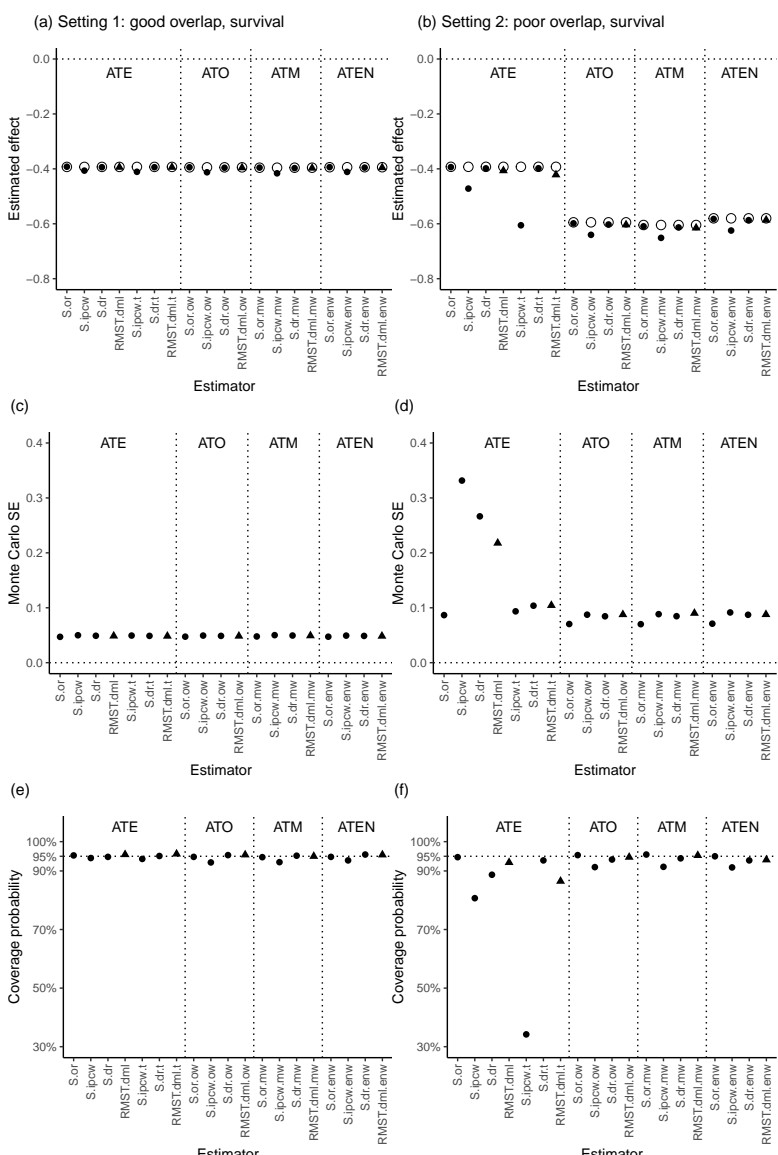

Figure 2: Estimated cumulative treatment effect, Monte Carlo standard errors, and coverage probabilities from different estimators grouped by estimand at $\tau = 4$. Estimands in each plot are separated by vertical dotted lines. The order of the estimands from left to right in each plot is: ATE, ATO, ATM, ATEN. Hollow circles in (a) and (b) represent true effects; solid circles represent the comparator estimators; and solid triangles represent the proposed estimators. Abbreviations of caption can be found in Table 3.

To summarize, we have demonstrated through simulation that our proposed estimators perform competitively with comparator estimators, even when the comparator estimators are given access to correct parametric nuisance model specifications and our methods must estimate the nuisance models via machine learning. This suggests that our proposed estimator is a much safer choice than alternatives, as the decreased risk of bias from model misspecification under our approach does not come at the cost of decreased efficiency.

# 6 APPLICATION

In this section, we apply the proposed methods to data from the Clinical Research Practice Datalink (CPRD) to assess the effects of first-line anti-hyperglycemia monotherapy on: (a) cancer incidence among type II diabetic patients with no history of cancer; and (b) on all-cause deaths among type II diabetic patients with cancer at time of initiation. The CPRD is a continuing general practice primary care database containing more than 60 million patients among 674 practices in the UK (Herrett et al., 2015). The study population included all type II diabetic patients aged over 50 before April 30, 2018. They were followed up until the earliest of death, leaving practice/CPRD database, or the end of data inclusion on April 30, 2018.

For simplicity and coherence, we follow Tsilidis et al. (2014) in our exposure and outcome definitions and covariate selection. We compare metformin initiators ($A = 1$) versus sulphonylurea initiators ($A = 0$). When the clinical outcome is all-cause mortality among cancer patients, then there is only one type of failure with no competing risks. However, when the outcome of interest is cancer incidence, mortality becomes a competing event. In both studies, we took baseline to be the time of initial first-line anti-hypoglycemia drug prescription. The failure times are subject to right censoring due to loss to follow-up or administrative end of follow up on April 30, 2018. The pretreatment confounders $X$ include demographic, clinical, and geographic variables (detailed in Appendix C). Compared with sulphonylurea, metformin starters are generally younger, heavier, and initiated later in calendar time.

We apply the proposed methods to estimate IPTW and overlap weighted RMST $\psi_0^{\text{RMST},h}(\tau, a)$ and cause-specific RMTL $\psi_{j,0}^{\text{RMTL},h}(\tau, a)$. Under the stated assumptions, these estimate counterfactual survival and cause-specific time lost had all participants received treatment $a$.

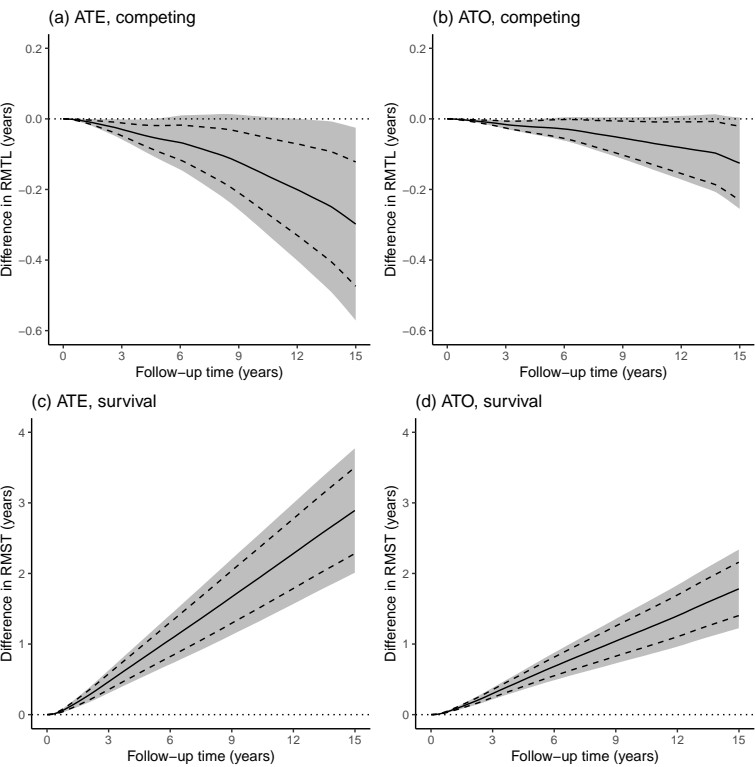

Figure 3: Curves of estimated cumulative treatment effect on the CPRD data; (a) cause-specific ATE of cancer incidence; (b) cause-specific ATO of cancer incidence; (c) ATE of mortality after cancer diagnosis; and (d) ATO of mortality after cancer diagnosis. In all figures, dotted line: $y = 0$; solid lines: point estimates; dashed lines: 95% pointwise confidence intervals; shaded area: 95% uniform simultaneous confidence bands.

As in our simulations, nuisance functions are again obtained by super learners (Van der Laan et al., 2007). We specify the ensemble nuisance learners in Appendix C. The propensity score distributions demonstrating limited overlap between treatment groups are shown in Figure 7 in Appendix C.

Figure 3 shows the cumulative treatment effects until restriction time $\tau = 15$ years. The first row shows the estimated cumulative cause-specific ATE and ATO of cancer incidence by the DML estimators. Estimates of cumulative ATE and ATO of all-cause mortality after cancer diagnosis are shown in the second row. These effects are the difference between metformin initiators and sulphonylurea initiators, respectively. The simultaneous confidence bands are constructed from $1,000$ realizations of sample paths of Gaussian processes with $[\tau_l, \tau_u] = [1/365.25, 7779/365.25]$ years for cancer incidence and $[\tau_l, \tau_u] = [2/365.25, 4533/365.25]$ years, which correspond to the maximum follow-up interval that the estimated standard errors are strictly positive. Our estimates indicate that starting metformin is protective over sulphonylurea on all cause deaths among cancer patients while its advantage on cancer incidence is marginal, which aligns with the conclusions from Morgan et al. (2014); Tsilidis et al. (2014). In each analysis, the point estimate of the ATO was closer to the null and had lower standard error than the point estimate of the ATE. We provide additional results in Appendix C.2.

## 7 DISCUSSION

To summarize, we have extended the DML approach to estimators of WATEs in survival settings. There is a broad literature on targeting WATEs as a response to poor overlap, and it is important to equip practitioners interested in these quantities with estimators that are not sensitive to misspecification of parametric nuisance models. Still, there remain interesting avenues for future work.

First, unlike the one-step estimator which targets the entire survival distribution (Cai & van der Laan, 2020; Rytgaard & van der Laan, 2023), the proposed estimators may still violate the constraints that the sum of RMST and RMTL from all causes is equal to $\tau$. One possible direction of future research is to ensure that this constraint and shape constraints are satisfied without parametric modeling assumptions and even when hazard models are misspecified.

Second, other estimands in the competing risks setting include the survivor average causal effect (SACE) (Tchetgen Tchetgen, 2014) and separable effects (Stensrud et al., 2021; 2022b;a). Future work can develop DML estimators for overlap weighted versions of these estimands. We conjecture that the regularity conditions for estimators of separable effects will be equivalent to ours as both estimators involve analogous nuisance functions (Martinussen & Stensrud, 2023).

## SUPPLEMENTARY MATERIAL

Supplementary material includes technical details of extended proofs, additional simulation information, and numerical results from application.

## ETHICS STATEMENT

This research uses de-identified UK primary care data (CPRD) authorized by the Independent Scientific Advisory Committee (ISAC protocol 20_000207). Potential risks include clinical misapplication of statistical methods and possible inequitable effects of overlap weighting on patient subgroups. Our methodological contributions are intended for statistical research, not direct clinical recommendations.

## REPRODUCIBILITY STATEMENT

Mathematical derivations, simulation specifications, and analysis procedures are detailed in the main text and appendices. While CPRD data cannot be shared due to privacy restrictions, our analysis code implementing the proposed estimators will be made available upon publication.

