# Supplementary material to "Efficient estimation of weighted cumulative treatment effects by double/debiased machine learning"

## LLM Usage Disclosure

Large language models were used to aid and polish writing for grammar and clarity improvements in select text portions. LLMs were not used for research design, methodology, analysis, or scientific content generation. All scientific work was conceived and executed entirely by the authors.

## A    Proposed methods

### A.1    Additional definitions

The all-cause conditional cumulative hazards, cause-specific conditional cumulative hazards, and censoring conditional cumulative hazards can be defined as $\Lambda(t \mid a, X) = -\int_0^t dS(u \mid a, X)/S(u \mid a, X)$, $\Lambda_j(t \mid a, X) = \int_0^t dF_j(u \mid a, X)/S(u \mid a, X)$, $\Lambda^C(t \mid a, X) = -\int_0^t dG(u \mid a, X)/G(u \mid a, X)$, then the all-cause martingale, cause-specific martingale, and censoring martingale can be defined as $dM(t \mid a, X) = dN(t) - R(t)d\Lambda(t \mid a, X)$, $dM_j(t \mid a, X) = dN_j(t) - R(t)d\Lambda_j(t \mid a, X)$, and $dM^C(t \mid a, X) = dN^C(t) - R(t)d\Lambda^C(t \mid a, X)$, respectively. Also denote $\eta_0^\Lambda = \{\pi_0(a \mid X), \Lambda_0^C(t \mid a, X), \Lambda_0(t \mid a, X)\}$, $\eta_0^{\Lambda,h} = \{h_0(X), \eta_0^\Lambda\}$, $\eta_{j,0}^\Lambda = \{\pi_0(a \mid X), \Lambda_0^C(t \mid a, X), \Lambda_0(t \mid a, X), \Lambda_{j,0}(t \mid a, X)\}$, and $\eta_{j,0}^{\Lambda,h} = \{h_0(X), \eta_{j,0}^\Lambda\}$.

### A.2    Implementation procedure of the proposed DML estimators

**Step 1** *Sample-splitting. We randomly split the data $O_1, \ldots, O_n$ into $K$ disjoint validation sets $\mathcal{V}_1, \ldots, \mathcal{V}_K$ with sizes $n_1, \ldots, n_K$, where $K \in \{2, 3, \ldots, \lfloor n/2 \rfloor\}$. For each $k = 1, \ldots, K$, we define training set $\overline{\mathcal{V}}_k = \{O_i : i \notin \mathcal{V}_k\}$ and treatment-specific training set $\overline{\mathcal{V}}_{a,k} = \{O_i : i \in \overline{\mathcal{V}}_k, A_i = a\}$.*

**Step 2** *Adaptive estimation. For every $k = 1, \ldots, K$, (1) train the propensity model on $\overline{\mathcal{V}}_k$ and predict $\widehat{\pi}_k(a \mid X)$ on $\mathcal{V}_k$; (2) train the time-to-event models on $\overline{\mathcal{V}}_{a=0,k}$ and $\overline{\mathcal{V}}_{a=1,k}$ separately, and evaluate $\widehat{\Lambda}_k(t \mid a, X)$, $\widehat{\Lambda}_k^C(t \mid a, X)$, and perhaps $\widehat{\Lambda}_{j,k}(t \mid a, X)$ for $a = 0, 1$ using $\mathcal{V}_k$ at unique observed survival times $\widetilde{T}$ in $\mathcal{V}_k$ to capture jump information from observed counting processes $dN(t)$, $dN^C(t)$, and perhaps $dN_j(t)$. Counting process martingales can be obtained by directly plugging in corresponding cumulative hazards.*

**Step 3** *Asymptotically efficient estimators. Construct $K$ estimators $\widehat{\phi}_k^{\mathrm{RMST},h}(\tau, a; \widehat{\eta}_k^{\Lambda,h})$ or $\widehat{\phi}_{j,k}^{\mathrm{RMTL},h}(\tau, a; \widehat{\eta}_{j,k}^{\Lambda,h})$ by plugging estimated nuisance functions from Step 2 into*

$$\widehat{\phi}_k^{\mathrm{RMST},h}(\tau, a; \widehat{\eta}_k^{\Lambda,h}) = [P_{n,k}\{\widehat{h}_k(X)\}]^{-1}\widehat{h}_k(X)\widehat{\mathrm{RMST}}_k(\tau \mid a, X) + [P_{n,k}\{\widehat{w}_k^h(a, X)\}]^{-1}\widehat{w}_k^h(a, X)$$

$$\times \left\{ \frac{\Delta(\tau)\min(\widetilde{T}, \tau)}{\widehat{G}_k(\tau \wedge \widetilde{T}- \mid a, X)} - \widehat{\mathrm{RMST}}_k(\tau \mid a, X) + \int_0^{\tau \wedge \widetilde{T}} \frac{t\mathrm{d}\widehat{M}_k^C(t \mid a, X)}{\widehat{G}_k(t- \mid a, X)} \right. \quad (3)$$

$$\left. \int_0^{\tau \wedge \widetilde{T}} \frac{\{\widehat{\mathrm{RMST}}_k(\tau \mid a, X) - \widehat{\mathrm{RMST}}_k(t \mid a, X)\}\mathrm{d}M_0^C(t \mid a, X)}{S_k(t \mid a, X)G_k(t- \mid a, X)} \right\}$$

*or*

$$\widehat{\phi}_{j,k}^{\mathrm{RMTL},h}(\tau, a; \widehat{\eta}_{j,k}^{\Lambda,h}) = [P_{n,k}\{\widehat{h}_k(X)\}]^{-1}\widehat{h}_k(X)\widehat{\mathrm{RMTL}}_{j,k}(\tau \mid a, X) + [P_{n,k}\{\widehat{w}_k^h(a, X)\}]^{-1}\widehat{w}_k^h(a, X)$$

$$\times \left\{ \frac{(\tau - \min(\widetilde{T}, \tau))I(\widetilde{J} = j)}{\widehat{G}_k(\widetilde{T}- \mid a, X)} - \widehat{\mathrm{RMTL}}_{j,k}(\tau \mid a, X) \right.$$

$$- \int_0^{\tau \wedge \widetilde{T}} \frac{(\tau - t)\widehat{F}_{j,k}(t \mid a, X)\mathrm{d}\widehat{M}_k^C(t \mid a, X)}{\widehat{S}_k(t \mid a, X)\widehat{G}_k(t- \mid a, X)}$$

$$\left. + \int_0^{\tau \wedge \widetilde{T}} \frac{\{\widehat{\mathrm{RMTL}}_{j,k}(\tau \mid a, X) - \widehat{\mathrm{RMTL}}_{j,k}(t \mid a, X)\}\mathrm{d}\widehat{M}_k^C(t \mid a, X)}{\widehat{S}_k(t \mid a, X)\widehat{G}_k(t- \mid a, X)} \right\}$$

$$(4)$$

*Combine k-folds together as $\widehat{\phi}^{\mathrm{RMST},h}(\tau, a; \widehat{\eta}^{\Lambda,h})$ or $\widehat{\phi}_j^{\mathrm{RMTL},h}(\tau, a; \widehat{\eta}_j^{\Lambda,h})$ where $\widehat{\eta}^{\Lambda,h} = \{\widehat{\eta}_{k=1}^{\Lambda,h}, \dots, \widehat{\eta}_{k=K}^{\Lambda,h}\}$ and $\widehat{\eta}_j^{\Lambda,h} = \{\widehat{\eta}_{j,k=1}^{\Lambda,h}, \dots, \widehat{\eta}_{j,k=K}^{\Lambda,h}\}$.*

*To reduce sampling variation in Equation 3 and Equation 4, balancing weights used in their numerators are averaged to be their denominators to control for numerical instability (Hájek, 1971). However, we do not further regulate numerical instability for the IPCW term using $P_{n,k}[\{\Delta(\tau)/\widehat{G}_k(\tau \wedge \widetilde{T}- \mid a, X)\}]^{-1}$ in Equation 3 or $P_{n,k}[\{\Delta/\widehat{G}_k(\widetilde{T}- \mid a, X)\}]^{-1}$ in Equation 4. Note that the one-step estimators and the estimating equation estimators are equivalent as the $\varphi_0^{\mathrm{RMST},h}(\tau, a; \eta_0^{\Lambda,h})$ and $\varphi_{j,0}^{\mathrm{RMTL},h}(\tau, a; \eta_{j,0}^{\Lambda,h})$ are linear in $\psi_0^{\mathrm{RMST},h}(\tau, a)$ and $\psi_{j,0}^{\mathrm{RMST},h}(\tau, a)$ (Kennedy, 2022). Time-to-event nuisance functions require conditional hazard estimation (see Appendix A.3 for details). The proposed estimators naturally satisfy shape constraints asymptotically (see Appendix A.3).*

**Step 4** *Shape-corrected point estimators.* Compute $\widehat{\psi}^{\mathrm{RMST},h}(\tau, a) = P_n\{\widehat{\phi}^{\mathrm{RMST},h}(\tau, a; \widehat{\eta}^{\Lambda,h})\}$ and $\widehat{\psi}_j^{\mathrm{RMTL},h}(\tau, a) = P_n\{\widehat{\phi}_j^{\mathrm{RMTL},h}(\tau, a; \widehat{\eta}_j^{\Lambda,h})\}$. *For each $a \in \{0, 1\}$, we project $\widehat{\psi}^{\mathrm{RMST},h}(\tau, a)$ onto a space of concave functions using least concave majorant to obtain $\widehat{\psi}^{\mathrm{RMST},h,lcm}(\tau, a)$ and the bounded $\widehat{\psi}_j^{\mathrm{RMTL},h}(\tau, a)$ onto a space of convex functions using greatest convex minorant to acquire $\widehat{\psi}_j^{\mathrm{RMTL},h,gcm}(\tau, a)$. Then*

$$\widehat{\psi}^{\mathrm{RMST},h,lcm}(\tau) = \widehat{\psi}^{\mathrm{RMST},h,lcm}(\tau, 1) - \widehat{\psi}^{\mathrm{RMST},h,lcm}(\tau, 0)$$
$$\widehat{\psi}_j^{\mathrm{RMTL},h,gcm}(\tau) = \widehat{\psi}_j^{\mathrm{RMTL},h,gcm}(\tau, 1) - \widehat{\psi}_j^{\mathrm{RMTL},h,gcm}(\tau, 0)$$

**Step 5** *Shape-corrected estimators for asymptotic standard errors.*

$$\widehat{\varphi}^{\mathrm{RMST},h}(\tau, a; \widehat{\eta}^{\Lambda,h}) = \widehat{\phi}^{\mathrm{RMST},h}(\tau, a; \widehat{\eta}^{\Lambda,h}) - \widehat{\psi}^{\mathrm{RMST},h,lcm}(\tau, a)$$
$$\widehat{\varphi}_j^{\mathrm{RMTL},h}(\tau, a; \widehat{\eta}_j^{\Lambda,h}) = \widehat{\phi}_j^{\mathrm{RMTL},h}(\tau, a; \widehat{\eta}_j^{\Lambda,h}) - \widehat{\psi}_j^{\mathrm{RMTL},h,gcm}(\tau, a)$$
$$\widehat{\varphi}^{\mathrm{RMST},h}(\tau; \widehat{\eta}^{\Lambda,h}) = \widehat{\varphi}^{\mathrm{RMST},h}(\tau, 1; \widehat{\eta}^{\Lambda,h}) - \widehat{\varphi}^{\mathrm{RMST},h}(\tau, 0; \widehat{\eta}^{\Lambda,h})$$
$$\widehat{\varphi}_j^{\mathrm{RMTL},h}(\tau; \widehat{\eta}_j^{\Lambda,h}) = \widehat{\varphi}_j^{\mathrm{RMTL},h}(\tau, 1; \widehat{\eta}_j^{\Lambda,h}) - \widehat{\varphi}_j^{\mathrm{RMTL},h}(\tau, 0; \widehat{\eta}_j^{\Lambda,h})$$
$$\widehat{\sigma}^{\mathrm{RMST},h}(\tau, a) = [P_n\{\widehat{\varphi}^{\mathrm{RMST},h}(\tau, a; \widehat{\eta}^{\Lambda,h})\}^2]^{1/2}, \quad \widehat{\sigma}_j^{\mathrm{RMTL},h}(\tau, a) = [P_n\{\widehat{\varphi}_j^{\mathrm{RMTL},h}(\tau, a; \widehat{\eta}_j^{\Lambda,h})\}^2]^{1/2}$$
$$\widehat{\sigma}^{\mathrm{RMST},h}(\tau) = [P_n\{\widehat{\varphi}^{\mathrm{RMST},h}(\tau; \widehat{\eta}^{\Lambda,h})\}^2]^{1/2}, \quad \widehat{\sigma}_j^{\mathrm{RMTL},h}(\tau) = [P_n\{\widehat{\varphi}_j^{\mathrm{RMTL},h}(\tau; \widehat{\eta}_j^{\Lambda,h})\}^2]^{1/2}$$

*When parametric models are employed to estimate nuisance parameters, the asymptotic expansion of the resulting estimator becomes the original plug-in influence function plus a function related to the influence functions of the nuisance models (Ozenne et al., 2020). The sandwich variance of the plug-in estimator may be underestimated when this additional term that reflects nuisance model uncertainty is omitted.*

*We project $\widehat{\sigma}^{\mathrm{RMST},h}(\tau, a)$, $\widehat{\sigma}^{\mathrm{RMST},h}(\tau)$, $\widehat{\sigma}_j^{\mathrm{RMTL},h}(\tau, a)$ and $\widehat{\sigma}_j^{\mathrm{RMTL},h}(\tau)$ onto a space of non-decreasing functions by taking their cumulative maximum and denote as*

$$\widehat{\sigma}^{\mathrm{RMST},h,+}(\tau, a) = \max_{t \le \tau} \widehat{\sigma}^{\mathrm{RMST},h}(t, a), \quad \widehat{\sigma}_j^{\mathrm{RMTL},h,+}(\tau, a) = \max_{t \le \tau} \widehat{\sigma}_j^{\mathrm{RMTL},h}(t, a)$$
$$\widehat{\sigma}^{\mathrm{RMST},h,+}(\tau) = \max_{t \le \tau} \widehat{\sigma}^{\mathrm{RMST},h}(t), \quad \widehat{\sigma}_j^{\mathrm{RMTL},h,+}(\tau) = \max_{t \le \tau} \widehat{\sigma}_j^{\mathrm{RMTL},h}(t)$$

*Eventually, the finite sample standard errors of $\psi^{\mathrm{RMST},h}(\tau, a)$, $\psi^{\mathrm{RMST},h}(\tau)$, $\psi_j^{\mathrm{RMTL},h}(\tau, a)$, and $\psi_j^{\mathrm{RMTL},h}(\tau)$ can be calculated as $\widehat{\sigma}^{\mathrm{RMST},h,+}(\tau, a)/\sqrt{n}$, $\widehat{\sigma}^{\mathrm{RMST},h,+}(\tau)/\sqrt{n}$, $\widehat{\sigma}_j^{\mathrm{RMTL},h,+}(\tau, a)/\sqrt{n}$, and $\widehat{\sigma}_j^{\mathrm{RMTL},h,+}(\tau)/\sqrt{n}$.*

### A.3 REMARKS

**Conditional hazard estimation.** Below are some identities between conditional uncensored time-to-event distributions and conditional censored time-to-event distributions weighted by IPCW.

$$S(t \mid a, x) = \frac{P(C > t \mid A = a, X = x)P(T > t \mid A = a, X = x)}{P(C > t \mid A = a, X = x)}$$

$$= \frac{P(\widetilde{T} > t \mid A = a, X = x)}{G(t \mid a, x)} = E\left\{\frac{I(\widetilde{T} > t)}{G(t \mid a, x)} \mid A = a, X = x\right\}$$

$$S(t \mid a, x) = \frac{E(\Delta \mid A = a, X = x)E\{I(T > t) \mid A = a, X = x\}}{G(\widetilde{T}- \mid a, x)}$$

$$= \frac{E\{\Delta I(\widetilde{T} > t) \mid A = a, X = x\}}{G(\widetilde{T}- \mid a, x)} = E\left\{\frac{\Delta I(\widetilde{T} > t)}{G(\widetilde{T}- \mid a, x)} \mid A = a, X = x\right\}$$

$$\mathrm{RMST}(\tau \mid a, x) = \frac{E\{\Delta(\tau) \mid A = a, X = x\}E\{\min(T, \tau) \mid A = a, X = x\}}{G(\tau \wedge \widetilde{T}- \mid a, x)}$$

$$= \frac{E\{\Delta(\tau)\min(\widetilde{T}, \tau) \mid A = a, X = x\}}{G(\tau \wedge \widetilde{T}- \mid a, x)} = E\left\{\frac{\Delta(\tau)\min(\widetilde{T}, \tau)}{G(\tau \wedge \widetilde{T}- \mid a, x)} \mid A = a, X = x\right\}$$

$$F_j(t \mid a, x) = \frac{E(\Delta \mid A = a, X = x)E\{I(T \le t, J = j) \mid A = a, X = x\}}{G(\widetilde{T}- \mid a, x)}$$

$$= \frac{E\{I(\widetilde{T} \le t, \widetilde{J} = j) \mid A = a, X = x\}}{G(\widetilde{T}- \mid a, x)} = E\left\{\frac{I(\widetilde{T} \le t, \widetilde{J} = j)}{G(\widetilde{T}- \mid a, x)} \mid A = a, X = x\right\}$$

$$\mathrm{RMTL}_j(\tau \mid a, x) = \frac{E(\Delta \mid A = a, X = x)E[\{\tau - \min(T, \tau)\}I(J = j) \mid A = a, X = x]}{G(\widetilde{T}- \mid a, x)}$$

$$= \frac{E[\{\tau - \min(\widetilde{T}, \tau)\}I(\widetilde{J} = j) \mid A = a, X = x]}{G(\widetilde{T}- \mid a, x)}$$

$$= E\left[\frac{\{\tau - \min(\widetilde{T}, \tau)\}I(\widetilde{J} = j)}{G(\widetilde{T}- \mid a, x)} \mid A = a, X = x\right]$$

Build pooled classification or regression models by taking observable $I(\widetilde{T} > t)$, $I(\widetilde{T} > t)/G(t \mid A, X)$, $\Delta I(\widetilde{T} > t)$, $\Delta I(\widetilde{T} > t)/G(\widetilde{T}- \mid A, X)$, $\Delta(\tau)\min(\widetilde{T}, \tau)$, $\Delta(\tau)\min(\widetilde{T}, \tau)/G(\tau \wedge \widetilde{T}- \mid A, X)$, $I(\widetilde{T} \le t, \widetilde{J} = j)$, $I(\widetilde{T} \le t, \widetilde{J} = j)/G(\widetilde{T}- \mid A, X)$, $\{\tau - \min(\widetilde{T}, \tau)\}I(\widetilde{J} = j)$, and $\{\tau - \min(\widetilde{T}, \tau)\}I(\widetilde{J} = j)/G(\widetilde{T}- \mid A, X)$ as response and making $A$, $X$ and $t$ or $\tau$ as features has three major drawbacks.

First of all, converting data from a wide format to a long format with $t$ or $\tau$ as pivot only allows discrete-time survival times (Stitelman & van der Laan, 2010). Coarsening continous survival times into discrete often leads to loss of information and accuracy. Second, training the long format requires much more space and can be very time-consuming. Last, but not the least, most standard statistical learning algorithms require data are i.i.d. However, this type of "counting process learning" keeps all rows even after a subject leaves the risk set, which leaves strong dependency between rows in the long format. On the other hand, discrete-time conditional hazard type classification satisfies the i.i.d assumption. By definition,

$$\lambda(t \mid a, x) = P(T = t \mid T \ge t, A = a, X = x) = \frac{P(T = t, T \ge t, A = a, X = x)}{P(T \ge t, A = a, X = x)}$$

$$= \frac{P(T = t, A = a, X = x)P(C \ge t, A = a, X = x)}{P(T \ge t, A = a, X = x)P(C \ge t, A = a, X = x)} = \frac{P(\widetilde{T} = t, T \le C, A = a, X = x)}{P(\widetilde{T} \ge t, A = a, X = x)}$$

$$= \frac{P(\widetilde{T} = t, \Delta, \widetilde{T} \ge t, A = a, X = x)}{P(\widetilde{T} \ge t, A = a, X = x)} = P(\widetilde{T} = t, \Delta \mid \widetilde{T} \ge t, A = a, X = x)$$

The condition in the last probability above assures units who left the risk set are not trained any more such that for all units at risk, whether each unit has the event at each time point becomes i.i.d. Censoring hazards and cause-specific hazards follow a similar derivation and reasoning.

**Shape constraints.** $\psi_0^{\mathrm{RMST},h}(\tau, a)$ and $\psi_{j,0}^{\mathrm{RMTL},h}(\tau, a)$ are naturally concave and convex functions with respect to $\tau$ as their derivatives are monotonically non-increasing survival functions and non-decreasing cause-specific cumulative incidence functions. Aside from shape constraints on point estimates, asymptotic standard

errors $\sigma_0^{\mathrm{RMST},h}(\tau,a)$, $\sigma_0^{\mathrm{RMST},h}(\tau)$, $\sigma_{j,0}^{\mathrm{RMTL},h}(\tau,a)$ and $\sigma_{j,0}^{\mathrm{RMTL},h}(\tau)$ are strictly non-decreasing as censoring is always monotonically coarsened with respect to follow-up time.

However, the proposed estimators are not guaranteed to be shape-restricted in any finite sample since estimates are separately made at each time point without any global constraints. In this paper, we ensure convexity/concavity of point estimates using Grenander-type estimators (Westling & Carone, 2020) and monotonicity of asymptotic variance using projection-type estimators (Daouia & Park, 2013). Since shape-corrected estimators are guaranteed to be no further from the truth than the unrestricted estimators in all finite samples, and both estimators are asymptotically equivalent, we focus on large-sample properties of the uncorrected estimators.

**Multiple treatments.** The proposed estimators can easily be extended to multiple treatments $A \in \{0, 1, 2, ...\}$. One can simply cross-fit multi-class classifiers for $\widehat{\pi}_k(a \mid X)$ and hazard models to obtain $\widehat{\Lambda}_k(t \mid a, X)$, $\widehat{\Lambda}_k^C(t \mid a, X)$, and perhaps $\widehat{\Lambda}_{j,k}(t \mid a, X)$ for every treatment $a$.

**Mixed survival times.** Similar to Westling et al. (2023), the proposed method allows for continuous, discrete, and mixed survival times. One can simply replace continuous time hazards and martingale integrals by their discrete or mixed survival time equivalents.

# B  SIMULATION STUDY

## B.1  DATA GENERATING PROCESSES

We studied four data generating processes to evaluate the finite-sample performance of comparator and proposed estimators on the ATE and WATE. The simulations resemble observational studies under four settings: setting 1 - good overlap and no competing risks; setting 2 - poor overlap and no competing risks; setting 3 - good overlap with competing risks; setting 4 - poor overlap with competing risks. We generate 1000 replications for each setting. Within each replicate, consider six covariates $X_1$-$X_6$ following practices from Cheng et al. (2022). The covariates $X_4$, $X_5$, and $X_6$ are drawn independently from a Bernoulli distribution with 50% probability. The covariates $X_1$, $X_2$, and $X_3$ are generated from a multivariate normal distribution with mean zero, unit variance, and 0.5 pairwise correlation.

*Setting* 1 (good overlap, no competing risks): We consider a sample size of $n = 4000$. The exposure is generated from a Bernoulli distribution with propensity score

$$\mathrm{expit}(-(0.3 + 0.2X_1 + 0.3X_2 + 0.3X_3 - 0.2X_4 - 0.3X_5 - 0.2X_6))$$

We generate potential survival times $T^{a=0}$ and $T^{a=1}$ as well as potential censoring times $C^{a=0}$ and $C^{a=1}$ from exponential distribution with hazards

$$\lambda^{a=0}(t \mid X) = 0.12 \exp\{0.1 + 0.1X_1 - 0.2X_2 + 0.2X_3 + 0.1X_4 + 0.8X_5 - 0.2X_6\}$$
$$\lambda^{a=1}(t \mid X) = 0.15 \exp\{0.17 + 0.2X_1 - 0.1X_2 + 0.4X_3 + 0.2X_4 + 0.3X_5 + 0.4X_6\}$$
$$\lambda^C(t \mid A = 0, X) = 0.06 \exp\{0.1 + 0.4X_1 - 0.7X_2 - 0.4X_3 - 0.5X_4 + 0.8X_5 - 0.6X_6\}$$
$$\lambda^C(t \mid A = 1, X) = 0.08 \exp\{0.5X_1 - 0.6X_2 + 0.2X_3 + 0.6X_4 + 0.9X_5 - 0.5X_6\}$$

*Setting* 2 (poor overlap, no competing risks): We consider a sample size of $n = 4000$. The exposure is generated from a Bernoulli distribution with propensity score

$$\mathrm{expit}(-(-1 + X_1 + 1.5X_2 + 1.5X_3 - X_4 - 1.5X_5 - X_6))$$

Potential survival and censoring times are generated the same as *Setting* 1.

*Setting* 3 (good overlap, with competing risks): We consider a sample size of $n = 4000$. The exposure is simulated the same as *Setting* 1. Potential failure times of two causes and censoring times are generated with hazards

$$\lambda_{j=1}^{a=0}(t \mid X) = 0.12 \exp\{0.1 + 0.1X_1 - 0.2X_2 + 0.2X_3 + 0.1X_4 + 0.8X_5 - 0.2X_6\}$$
$$\lambda_{j=1}^{a=1}(t \mid X) = 0.15 \exp\{0.17 + 0.2X_1 - 0.1X_2 + 0.4X_3 + 0.2X_4 + 0.3X_5 + 0.4X_6\}$$
$$\lambda_{j=2}^{a=0}(t \mid X) = 0.1 \exp\{0.12 - 0.1X_1 + 0.3X_2 + 0.1X_3 + 0.2X_4 - 0.4X_5 + 0.5X_6\}$$
$$\lambda_{j=2}^{a=1}(t \mid X) = 0.08 \exp\{0.1 - 0.2X_1 - 0.1X_2 + 0.2X_3 + 0.3X_4 + 0.3X_5 - 0.3X_6\}$$
$$\lambda^C(t \mid A = 0, X) = 0.12 \exp\{0.1 + 0.4X_1 - 0.7X_2 - 0.4X_3 - 0.5X_4 + 0.8X_5 - 0.6X_6\}$$
$$\lambda^C(t \mid A = 1, X) = 0.14 \exp\{0 + 0.5X_1 - 0.6X_2 + 0.2X_3 + 0.6X_4 + 0.9X_5 - 0.5X_6\}$$

Cause 1 is of our interest.

*Setting* 4 (poor overlap, with competing risks): We consider a sample size of $n = 4000$. The exposure is simulated the same as *Setting* 2. Potential failure times of two causes and censoring times are generated the same as *Setting* 3. Cause 1 is of our interest.

The summary statistics of the simulated data sets can be found in Table 1.

## B.2 SUMMARY STATISTICS

Summary statistics, including treatment prevalence, censoring probability, average rate of violation of proportional hazards assumption from 1000 simulations, and event rate before $\tau = 4$, of *Setting* 1-4 is tabulated in Table 1.

Table 1: Summary of data generating mechanisms *Setting* 1-*Setting* 4. %PH violation: percentage of violation of proportional hazards (PH) assumption, 4 hazards models ($\lambda^{a=0}, \lambda^{a=1}, \lambda^{a=0,C}, \lambda^{a=1,C}$) for *Setting* 1 and *Setting* 2 and 6 hazards models ($\lambda^{a=0}_{j=1}, \lambda^{a=1}_{j=1}, \lambda^{a=0}, \lambda^{a=1}, \lambda^{a=0,C}, \lambda^{a=1,C}$) for *Setting* 3 and *Setting* 4. Event rate before $\tau = 4$: number of event-of-interest that happened before $\tau = 4$ divided by $n$.

|  | *Setting* 1 | *Setting* 2 | *Setting* 3 | *Setting* 4 |
|---|---|---|---|---|
| Percentage of treatment | 49% | 25% | 49% | 25% |
| Censoring probability | 35.7% | 35.4% | 35.9% | 33.7% |
| %PH violation | 33.3% | 29.8% | 8.2% | 13.7% |
| Event rate before $\tau = 4$ | 48.7% | 50.2% | 36.2% | 37.6% |

## B.3 ESTIMANDS AND COMPARATOR ESTIMATORS

We consider estimands corresponding to four types of balancing weights summarized in Table 2. Specification of $h(x)$ defines the target parameter and weights.

Table 2: Estimands, tilting functions, and their origins. ATO: Average treatment effect among the overlap population (Li et al., 2018); ATM: Average treatment effect among the matched population (Li & Greene, 2013); ATEN: Average treatment effect of entropy weighted population (Zhou et al., 2020)

| Estimands | Tilting function $h(x)$ | Origin |
|---|---|---|
| ATE | 1 | Survey |
| ATO | $\pi(0 \mid x)\pi(1 \mid x)$ | Gini index |
| ATM | $\min_{a=0,1} \{\pi(a \mid x)\}$ | Misclassification error |
| ATEN | $-\sum_{a=0,1} \pi(a \mid x) \log \pi(a \mid x)$ | Cross-entropy or deviance |

Some of these comparator estimators haven't been explicitly proposed, and we spell out these estimators here for completeness. In the survival setting, the g-formula estimator using outcome regression is

$$\widehat{\psi}^{S,\mathrm{OR},h}(t,a) = [P_n\{\widehat{h}(X)\}]^{-1} P_n[\widehat{h}(X)\{\widehat{S}(t \mid a, X)\}]$$

and the IPCW estimator (Cheng et al., 2022; Kvamme & Borgan, 2023) is

$$\widehat{\psi}^{S,\mathrm{IPCW},h}(t,a) = [P_n\{\widehat{w}^h(a,X)\}]^{-1} P_n \left\{ \frac{\widehat{w}^h(a,X)I(\widetilde{T} > t)}{\widehat{G}(t \mid a, X)} \right\}$$

Another IPCW estimator can be constructed as

$$\widehat{\psi}^{S,\mathrm{IPCW},h}(t,a) = [P_n\{\widehat{w}^h(a,X)\}]^{-1} P_n \left\{ \frac{\widehat{w}^h(a,X)\Delta I(\widetilde{T} > t)}{\widehat{G}(\widetilde{T}- \mid a, X)} \right\}$$

The doubly robust estimator takes the form

$$\widehat{\psi}^{S,\mathrm{DR},h}(t,a) = [P_n\{\widehat{h}(X)\}]^{-1} P_n[\widehat{h}(X)\widehat{S}(t \mid a, X)] + [P_n\{\widehat{w}^h(a,X)\}]^{-1} P_n \left[ \widehat{w}^h(a,X) \right.$$

$$\left. \times \left\{ \frac{I(\widetilde{T} > t)}{\widehat{G}(t \mid a, X)} - \widehat{S}(t \mid a, X) + \widehat{S}(t \mid a, X) \int_0^{t \wedge \widetilde{T}} \frac{\mathrm{d}\widehat{M}^C(u \mid a, X)}{\widehat{S}(u \mid a, X)\widehat{G}(u- \mid a, X)} \right\} \right]$$

This is a right continuous function with respect to $t$ while Bai et al. (2013) provided a left continuous version. As the weighted group-specific survival functions are monotonically non-increasing, we take their cumulative minimum and denote as $\widehat{\psi}^{S,h,-}(t,a) = \min_{t \leq \tau} \widehat{\psi}^{S,h}(t,a)$. The causal contrasts are $\widehat{\psi}^{S,h}(t) = \widehat{\psi}^{S,h,-}(t,1) - \widehat{\psi}^{S,h,-}(t,0)$. Then $\widehat{\psi}^{\mathrm{RMST},h}(\tau)$ can be obtained by integrating each estimator above, $\int_0^\tau \widehat{\psi}^{S,h}(t)\mathrm{d}t$.

In the competing risks setting, the g-formula estimator using outcome regressions for the cause-specific CIF is,

$$\widehat{\psi}_j^{F,\mathrm{OR},h}(t,a) = [P_n\{\widehat{h}(X)\}]^{-1} P_n[\widehat{h}(X)\{\widehat{F}_j(t \mid a, X)\}]$$

and the IPCW estimator is

$$\widehat{\psi}_j^{F,\mathrm{IPCW},h}(t,a) = [P_n\{\widehat{w}^h(a,X)\}]^{-1} P_n\left\{ \frac{\widehat{w}^h(a,X) I(\widetilde{T} \leq t, \widetilde{J} = j)}{\widehat{G}(\widetilde{T}- \mid a, X)} \right\}$$

The doubly robust estimator takes the form

$$\widehat{\psi}_j^{F,\mathrm{DR},h}(t,a) = [P_n\{\widehat{h}(X)\}]^{-1} P_n[\widehat{h}(X)\widehat{F}_j(t \mid a, X)] + [P_n\{\widehat{w}^h(a,X)\}]^{-1} P_n \left[ \widehat{w}^h(a,X) \right.$$

$$\times \left\{ \frac{I(\widetilde{T} \leq t, \widetilde{J} = j)}{\widehat{G}(\widetilde{T}- \mid a, X)} - \widehat{F}_j(t \mid a, X) + \widehat{F}_j(t \mid a, X) \int_0^{t \wedge \widetilde{T}} \frac{\mathrm{d}\widehat{M}^C(u \mid a, X)}{\widehat{S}(u \mid a, X)\widehat{G}(u- \mid a, X)} \right.$$

$$\left. \left. - \int_0^{t \wedge \widetilde{T}} \frac{\widehat{F}_j(u \mid a, X)\mathrm{d}\widehat{M}^C(u \mid a, X)}{\widehat{S}(u \mid a, X)\widehat{G}(u- \mid a, X)} \right\} \right]$$

which is similar to Ozenne et al. (2020). As the weighted group-specific cause-specific CIFs are monotonically non-decreasing, we take their cumulative maximum and denote as $\widehat{\psi}_j^{F,h,+}(t,a) = \max_{t \leq \tau} \widehat{\psi}_j^{F,h}(t,a)$. The causal contrasts are $\widehat{\psi}_j^{F,h}(t) = \widehat{\psi}_j^{F,h,+}(t,1) - \widehat{\psi}_j^{F,h,+}(t,0)$. Then $\widehat{\psi}_j^{\mathrm{RMTL},h}(\tau)$ can be obtained by integrating each estimator above, $\int_0^\tau \widehat{\psi}_j^{F,h}(t)\mathrm{d}t$.

## B.4 ADDITIONAL SIMULATION RESULTS

### B.4.1 DETAILED ANALYSIS OF SIMULATION PERFORMANCE

Figure 2 (a) and (b) show the estimated weighted cumulative treatment effects and true effects grouped by estimand. The true ATE and WATE are similar when the overlap is adequate (Figure 2 (a)), while the difference between ATE and WATE is large when the overlap is poor (Figure 2 (b)). The bias (hollow vs solid) is generally quite low under adequate overlap (Figure 2 (a)) but is occasionally high under poor overlap (Figure 2 (b)). For each estimand, even with correctly specified nuisance models, the IPCW estimators have larger finite sample bias compared with other estimators, and this is exacerbated as overlap deteriorates. When overlap is limited, Winsorizing the IPTW at its 99% percentile significantly increases the bias. In addition, it can be observed that estimators with (correctly specified) outcome models have less bias than those without. In all settings, the proposed estimators appear to be approximately unbiased.

Figure 2 (c) and (d) show Monte Carlo standard errors grouped by estimand at restriction time $\tau = 4$. The Monte Carlo standard errors agree nicely with estimated standard errors. When overlap is sufficient, the standard errors are similar across estimands and estimators. However, estimators for WATE are more efficient than estimators for ATE when overlap is limited, while Winsorization improves efficiency for ATE (at the cost of bias, as we saw in Figure 2 (b)). Within each estimand in the limited overlap setting, it is obvious that the outcome regression estimators are the most efficient, followed by the doubly robust estimators and the proposed estimators, while the IPCW estimators have larger finite sample variance. This demonstrates that a proper augmentation term can restore large amount of information which leads to efficiency gains.

This is expected because the doubly robust estimators and the proposed estimators are the most efficient among the class of IPTW estimators, but they are slightly less efficient than the outcome regression estimators (Robins et al., 1994; Bai et al., 2013). Of course, this increased efficiency would be at the cost of bias if the outcome model is misspecified, a concern which is greatly mitigated by the data adaptive nuisance model estimation in our estimators.

Figure 2 (e) and (f) show the empirical coverage of 95% pointwise confidence intervals grouped by estimand at restriction time $\tau = 4$. When overlap is sufficient, Figure 2 (e) reveals that the comparator and the proposed estimators have empirical coverage probabilities quite close to the nominal value of 0.95. However, when overlap is poor, the IPCW estimators are undercovered more than other estimators within each estimand, shown in Figure 2 (f). Winsorizing the IPTW lowers the coverage since it has larger bias and smaller estimated

variance. When overlap is limited, the outcome regression estimators, the doubly robust estimators, and the proposed estimators all have decent coverage close to 0.95 for WATEs.

Table 3: Abbreviation mapping of comparator estimators in figures

| Abbreviation | Stands for |
|---|---|
| RMST | restricted mean survival time |
| RMTLj | restricted mean time lost of cause $j$ |
| S | obtain RMST by integrating the survival function |
| Fj | obtain RMTL$j$ by integrating the cumulative incidence function |
| or | outcome regression estimator |
| ipcw | IPCW estiamtor |
| dml | double/debiased machine learning estimator |
| dr | doubly robust |
| .t | IPTW Winsorized at its 99th percentile |
| ow | overlap weighted |
| mw | matching weighted |
| enw | entropy weighted |

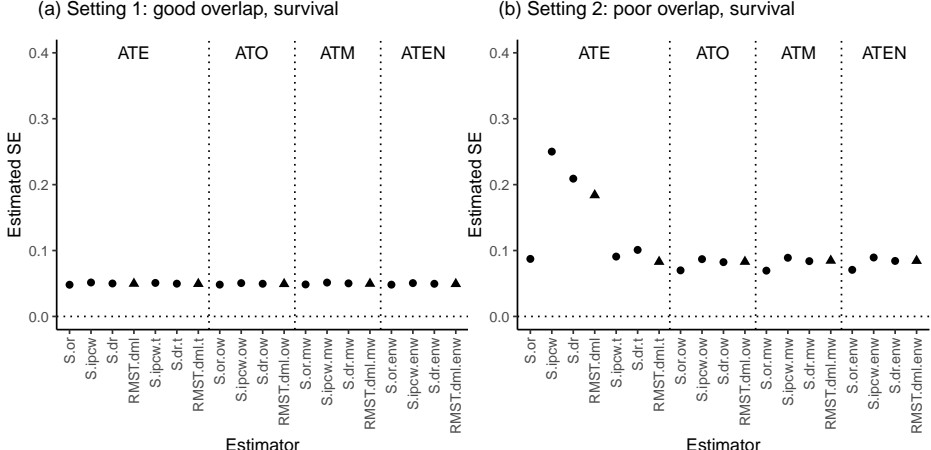

Figure 4: Average standard errors of different estimators grouped by estimand at $\tau = 4$. Solid circles represent average bootstrap standard errors of the comparator estimators; and solid triangles represent average estimated standard errors of the proposed estimators. Other symbols and notations are similar to Figure 2.

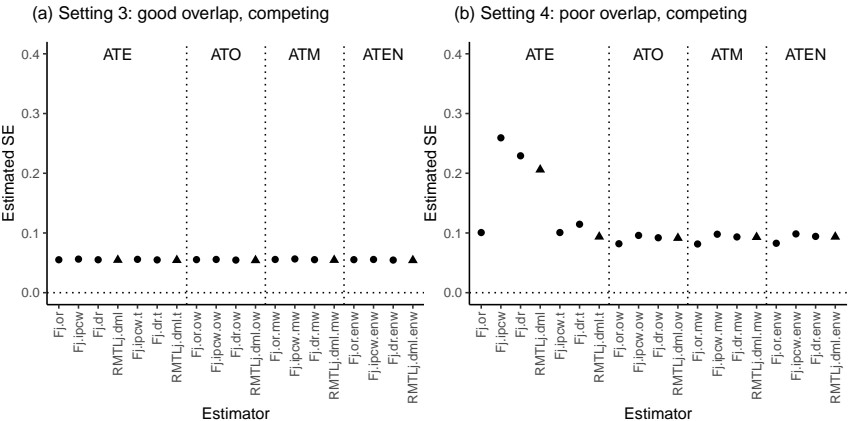

Figure 5: Average standard errors of different estimators grouped by estimand at $\tau = 4$. Solid circles represent average bootstrap standard errors of the comparator estimators; and solid triangles represent average estimated standard errors of the proposed estimators. Other symbols and notations are similar to Figure 6.

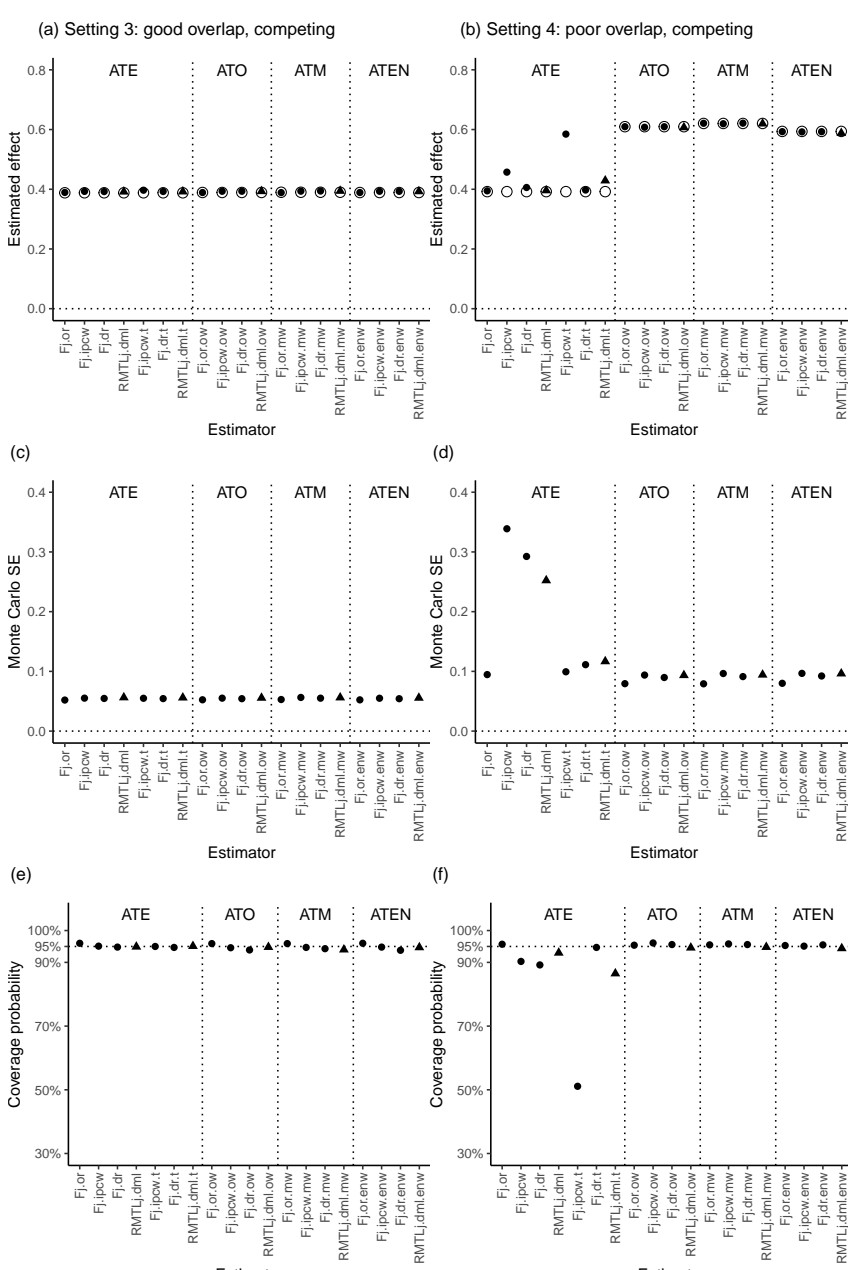

Figure 6: Estimated cumulative treatment effect, Monte Carlo standard errors, and coverage probabilities from different estimators grouped by estimand at $\tau = 4$. Estimands in each plot are separated by vertical dotted lines. The order of the estimands from left to right in each plot is: ATE, ATO, ATM, ATEN. Hollow circles in (a) and (b) represent true effects; solid circles represent the comparator estimators; and solid triangles represent the proposed estimators. Abbreviations of caption can be found in Table 3.

## B.5 CANDIDATE LEARNERS

To speed up training, we use two-fold cross-validation throughout this paper and select relatively fast learners among all possible candidate learners. For the propensity score, we employed marginal mean (SL.mean), feed-forward neural networks (SL.nnet), kernel k-nearest neighbors (SL.kernelKnn), classification trees (SL.rpartPrune), extreme gradient boosting (SL.xgboost), random forests (SL.ranger), logistic regression (SL.glm), forward/backwards stepwise logistic regression (SL.step), generalized additive model (SL.gam), regularized logistic regression (SL.glmnet), and multivariate adaptive regression splines (SL.earth). In terms of

conditional failure-time distributions, we considered the Kaplan-Meier estimator (survSL.km), piecewise constant hazard regression (survSL.pchreg), survival random forest (survSL.rfsrc), Cox proportional hazards regression (survSL.coxph), and regression assuming event and censoring times follow conditional exponential (survSL.expreg), Weibull (survSL.weibreg), and log-logistic (survSL.loglogreg) distributions.

## C  APPLICATION

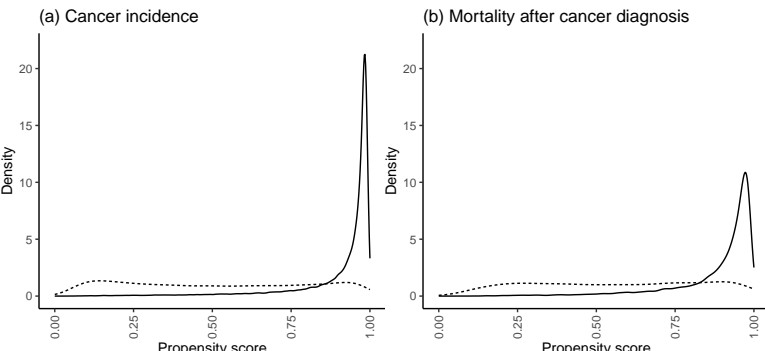

Figure 7: Density of estimated propensity score; dotted: sulphonylurea; solid: metformin; (a) cancer incidence: $n$ (sulphonylurea)$=$ $13,864$ and $n$ (metformin)$=$ $79,489$; (b) mortality after cancer diagnosis: $n$ (sulphonylurea)$= 1,636$ and $n$ (metformin)$= 8,172$.

### C.1  COVARIATE SPECIFICATIONS

The pretreatment confounders $X$ include gender, prescription age, calendar year of treatment, body mass index (BMI), Hemoglobin A1C (HbA1c), heart failure, coronary heart disease, atrial fibrillation, stroke, hypertension, peripheral vascular disease, chronic kidney disease, chronic obstructive pulmonary disease, smoking, index for multiple deprivation (IMD), region, and time since diabetes diagnosis. These confounders were selected by epidemiologists and physicians based on subject matter knowledge and are expected to be common causes of treatment decision, event process, and censoring distribution.

### C.2  DEALING WITH NON-POSITIVE SEMIDEFINITE COVARIANCE ESTIMATOR

When the sample size is large, the observed survival times can be close at earlier survival times. Under such circumstances, the covariance estimator may fail to be positive semidefinite (PSD) such that sample paths can't be generated directly. One may alternatively consider generating sample paths by perturbing centered IFs with a standard normal random variable $\xi$ as

$$\xi \frac{\widehat{\varphi}^{\mathrm{RMST},h}(t,a;\widehat{\eta}^{\Lambda,h})}{\widehat{\sigma}^{\mathrm{RMST},h,+}(t,a)} \sim \mathcal{N}\left(0, \frac{\widehat{\varphi}^{\mathrm{RMST},h}(t,a;\widehat{\eta}^{\Lambda,h})\{\widehat{\varphi}^{\mathrm{RMST},h}(t,a;\widehat{\eta}^{\Lambda,h})\}^T}{\{\widehat{\sigma}^{\mathrm{RMST},h,+}(t,a)\}^2}\right)$$

such that the PSD requirement for the cross-fitted covariance function can be avoided. Aside from perturbing the centered IFs, Lin et al. (1993) proposed to approximate the limiting distribution of certain martingale integrals by perturbing corresponding counting processes, i.e. $\mathrm{d}M(t \mid A, X) = \xi I(\widetilde{T} = t, \Delta)$, $\mathrm{d}M^C(t \mid A, X) = \xi I(\widetilde{T} = t, \Delta = 0)$, and $\mathrm{d}M_j(t \mid A, X) = \xi I(\widetilde{T} = t, \widetilde{J} = j)$. Besides perturbation methods, the non-PSD estimated covariance may be converted to its nearest PSD matrix (Higham, 1988), but the asymptotic equivalence of this transformation has not been proved yet. The inference of causal contrasts between treatment-specific RMSTs and cause-specific RMTLs can be established similarly by replacing corresponding shape-corrected estimates.

### C.3  RESULTS

As piecewise constant hazard regression (survSL.pchreg) is too time-consuming for large data sets, we remove it from the simulation candidate library for all conditional time-to-event distributions. We estimate the propensity scores using the simulation candidate library without kernel k-nearest neighbors (SL.kernelKnn) and generalized additive model (SL.gam).

Table 4: Results (in years) of estimated counterfactual RMTL/RMST and cumulative treatment effect between metformin initiators and sulphonylurea initiators on the CPRD data at $\tau = 10$ years. SE: standard error; CI: confidence interval; CB: confidence band.

|  | Point estimates (SE) | 95% pointwise CI | 95% uniform CB |
|---|---|---|---|
| Cancer incidence ATE |  |  |  |
| Sulphonylurea (RMTL) | 0.914 (0.050) | (0.815, 1.013) | (0.764, 1.063) |
| Metformin (RMTL) | 0.765 (0.009) | (0.747, 0.783) | (0.743, 0.788) |
| Metformin-Sulphonylurea | -0.149 (0.051) | (-0.249, -0.048) | (-0.304, 0.007) |
|  |  |  |  |
| Cancer incidence ATO |  |  |  |
| Sulphonylurea (RMTL) | 0.885 (0.025) | (0.837, 0.934) | (0.819, 0.952) |
| Metformin (RMTL) | 0.822 (0.015) | (0.792, 0.852) | (0.777, 0.867) |
| Metformin-Sulphonylurea (difference in RMTL) | -0.064 (0.029) | (-0.120, -0.008) | (-0.133, 0.005) |
|  |  |  |  |
| Mortality after cancer ATE |  |  |  |
| Sulphonylurea (RMST) | 6.292 (0.208) | (5.885, 6.699) | (5.709, 6.875) |
| Metformin (RMST) | 8.163 (0.039) | (8.087, 8.240) | (8.069, 8.258) |
| Metformin-Sulphonylurea difference in (RMST) | 1.871 (0.211) | (1.457, 2.285) | (1.274, 2.469) |
|  |  |  |  |
| Mortality after cancer diagnosis ATO |  |  |  |
| Sulphonylurea (RMST) | 6.540 (0.124) | (6.297, 6.783) | (6.186, 6.893) |
| Metformin (RMST) | 7.699 (0.081) | (7.541, 7.858) | (7.474, 7.925) |
| Metformin-Sulphonylurea (difference in RMST) | 1.160 (0.123) | (0.919, 1.401) | (0.805, 1.515) |

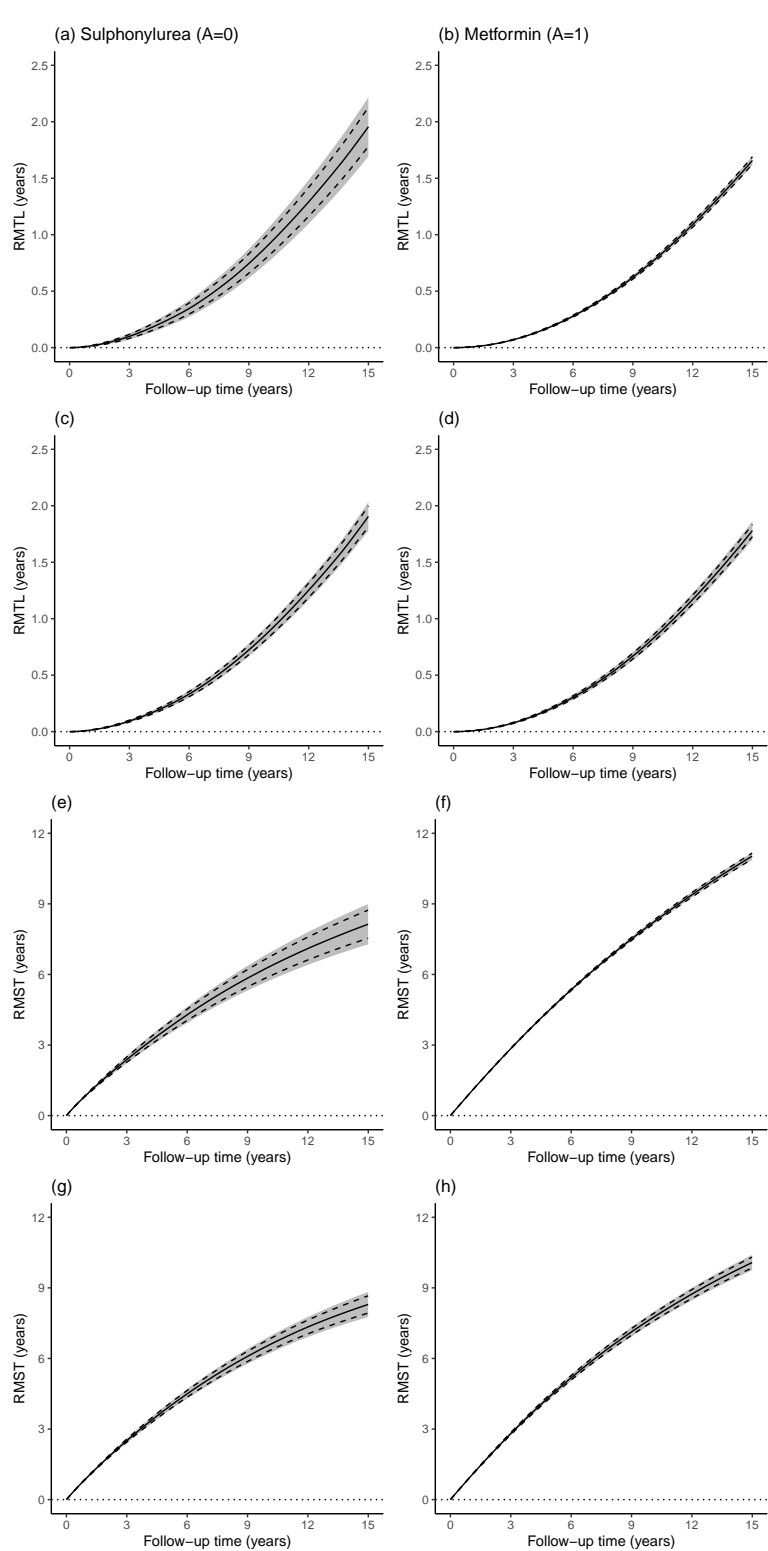

Figure 8: Curves of estimated counterfactual RMTL/RMST on the CPRD data; first column (a), (c), (e), and (g): sulphonylurea initiators; second column (b), (d), (f), and (h): metformin initiators; first row (a) and (b): IPTW treatment-specific cause-specific RMTL of cancer incidence; second row (c) and (d): OW treatment-specific cause-specific RMTL of cancer incidence; (e) and (f): IPTW treatment-specific RMST of mortality after cancer diagnosis; (g) and (h) OW treatment-specific RMST of mortality after cancer diagnosis. In all figures, dotted line: $y = 0$; solid lines: point estimates; dashed lines: 95% pointwise confidence intervals; shaded area: 95% uniform simultaneous confidence bands.

## D  THEORY

### D.1  ASYMPTOTIC INFERENCE

Building on the asymptotic linearity established in Theorem 3, we now provide the detailed procedures for constructing pointwise confidence intervals and simultaneous confidence bands. The pointwise confidence intervals (CIs) with significance level $\alpha$ are defined as

$$P_0\left(\widehat{l}^{\text{RMST},h}(\tau,a) \leq \widehat{\psi}^{\text{RMST},h,lcm}(\tau,a) \leq \widehat{u}^{\text{RMST},h}(\tau,a)\right) = 1-\alpha$$

Here, $\widehat{l}^{\text{RMST},h}(\tau,a)$ and $\widehat{u}^{\text{RMST},h}(\tau,a)$ are two stochastic processes, and $z_q$ is the $q$-quantile of the standard normal distribution. The Wald type symmetric CIs can be constructed as $\widehat{\psi}^{\text{RMST},h,lcm}(\tau,a) \pm z_{1-\alpha/2}\widehat{\sigma}^{\text{RMST},h,+}(\tau,a)/\sqrt{n}$. Unlike the confidence intervals which only describe the estimated curves at a single time point, simultaneous confidence bands encapsulate the entire estimated curves over a user-specified follow-up time, which can be used to evaluate equivalence or noninferiority claims in comparative effectiveness studies (Zhao et al., 2016).

Under Condition 4 and 11, simultaneous confidence bands can be constructed at $(1-\alpha)$ coverage level on the interval $[\tau_l, \tau_u]$ such that

$$P_0\left(\widehat{l}^{\text{RMST},h}(\tau,a) \leq \widehat{\psi}^{\text{RMST},h,lcm}(\tau,a) \leq \widehat{u}^{\text{RMST},h}(\tau,a) \quad \forall \tau \in [\tau_l, \tau_u]\right) \geq 1-\alpha$$

We refer to Chen et al. (2021) for the development of various types of simultaneous confidence bands for survival functions. Here, we consider an increasing-width confidence band $\widehat{\psi}^{\text{RMST},h,lcm}(\tau,a) \pm \widehat{c}_\alpha\widehat{\sigma}^{\text{RMST},h,+}(\tau,a)/\sqrt{n}$, where $\widehat{c}_\alpha$ is the $(1-\alpha)$-quantile of the supremum of the absolute value of sample paths of a zero mean Gaussian process over $[\tau_l, \tau_u]$ with cross-fitted covariance function

$$\widehat{\sigma}^{\text{RMST},h}(u,t,a) = \frac{P_n\{\widehat{\varphi}^{\text{RMST},h}(u,a;\widehat{\eta}_k^{\Lambda,h})\widehat{\varphi}^{\text{RMST},h}(t,a;\widehat{\eta}_k^{\Lambda,h})\}}{\widehat{\sigma}^{\text{RMST},h,+}(u,a)\widehat{\sigma}^{\text{RMST},h,+}(t,a)}$$

In practice, we choose $\tau_l \geq \text{argmin}_\tau\{\widehat{\sigma}^{\text{RMST},h,+}(\tau,a) > 0\}$ to avoid zero denominators in the covariance estimator and $\tau_u \leq \tau_{\max}$. We discuss how to address non-positive semidefinite covariance estimators in Appendix C.

### D.2  THEOREM 1

The unweighted nonparametric uncentered influence function is

$$\begin{aligned}
\phi_0^{\text{RMST}}(\tau,a;\eta_0^\Lambda) =\ & \frac{I(A=a)\Delta(\tau)\min(\widetilde{T},\tau)}{\pi_0(a\mid X)G_0(\tau\wedge\widetilde{T}-\mid a,X)} + \left\{1 - \frac{I(A=a)}{\pi_0(a\mid X)}\right\}\text{RMST}_0(\tau\mid a,X) \\
& + \frac{I(A=a)}{\pi_0(a\mid X)}\int_0^{\tau\wedge\widetilde{T}} \frac{t\mathrm{d}M_0^C(t\mid a,X)}{G_0(t-\mid a,X)} \\
& + \frac{I(A=a)}{\pi_0(a\mid X)}\int_0^{\tau\wedge\widetilde{T}} \frac{\{\text{RMST}_0(\tau\mid a,X) - \text{RMST}_0(t\mid a,X)\}\mathrm{d}M_0^C(t\mid a,X)}{S_0(t\mid a,X)G_0(t-\mid a,X)}
\end{aligned} \tag{5}$$

The unweighted nonparametric uncentered IF for competing risks is

$$\begin{aligned}
\phi_{j,0}^{\text{RMTL}}(\tau,a;\eta_{j,0}^\Lambda) =\ & \frac{I(A=a)(\tau-\min(\widetilde{T},\tau))I(\widetilde{J}=j)}{\pi_0(a\mid X)G_0(\widetilde{T}-\mid a,X)} + \left\{1 - \frac{I(A=a)}{\pi_0(a\mid X)}\right\}\text{RMTL}_{j,0}(\tau\mid a,X) \\
& - \frac{I(A=a)}{\pi_0(a\mid X)}\int_0^{\tau\wedge\widetilde{T}} \frac{(\tau-t)F_{j,0}(t\mid a,X)\mathrm{d}M_0^C(t\mid a,X)}{S_0(t\mid a,X)G_0(t-\mid a,X)} \\
& + \frac{I(A=a)}{\pi_0(a\mid X)}\int_0^{\tau\wedge\widetilde{T}} \frac{\{\text{RMTL}_{j,0}(\tau\mid a,X) - \text{RMTL}_{j,0}(t\mid a,X)\}\mathrm{d}M_0^C(t\mid a,X)}{S_0(t\mid a,X)G_0(t-\mid a,X)}
\end{aligned} \tag{6}$$

Proof of Theorem 1: following Robins & Rotnitzky (1992); Tsiatis (2006), without additional assumptions on the time-to-event distributions, $\psi_0^{\text{RMST},h}(\tau,a)$ can be formed as augmented inverse probability weighted

complete case (AIPWCC) estimators with estimating function

$$
\begin{aligned}
m^{\text{AIPWCC}}\{\psi_0^{\text{RMST},h}(\tau,a);\eta_0^{\Lambda,h}\} = & \frac{w_0^h(a,X)\Delta}{G_0(\widetilde{T}-\mid a,X)}\{\min(\widetilde{T},\tau) - \psi_0^{\text{RMST},h}(\tau,a)\} + \{h_0(X) - w_0^h(a,X)\} \\
& \times E_0\{\min(T,\tau) - \psi_0^{\text{RMST},h}(\tau,a)\mid A=a,X\} + w_0^h(a,X)\int_0^\infty \frac{\mathrm{d}M_0^C(t\mid a,X)}{G_0(t-\mid a,X)} \\
& \times E_0\{\min(T,\tau) - \psi_0^{\text{RMST},h}(\tau,a)\mid A=a,X,T>t\}
\end{aligned}
$$

Note that

$$
\begin{aligned}
& \int_0^\infty \frac{\mathrm{d}M_0^C(t\mid a,X)}{G_0(t-\mid a,X)} E_0\{\min(T,\tau) - \psi_0^{\text{RMST},h}(\tau,a)\mid A=a,X,T>t\} \\
= & \int_0^\infty \frac{\mathrm{d}M_0^C(t\mid a,X)}{G_0(t-\mid a,X)}[(t\le\tau)[E_0\{\min(T,\tau)\mid A=a,X,T>t\} - \psi_0^{\text{RMST},h}(\tau,a) + (t>\tau)\{\tau - \psi_0^{\text{RMST},h}(\tau,a)\}]] \\
= & \int_0^{\tau\wedge\widetilde{T}} \frac{\mathrm{d}M_0^C(t\mid a,X)}{G_0(t-\mid a,X)} E_0\{\min(T,\tau)\mid A=a,X,T>t\} + \tau\int_{\tau+}^\infty \frac{\mathrm{d}M_0^C(t\mid a,X)}{G_0(t-\mid a,X)} \\
& - \psi_0^{\text{RMST},h}(\tau,a)\int_0^\infty \frac{\mathrm{d}M_0^C(t\mid a,X)}{G_0(t-\mid a,X)}
\end{aligned}
$$

where

$$
\begin{aligned}
& E_0\{\min(T,\tau)\mid A=a,X,T>t\} = \int_t^\infty \min(t,\tau)\mathrm{d}P_0(T\le u\mid T>t,A=a,X) \\
= & \int_t^\infty \min(t,\tau)\mathrm{d}\left\{\frac{P_0(T>t,A=a,X,T\le u)}{P_0(T>t,A=a,X)}\right\} = \int_t^\infty \min(t,\tau)\mathrm{d}\left\{\frac{P_0(t<T\le u\mid A=a,X)}{P_0(T>t\mid A=a,X)}\right\} \\
= & \int_t^\infty \min(t,\tau)\mathrm{d}\left\{\frac{S_0(t\mid a,X) - S_0(u\mid a,X)}{S_0(t\mid a,X)}\right\} = \frac{\int_t^\infty \min(t,\tau)\{-\mathrm{d}S_0(u\mid a,X)\}}{S_0(t\mid a,X)} \\
= & \frac{-\int_t^\tau u\mathrm{d}S_0(u\mid a,X) + \tau S_0(\tau\mid a,X)}{S_0(t\mid a,X)} = t + \frac{\int_t^\tau S_0(u\mid a,X)\mathrm{d}t}{S_0(t\mid a,X)} \\
= & t + \frac{\text{RMST}_0(\tau\mid a,X) - \text{RMST}_0(t\mid a,X)}{S_0(t\mid a,X)}
\end{aligned}
$$

Note that

$$
\begin{aligned}
& \frac{\Delta}{G_0(\widetilde{T}-\mid a,X)} + \frac{I(\widetilde{T}>\tau)}{G_0(\tau\mid a,X)} - \frac{\Delta I(\widetilde{T}>\tau)}{G_0(\widetilde{T}-\mid a,X)} = \frac{\Delta I(\widetilde{T}\le\tau)}{G_0(\tau\wedge\widetilde{T}-\mid a,X)} + \frac{I(\widetilde{T}>\tau)}{G_0(\tau\wedge\widetilde{T}-\mid a,X)} \\
= & \frac{I(C>\tau\wedge T-)}{G_0(\tau\wedge\widetilde{T}-\mid a,X)} = \frac{\Delta(\tau)}{G_0(\tau\wedge\widetilde{T}-\mid a,X)}
\end{aligned}
$$

Therefore,

$$
m^{\text{AIPWCC}}\{\psi_0^{\text{RMST},h}(\tau,a);\eta_0^{\Lambda,h}\}
$$

$$
=\frac{w_0^h(a,X)\Delta\min(\widetilde{T},\tau)}{G_0(\widetilde{T}-\mid a,X)}-\frac{w_0^h(a,X)\Delta\psi_0^{\text{RMST},h}(\tau,a)}{G_0(\widetilde{T}-\mid a,X)}+\{h_0(X)-w_0^h(a,X)\}\text{RMST}_0(\tau\mid a,X)
$$

$$
-\{h_0(X)-w_0^h(a,X)\}\psi_0^{\text{RMST},h}(\tau,a)+w_0^h(a,X)\int_0^{\tau\wedge\widetilde{T}}\frac{\mathrm{d}M_0^C(t\mid a,X)}{G_0(t-\mid a,X)}
$$

$$
\times\left\{t+\frac{\text{RMST}_0(\tau\mid a,X)-\text{RMST}_0(t\mid a,X)}{S_0(t\mid a,X)}\right\}+\frac{w_0^h(a,X)\tau I(\widetilde{T}>\tau)}{G_0(\tau\mid a,X)}
$$

$$
-\frac{w_0^h(a,X)\tau\Delta I(\widetilde{T}>\tau)}{G_0(\widetilde{T}-\mid a,X)}-w_0^h(a,X)\psi_0^{\text{RMST},h}(\tau,a)+\frac{w_0^h(a,X)\Delta\psi_0^{\text{RMST},h}(\tau,a)}{G_0(\widetilde{T}-\mid a,X)}
$$

$$
=\frac{w_0^h(a,X)\Delta\min(\widetilde{T},\tau)}{G_0(\widetilde{T}-\mid a,X)}+\frac{w_0^h(a,X)I(\widetilde{T}>\tau)\min(\widetilde{T},\tau)}{G_0(\tau\mid a,X)}-\frac{w_0^h(a,X)\Delta I(\widetilde{T}>\tau)\min(\widetilde{T},\tau)}{G_0(\widetilde{T}-\mid a,X)}
$$

$$
+\{h_0(X)-w_0^h(a,X)\}\text{RMST}_0(\tau\mid a,X)+w_0^h(a,X)
$$

$$
\times\int_0^{\tau\wedge\widetilde{T}}\frac{\mathrm{d}M_0^C(t\mid a,X)}{G_0(t-\mid a,X)}\left\{t+\frac{\text{RMST}_0(\tau\mid a,X)-\text{RMST}_0(t\mid a,X)}{S_0(t\mid a,X)}\right\}-h_0(X)\psi_0^{\text{RMST},h}(\tau,a)
$$

$$
=\frac{w_0^h(a,X)\Delta(\tau)\min(\widetilde{T},\tau)}{G_0(\tau\wedge\widetilde{T}-\mid a,X)}+\{h_0(X)-w_0^h(a,X)\}\text{RMST}_0(\tau\mid a,X)+w_0^h(a,X)
$$

$$
\times\int_0^{\tau\wedge\widetilde{T}}\frac{\mathrm{d}M_0^C(t\mid a,X)}{G_0(t-\mid a,X)}\left\{t+\frac{\text{RMST}_0(\tau\mid a,X)-\text{RMST}_0(t\mid a,X)}{S_0(t\mid a,X)}\right\}-h_0(X)\psi_0^{\text{RMST},h}(\tau,a)
$$

The influence function of the fully augmented M-estimating equation is

$$
\varphi_0^{\text{RMST},h}(\tau,a;\eta_0^{\Lambda,h})=-\left\{E_0\left[\frac{\partial m^{\text{AIPWCC}}\{\psi_0^{\text{RMST},h}(\tau,a);\eta_0^{\Lambda,h}\}}{\partial\psi_0^{\text{RMST},h}(\tau,a)\}}\right]\right\}^{-1}m^{\text{AIPWCC}}\{\psi_0^{\text{RMST},h}(\tau,a);\eta_0^{\Lambda,h}\}
$$

$$
=[E_0\{h_0(X)\}]^{-1}\left[w_0^h(a,X)\left\{\frac{\Delta(\tau)\min(\widetilde{T},\tau)}{G_0(\tau\wedge\widetilde{T}-\mid a,X)}-\text{RMST}_0(\tau\mid a,X)\right.\right.
$$

$$
\left.+\int_0^{\tau\wedge\widetilde{T}}\frac{t\mathrm{d}M_0^C(t\mid a,X)}{G_0(t-\mid a,X)}+\int_0^{\tau\wedge\widetilde{T}}\frac{\{\text{RMST}_0(\tau\mid a,X)-\text{RMST}_0(t\mid a,X)\}\mathrm{d}M_0^C(t\mid a,X)}{S_0(t\mid a,X)G_0(t-\mid a,X)}\right\}
$$

$$
\left.+h_0(X)\text{RMST}_0(\tau\mid a,X)-h_0(X)\psi_0^{\text{RMST},h}(\tau,a)\right]
$$

In the competing risks setting, the doubly robust estimating function for $\psi_{j,0}^{\text{RMTL},h}(\tau,a)$ is

$$
m^{\text{AIPWCC}}\{\psi_{j,0}^{\text{RMTL},h}(\tau,a);\eta_{j,0}^{\Lambda,h}\}
$$

$$
=\frac{w_0^h(a,X)}{G_0(\widetilde{T}-\mid a,X)}[\{\tau-\min(\widetilde{T},\tau)\}I(\widetilde{J}=j)-\Delta\psi_{j,0}^{\text{RMTL},h}(\tau,a)]+\{h_0(X)-w_0^h(a,X)\}
$$

$$
\times E_0[\{\tau-\min(T,\tau)\}I(J=j)-\psi_{j,0}^{\text{RMTL},h}(\tau,a)\mid A=a,X]+w_0^h(a,X)\int_0^\infty\frac{\mathrm{d}M_0^C(t\mid a,X)}{G_0(t-\mid a,X)}
$$

$$
\times E_0[\{\tau-\min(T,\tau)\}I(J=j)-\psi_{j,0}^{\text{RMTL},h}(\tau,a)\mid A=a,X,T>t]
$$

where

$$E_0[\{\tau - \min(T,\tau)\}(J=j) \mid A=a, X, T>t]$$

$$= \int_t^\infty \{\tau - \min(u,\tau)\}\mathrm{d}P_0(T \le u, J=j \mid A=a, X, T>t)$$

$$= \int_t^\infty \{\tau - \min(u,\tau)\}\mathrm{d}\left\{\frac{P_0(t < T \le u, J=j, A=a, X)}{P_0(A=a, X, T>t)}\right\}$$

$$= \int_t^\infty \{\tau - \min(u,\tau)\}\mathrm{d}\left\{\frac{P_0(t < T \le u, J=j \mid A=a, X)}{P_0(T>t \mid A=a, X)}\right\}$$

$$= \int_t^\infty \{\tau - \min(u,\tau)\}\mathrm{d}\left\{\frac{F_{j,0}(u \mid a, X) - F_{j,0}(t \mid a, X)}{S_0(t \mid a, X)}\right\} = \frac{\int_t^\infty \{\tau - \min(u,\tau)\}\mathrm{d}F_{j,0}(u \mid a, X)}{S_0(t \mid a, X)}$$

$$= \frac{\int_t^\tau (\tau - u)\mathrm{d}F_{j,0}(u \mid a, X)}{S_0(t \mid a, X)} = \frac{(t-\tau)F_{j,0}(t \mid a, X) + \mathrm{RMTL}_{j,0}(\tau \mid a, X) - \mathrm{RMTL}_{j,0}(t \mid a, X)}{S_0(t \mid a, X)}$$

Note that

$$\int_0^\infty \frac{\mathrm{d}M_0^C(t \mid a, X)}{G_0(t- \mid a, X)} E_0[\{\tau - \min(T,\tau)\}I(J=j) - \psi_{j,0}^{\mathrm{RMTL},h}(\tau, a) \mid A=a, X, T>t]$$

$$= \int_0^\infty \frac{\mathrm{d}M_0^C(t \mid a, X)}{G_0(t- \mid a, X)}[(t \le \tau)(E_0[\{\tau - \min(T,\tau)\}I(J=j) \mid A=a, X, T>t] - \psi_{j,0}^{\mathrm{RMTL},h}(\tau, a))$$

$$+ (t > \tau)\{-\psi_{j,0}^{\mathrm{RMTL},h}(\tau, a)\}]$$

$$= \int_0^{\tau \wedge \widetilde{T}} \frac{\{(t-\tau)F_{j,0}(t \mid a, X) + \mathrm{RMTL}_{j,0}(\tau \mid a, X) - \mathrm{RMTL}_{j,0}(t \mid a, X)\}\mathrm{d}M_0^C(t \mid a, X)}{S_0(t \mid a, X)G_0(t- \mid a, X)}$$

$$- \psi_{j,0}^{\mathrm{RMTL},h}(\tau, a) + \frac{\Delta\psi_{j,0}^{\mathrm{RMTL},h}(\tau, a)}{G_0(\widetilde{T}- \mid a, X)}$$

Consequently, the estimating function becomes

$$m^{\mathrm{AIPWCC}}\{\psi_{j,0}^{\mathrm{RMTL},h}(\tau, a); \eta_{j,0}^{\Lambda,h}\}$$

$$= \frac{w_0^h(a, X)\{\tau - \min(\widetilde{T},\tau)\}I(\widetilde{J}=j)}{G_0(\widetilde{T}- \mid a, X)} - \frac{w_0^h(a, X)\Delta\psi_{j,0}^{\mathrm{RMTL},h}(\tau, a)}{G_0(\widetilde{T}- \mid a, X)} + \{h_0(X) - w_0^h(a, X)\}$$

$$\times \mathrm{RMTL}_{j,0}(\tau \mid a, X) - \{h_0(X) - w_0^h(a, X)\}\psi_{j,0}^{\mathrm{RMTL},h}(\tau, a) + w_0^h(a, X)$$

$$\times \int_0^{\tau \wedge \widetilde{T}} \frac{\{(t-\tau)F_{j,0}(t \mid a, X) + \mathrm{RMTL}_{j,0}(\tau \mid a, X) - \mathrm{RMTL}_{j,0}(t \mid a, X)\}\mathrm{d}M_0^C(t \mid a, X)}{S_0(t \mid a, X)G_0(t- \mid a, X)}$$

$$- w_0^h(a, X)\psi_{j,0}^{\mathrm{RMTL},h}(\tau, a) + \frac{w_0^h(a, X)\Delta\psi_{j,0}^{\mathrm{RMTL},h}(\tau, a)}{G_0(\widetilde{T}- \mid a, X)}$$

$$= \frac{w_0^h(a, X)\{\tau - \min(\widetilde{T},\tau)\}I(\widetilde{J}=j)}{G_0(\widetilde{T}- \mid a, X)} + \{h_0(X) - w_0^h(a, X)\}\mathrm{RMTL}_{j,0}(\tau \mid a, X) + w_0^h(a, X)$$

$$\times \int_0^{\tau \wedge \widetilde{T}} \frac{\{(t-\tau)F_{j,0}(t \mid a, X) + \mathrm{RMTL}_{j,0}(\tau \mid a, X) - \mathrm{RMTL}_{j,0}(t \mid a, X)\}\mathrm{d}M_0^C(t \mid a, X)}{S_0(t \mid a, X)G_0(t- \mid a, X)}$$

$$- h_0(X)\psi_{j,0}^{\mathrm{RMTL},h}(\tau, a)$$

The centered IF is

$$\varphi_{j,0}^{\mathrm{RMTL},h}(\tau, a; \eta_{j,0}^{\Lambda,h})$$

$$= -\left\{E_0\left[\frac{\partial m^{\mathrm{AIPWCC}}\{\psi_{j,0}^{\mathrm{RMTL},h}(\tau, a); \eta_{j,0}^{\Lambda,h}\}}{\partial\psi_{j,0}^{\mathrm{RMTL},h}(\tau, a)}\right]\right\}^{-1} \partial m^{\mathrm{AIPWCC}}\{\psi_{j,0}^{\mathrm{RMTL},h}(\tau, a); \eta_{j,0}^{\Lambda,h}\}$$

$$= [E_0\{h_0(X)\}]^{-1}\left(w_0^h(a, X)\left[\frac{\{\tau - \min(\widetilde{T},\tau)\}I(\widetilde{J}=j)}{G_0(\widetilde{T}- \mid a, X)} - \mathrm{RMTL}_{j,0}(\tau \mid a, X)\right.\right.$$

$$\left.+ \int_0^{\tau \wedge \widetilde{T}} \frac{\{(t-\tau)F_{j,0}(t \mid a, X) + \mathrm{RMTL}_{j,0}(\tau \mid a, X) - \mathrm{RMTL}_{j,0}(t \mid a, X)\}\mathrm{d}M_0^C(t \mid a, X)}{S_0(t \mid a, X)G_0(t- \mid a, X)}\right]$$

$$\left.+ h_0(X)\mathrm{RMTL}_{j,0}(\tau \mid a, X) - h_0(X)\psi_{j,0}^{\mathrm{RMTL},h}(\tau, a)\right) \quad \blacksquare$$

## D.3 LEMMA 1

**Lemma 1**

$$
\phi_0^{\text{RMST}}(\tau, a; \eta_0^\Lambda) = \text{RMST}_0(\tau \mid a, X) - \frac{I(A = a)\text{RMST}_0(\tau \mid a, X)}{\pi_0(a \mid X)} \int_0^{\tau \wedge \widetilde{T}} \frac{\mathrm{d}M_0(t \mid a, X)}{S_0(t \mid a, X)G_0(t- \mid a, X)}
$$
$$
+ \frac{I(A = a)}{\pi_0(a \mid X)} \int_0^{\tau \wedge \widetilde{T}} \frac{\text{RMST}_0(t \mid a, X)\mathrm{d}M_0(t \mid a, X)}{S_0(t \mid a, X)G_0(t- \mid a, X)}
$$

*and*

$$
\phi_{j,0}^{\text{RMTL}}(\tau, a; \eta_{j,0}^\Lambda) = \text{RMTL}_{j,0}(\tau \mid a, X) + \frac{I(A = a)}{\pi_0(a \mid X)} \int_0^{\tau \wedge \widetilde{T}} \frac{(\tau - t)\mathrm{d}M_{j,0}(t \mid a, X)}{G_0(t- \mid a, X)}
$$
$$
- \frac{I(A = a)}{\pi_0(a \mid X)} \int_0^{\tau \wedge \widetilde{T}} \frac{\{\text{RMTL}_{j,0}(\tau \mid a, X) - \text{RMTL}_{j,0}(t \mid a, X)\}\mathrm{d}M_0(t \mid a, X)}{S_0(t \mid a, X)G_0(t- \mid a, X)}
$$
$$
+ \frac{I(A = a)}{\pi_0(a \mid X)} \int_0^{\tau \wedge \widetilde{T}} \frac{(\tau - t)F_{j,0}(t \mid a, X)\mathrm{d}M_0(t \mid a, X)}{S_0(t \mid a, X)G_0(t- \mid a, X)}
$$

Proof of Lemma 1: we start from the RHS of the first equation above

$$
\text{RMST}_0(\tau \mid a, X) - \frac{I(A = a)\text{RMST}_0(\tau \mid a, X)}{\pi_0(a \mid X)} \int_0^{\tau \wedge \widetilde{T}} \frac{\mathrm{d}M_0(t \mid a, X)}{S_0(t \mid a, X)G_0(t- \mid a, X)}
$$
$$
+ \frac{I(A = a)}{\pi_0(a \mid X)} \int_0^{\tau \wedge \widetilde{T}} \frac{\mathrm{d}M_0(t \mid a, X)\text{RMST}_0(t \mid a, X)}{S_0(t \mid a, X)G_0(t- \mid a, X)} - h_0(x)\psi_0^{\text{RMST},h}(\tau, a)
$$
$$
= \text{RMST}_0(\tau \mid a, X) - \frac{I(A = a)\text{RMST}_0(\tau \mid a, X)}{\pi_0(a \mid X)} \int_0^\tau \frac{\mathrm{d}N(t)}{S_0(t \mid a, X)G_0(t- \mid a, X)}
$$
$$
+ \frac{I(A = a)\text{RMST}_0(\tau \mid a, X)}{\pi_0(a \mid X)} \int_0^{\tau \wedge \widetilde{T}} \frac{\mathrm{d}\Lambda_0(t \mid a, X)}{S_0(t \mid a, X)G_0(t- \mid a, X)} + \frac{I(A = a)}{\pi_0(a \mid X)}
$$
$$
\times \int_0^\tau \frac{\text{RMST}_0(t \mid a, X)\mathrm{d}N(t)}{S_0(t \mid a, X)G_0(t- \mid a, X)} - \frac{I(A = a)}{\pi_0(a \mid X)} \int_0^{\tau \wedge \widetilde{T}} \frac{\text{RMST}_0(t \mid a, X)\mathrm{d}\Lambda_0(t \mid a, X)}{S_0(t \mid a, X)G_0(t- \mid a, X)}
$$

where

$$
\int_0^\tau \frac{\mathrm{d}N(t)}{S_0(t \mid a, X)G_0(t- \mid a, X)} = \frac{\Delta I(\widetilde{T} \le \tau)}{S_0(\tau \wedge \widetilde{T} \mid a, X)G_0(\tau \wedge \widetilde{T}- \mid a, X)}
$$

and

$$
\int_0^\tau \frac{\text{RMST}_0(t \mid a, X)\mathrm{d}N(t)}{S_0(t \mid a, X)G_0(t- \mid a, X)} = \frac{\Delta I(\widetilde{T} \le \tau)\text{RMST}_0(\tau \wedge \widetilde{T} \mid a, X)}{S_0(\tau \wedge \widetilde{T} \mid a, X)G_0(\tau \wedge \widetilde{T}- \mid a, X)}
$$

Note that $\mathrm{d}\Lambda_0^C(u- \mid a, X) = P_0(u- \le T < (u + \mathrm{d}u)- \mid C \ge u-, A = a, X) = P_0(u \le T < u + \mathrm{d}u \mid C \ge u, A = a, X) = \mathrm{d}\Lambda_0^C(u \mid a, X)$, $\mathrm{d}\Lambda_0(u- \mid a, X) = \mathrm{d}\Lambda_0(u \mid a, X)$, and $\mathrm{d}\Lambda_{j,0}(u- \mid a, X) =$

$d\Lambda_{j,0}(u \mid a, X)$. Thus

$$\int_0^{\tau \wedge \widetilde{T}} \frac{d\Lambda_0(t \mid a, X)}{S_0(t \mid a, X)G_0(t- \mid a, X)} = \int_0^{\tau \wedge \widetilde{T}} \frac{1}{G_0(t- \mid a, X)} d\left\{\frac{1}{S_0(t \mid a, X)}\right\}$$

$$= \frac{1}{S_0(t \mid a, X)G_0(t- \mid a, X)}\Big|_0^{\tau \wedge \widetilde{T}} - \int_0^{\tau \wedge \widetilde{T}} \frac{1}{S_0(t \mid a, X)} d\left\{\frac{1}{G_0(t- \mid a, X)}\right\}$$

$$= \frac{1}{S_0(\tau \wedge \widetilde{T} \mid a, X)G_0(\tau \wedge \widetilde{T}- \mid a, X)} - 1 - \int_0^{\tau \wedge \widetilde{T}} \frac{d\Lambda_0^C(t- \mid a, X)}{S_0(t \mid a, X)G_0(t- \mid a, X)}$$

$$= \frac{1}{S_0(\tau \wedge \widetilde{T} \mid a, X)G_0(\tau \wedge \widetilde{T}- \mid a, X)} - 1 + \int_0^{\tau \wedge \widetilde{T}} \frac{dM_0^C(t \mid a, X)}{S_0(t \mid a, X)G_0(t- \mid a, X)} - \int_0^{\tau} \frac{dN^C(t \mid a, X)}{S_0(t \mid a, X)G_0(t- \mid a, X)}$$

$$= \frac{1}{S_0(\tau \wedge \widetilde{T} \mid a, X)G_0(\tau \wedge \widetilde{T}- \mid a, X)} - 1 + \int_0^{\tau \wedge \widetilde{T}} \frac{dM_0^C(t \mid a, X)}{S_0(t \mid a, X)G_0(t- \mid a, X)}$$

$$- \frac{(1-\Delta)I(\widetilde{T} \le \tau)}{S_0(\tau \wedge \widetilde{T}- \mid a, X)G_0(\tau \wedge \widetilde{T}- \mid a, X)}$$

$$= \frac{\Delta(\tau)}{S_0(\tau \wedge \widetilde{T} \mid a, X)G_0(\tau \wedge \widetilde{T}- \mid a, X)} - 1 + \int_0^{\tau \wedge \widetilde{T}} \frac{dM_0^C(t \mid a, X)}{S_0(t \mid a, X)G_0(t- \mid a, X)}$$

Since

$$d\left\{\frac{\text{RMST}_0(t \mid a, X)}{G_0(t- \mid a, X)}\right\} = \frac{S_0(t \mid a, X)G_0(t- \mid a, X)dt + G_0(t- \mid a, X)d\Lambda_0^C(t- \mid a, X)\text{RMST}_0(t \mid a, X)}{G_0^2(t- \mid a, X)}$$

$$= \frac{S_0(t \mid a, X)dt}{G_0(t- \mid a, X)} + \frac{\text{RMST}_0(t \mid a, X)d\Lambda_0^C(t \mid a, X)}{G_0(t- \mid a, X)}$$

and

$$\int_0^{\tau \wedge \widetilde{T}} \frac{\text{RMST}_0(t \mid a, X)d\Lambda_0(t \mid a, X)}{S_0(t \mid a, X)G_0(t- \mid a, X)} = \int_0^{\tau \wedge \widetilde{T}} \frac{\text{RMST}_0(t \mid a, X)}{G_0(t- \mid a, X)} d\left\{\frac{1}{S_0(t \mid a, X)}\right\}$$

$$= \frac{\text{RMST}_0(t \mid a, X)}{S_0(t \mid a, X)G_0(t- \mid a, X)}\Big|_0^{\tau \wedge \widetilde{T}} - \int_0^{\tau \wedge \widetilde{T}} \frac{1}{S_0(t \mid a, X)} d\left\{\frac{\text{RMST}_0(t \mid a, X)}{G_0(t- \mid a, X)}\right\}$$

$$= \frac{\text{RMST}_0(\tau \wedge \widetilde{T} \mid a, X)}{G_0(\tau \wedge \widetilde{T}- \mid a, X)S_0(\tau \wedge \widetilde{T} \mid a, X)} - \int_0^{\tau \wedge \widetilde{T}} \frac{dt}{G_0(t- \mid a, X)} - \int_0^{\tau \wedge \widetilde{T}} \frac{\text{RMST}_0(t \mid a, X)d\Lambda_0^C(t \mid a, X)}{S_0(t \mid a, X)G_0(t- \mid a, X)}$$

$$= \frac{\text{RMST}_0(\tau \wedge \widetilde{T} \mid a, X)}{G_0(\tau \wedge \widetilde{T}- \mid a, X)S_0(\tau \wedge \widetilde{T} \mid a, X)} - \frac{\tau \wedge \widetilde{T}}{G_0(\tau \wedge \widetilde{T}- \mid a, X)} + \int_0^{\tau \wedge \widetilde{T}} \frac{t d\Lambda_0^C(t \mid a, X)}{G_0(t- \mid a, X)}$$

$$+ \int_0^{\tau \wedge \widetilde{T}} \frac{\text{RMST}_0(t \mid a, X)dM_0^C(t \mid a, X)}{S_0(t \mid a, X)G_0(t- \mid a, X)} - \int_0^{\tau} \frac{\text{RMST}_0(t \mid a, X)dN^C(t)}{S_0(t \mid a, X)G_0(t- \mid a, X)}$$

$$= \frac{\text{RMST}_0(\tau \wedge \widetilde{T} \mid a, X)}{G_0(\tau \wedge \widetilde{T}- \mid a, X)S_0(\tau \wedge \widetilde{T} \mid a, X)} - \frac{\tau \wedge \widetilde{T}}{G_0(\tau \wedge \widetilde{T}- \mid a, X)} - \int_0^{\tau \wedge \widetilde{T}} \frac{t dM_0^C(t \mid a, X)}{G_0(t- \mid a, X)}$$

$$+ \int_0^{\tau} \frac{t dN^C(t \mid a, X)}{G_0(t- \mid a, X)} + \int_0^{\tau \wedge \widetilde{T}} \frac{\text{RMST}_0(t \mid a, X)dM_0^C(t \mid a, X)}{S_0(t \mid a, X)G_0(t- \mid a, X)} - \frac{\text{RMST}_0(\tau \wedge \widetilde{T} \mid a, X)I(\widetilde{T} \le \tau)(1-\Delta)}{G_0(\tau \wedge \widetilde{T}- \mid a, X)S_0(\tau \wedge \widetilde{T} \mid a, X)}$$

$$= \frac{\text{RMST}_0(\tau \wedge \widetilde{T} \mid a, X)}{G_0(\tau \wedge \widetilde{T}- \mid a, X)S_0(\tau \wedge \widetilde{T} \mid a, X)} - \frac{\tau \wedge \widetilde{T}}{G_0(\tau \wedge \widetilde{T}- \mid a, X)} - \int_0^{\tau \wedge \widetilde{T}} \frac{t dM_0^C(t \mid a, X)}{G_0(t- \mid a, X)}$$

$$+ \frac{(1-\Delta)I(\widetilde{T} \le \tau)(\tau \wedge \widetilde{T})}{G_0(\tau \wedge \widetilde{T}- \mid a, X)} + \int_0^{\tau \wedge \widetilde{T}} \frac{\text{RMST}_0(t \mid a, X)dM_0^C(t \mid a, X)}{S_0(t \mid a, X)G_0(t- \mid a, X)} - \frac{(1-\Delta)I(\widetilde{T} \le \tau)\text{RMST}_0(\tau \wedge \widetilde{T} \mid a, X)}{G_0(\tau \wedge \widetilde{T}- \mid a, X)S_0(\tau \wedge \widetilde{T} \mid a, X)}$$

$$= \frac{\Delta(\tau)\text{RMST}_0(\tau \wedge \widetilde{T} \mid a, X)}{G_0(\tau \wedge \widetilde{T}- \mid a, X)S_0(\tau \wedge \widetilde{T} \mid a, X)} - \frac{\Delta(\tau)\min(\widetilde{T}, \tau)}{G_0(\tau \wedge \widetilde{T}- \mid a, X)} - \int_0^{\tau \wedge \widetilde{T}} \frac{t dM_0^C(t \mid a, X)}{G_0(t- \mid a, X)}$$

$$+ \int_0^{\tau \wedge \widetilde{T}} \frac{\text{RMST}_0(t \mid a, X)dM_0^C(t \mid a, X)}{S_0(t \mid a, X)G_0(t- \mid a, X)}$$

Substitute previous equations into the RHS of the first equation in Lemma 1 gives,

$$\text{RMST}_0(\tau \mid a, X) - \frac{I(A = a)\Delta I(\widetilde{T} \leq \tau)\text{RMST}_0(\tau \mid a, X)}{\pi_0(a \mid X)S_0(\tau \wedge \widetilde{T} \mid a, X)G_0(\tau \wedge \widetilde{T}- \mid a, X)} + \frac{I(A = a)\text{RMST}_0(\tau \mid a, X)}{\pi_0(a \mid X)}$$

$$\times \left\{ \frac{\Delta(\tau)}{S_0(\tau \wedge \widetilde{T} \mid a, X)G_0(\tau \wedge \widetilde{T}- \mid a, X)} - 1 + \int_0^{\tau \wedge \widetilde{T}} \frac{\mathrm{d}M_0^C(t \mid a, X)}{S_0(t \mid a, X)G_0(t- \mid a, X)} \right\}$$

$$+ \frac{I(A = a)\Delta I(\widetilde{T} \leq \tau)\text{RMST}_0(\tau \wedge \widetilde{T} \mid a, X)}{\pi_0(a \mid X)S_0(\tau \wedge \widetilde{T} \mid a, X)G_0(\tau \wedge \widetilde{T}- \mid a, X)} - \frac{I(A = a)}{\pi_0(a \mid X)} \left\{ \frac{\Delta(\tau)\text{RMST}_0(\tau \wedge \widetilde{T} \mid a, X)}{S_0(\tau \wedge \widetilde{T} \mid a, X)G_0(\tau \wedge \widetilde{T}- \mid a, X)} \right.$$

$$\left. - \frac{\Delta(\tau)\min(\widetilde{T}, \tau)}{G_0(\tau \wedge \widetilde{T}- \mid a, X)} - \int_0^{\tau \wedge \widetilde{T}} \frac{t\mathrm{d}M_0^C(t \mid a, X)}{G_0(t- \mid a, X)} + \int_0^{\tau \wedge \widetilde{T}} \frac{\text{RMST}_0(t \mid a, X)\mathrm{d}M_0^C(t \mid a, X)}{S_0(t \mid a, X)G_0(t- \mid a, X)} \right\}$$

$$= \text{RMST}_0(\tau \mid a, X) - \frac{I(A = a)\Delta I(\widetilde{T} \leq \tau)\text{RMST}_0(\tau \mid a, X)}{\pi_0(a \mid X)S_0(\tau \wedge \widetilde{T} \mid a, X)G_0(\tau \wedge \widetilde{T}- \mid a, X)} + \frac{I(A = a)\Delta(\tau)\text{RMST}_0(\tau \mid a, X)}{\pi_0(a \mid X)S_0(\tau \wedge \widetilde{T} \mid a, X)G_0(\tau \wedge \widetilde{T}- \mid a, X)}$$

$$- \frac{I(A = a)\text{RMST}_0(\tau \mid a, X)}{\pi_0(a \mid X)} + \frac{I(A = a)\text{RMST}_0(\tau \mid a, X)}{\pi_0(a \mid X)} \int_0^{\tau \wedge \widetilde{T}} \frac{\mathrm{d}M_0^C(t \mid a, X)}{S_0(t \mid a, X)G_0(t- \mid a, X)}$$

$$+ \frac{I(A = a)\Delta I(\widetilde{T} \leq \tau)\text{RMST}_0(\tau \wedge \widetilde{T} \mid a, X)}{\pi_0(a \mid X)S_0(\tau \wedge \widetilde{T} \mid a, X)G_0(\tau \wedge \widetilde{T}- \mid a, X)} - \frac{I(A = a)\Delta(\tau)\text{RMST}_0(\tau \wedge \widetilde{T} \mid a, X)}{\pi_0(a \mid X)S_0(\tau \wedge \widetilde{T} \mid a, X)G_0(\tau \wedge \widetilde{T}- \mid a, X)}$$

$$+ \frac{I(A = a)\Delta(\tau)\min(\widetilde{T}, \tau)}{\pi_0(a \mid X)G_0(\tau \wedge \widetilde{T}- \mid a, X)} + \frac{I(A = a)}{\pi_0(a \mid X)} \int_0^{\tau \wedge \widetilde{T}} \frac{t\mathrm{d}M_0^C(t \mid a, X)}{G_0(t- \mid a, X)}$$

$$- \frac{I(A = a)}{\pi_0(a \mid X)} \int_0^{\tau \wedge \widetilde{T}} \frac{\text{RMST}_0(t \mid a, X)\mathrm{d}M_0^C(t \mid a, X)}{S_0(t \mid a, X)G_0(t- \mid a, X)}$$

$$= \frac{I(A = a)\Delta(\tau)\min(\widetilde{T}, \tau)}{\pi_0(a \mid X)G_0(\tau \wedge \widetilde{T}- \mid a, X)} + \left\{ 1 - \frac{I(A = a)}{\pi_0(a \mid X)} \right\}\text{RMST}_0(\tau \mid a, X) + \frac{I(A = a)}{\pi_0(a \mid X)}$$

$$\times \int_0^{\tau \wedge \widetilde{T}} \frac{\mathrm{d}M_0^C(t \mid a, X)}{G_0(t- \mid a, X)} \left\{ t + \frac{\text{RMST}_0(\tau \mid a, X) - \text{RMST}_0(t \mid a, X)}{S_0(t \mid a, X)} \right\}$$

$$= \phi_0^{\text{RMST}, h}(\tau, a; \eta_0^\Lambda)$$

Note that $\text{RMST}_0(\tau \wedge \widetilde{T} \mid a, X) \neq \text{RMST}_0(\tau \mid a, X)$ and

$$- \Delta I(\widetilde{T} \leq \tau)\text{RMST}_0(\tau \mid a, X) + \Delta(\tau)\text{RMST}_0(\tau \mid a, X) + \text{RMST}(\tau \wedge \widetilde{T} \mid a, X) - \Delta(\tau)\text{RMST}_0(\tau \wedge \widetilde{T} \mid a, X)$$

$$= - \Delta I(\widetilde{T} \leq \tau)\text{RMST}_0(\tau \mid a, X) + \Delta I(\widetilde{T} \leq \tau)\text{RMST}_0(\tau \mid a, X) + I(\widetilde{T} > \tau)\text{RMST}_0(\tau \mid a, X)$$

$$+ \Delta I(\widetilde{T} \leq \tau)\text{RMST}_0(\tau \wedge \widetilde{T} \mid a, X) - \Delta I(\widetilde{T} \leq \tau)\text{RMST}_0(\tau \wedge \widetilde{T} \mid a, X) - I(\widetilde{T} > \tau)\text{RMST}_0(\tau \wedge \widetilde{T} \mid a, X) = 0$$

In the competing risks setting, we transform several conditional event martingale terms

$$\int_0^{\tau \wedge \widetilde{T}} \frac{(\tau - t)\mathrm{d}M_{j,0}(t \mid a, X)}{G_0(t- \mid a, X)} = \tau \int_0^\tau \frac{\mathrm{d}M_{j,0}(t \mid a, X)}{G_0(t- \mid a, X)} - \int_0^\tau \frac{t\mathrm{d}M_{j,0}(t \mid a, X)}{G_0(t- \mid a, X)}$$

$$= \frac{\tau I(\widetilde{T} \leq \tau, \widetilde{J} = j)}{G_0(\widetilde{T}- \mid a, X)} - \tau \int_0^{\tau \wedge \widetilde{T}} \frac{\mathrm{d}\Lambda_{j,0}(t \mid a, X)}{G_0(t- \mid a, X)} - \frac{\widetilde{T}I(\widetilde{T} \leq \tau, \widetilde{J} = j)}{G_0(\widetilde{T}- \mid a, X)} + \int_0^{\tau \wedge \widetilde{T}} \frac{t\mathrm{d}\Lambda_{j,0}(t \mid a, X)}{G_0(t- \mid a, X)}$$

$$= \frac{\{\tau - \min(\widetilde{T}, \tau)\}I(\widetilde{J} = j)}{G_0(\widetilde{T}- \mid a, X)} - \tau \int_0^{\tau \wedge \widetilde{T}} \frac{\mathrm{d}\Lambda_{j,0}(t \mid a, X)}{G_0(t- \mid a, X)} + \int_0^{\tau \wedge \widetilde{T}} \frac{t\mathrm{d}\Lambda_{j,0}(t \mid a, X)}{G_0(t- \mid a, X)}$$

and

$$\mathrm{d}\left\{ \frac{\text{RMTL}_{j,0}(t \mid a, X)}{G_0(t- \mid a, X)} \right\} = \frac{F_{j,0}(t \mid a, X)G_0(t- \mid a, X)\mathrm{d}t + \text{RMTL}_{j,0}(t \mid a, X)G_0(t- \mid a, X)\mathrm{d}\Lambda_0^C(t- \mid a, X)}{G_0^2(t- \mid a, X)}$$

$$= \frac{F_{j,0}(t \mid a, X)\mathrm{d}t}{G_0(t- \mid a, X)} + \frac{\text{RMTL}_{j,0}(t \mid a, X)\mathrm{d}\Lambda_0^C(t \mid a, X)}{G_0(t- \mid a, X)}$$

and

$$\int_0^{\tau\wedge\widetilde{T}} \frac{\mathrm{RMTL}_{j,0}(t\mid a,X)\mathrm{d}M_0(t\mid a,X)}{S_0(t\mid a,X)G_0(t-\mid a,X)}$$

$$= \int_0^{\tau} \frac{\mathrm{RMTL}_{j,0}(t\mid a,X)\mathrm{d}N(t)}{S_0(t\mid a,X)G_0(t-\mid a,X)} - \int_0^{\tau\wedge\widetilde{T}} \frac{\mathrm{RMTL}_{j,0}(t\mid a,X)\mathrm{d}\Lambda_0(t\mid a,X)}{S_0(t\mid a,X)G_0(t-\mid a,X)}$$

$$= \frac{\Delta I(\widetilde{T}\le\tau)\mathrm{RMTL}_{j,0}(\widetilde{T}\mid a,X)}{S_0(\widetilde{T}\mid a,X)G_0(\widetilde{T}-\mid a,X)} - \int_0^{\tau\wedge\widetilde{T}} \frac{\mathrm{RMTL}_{j,0}(t\mid a,X)}{G_0(t-\mid a,X)}\mathrm{d}\left\{\frac{1}{S_0(t\mid a,X)}\right\}$$

$$= \frac{\Delta I(\widetilde{T}\le\tau)\mathrm{RMTL}_{j,0}(\tau\wedge\widetilde{T}\mid a,X)}{S_0(\tau\wedge\widetilde{T}\mid a,X)G_0(\tau\wedge\widetilde{T}-\mid a,X)} - \left.\frac{\mathrm{RMTL}_{j,0}(t\mid a,X)}{S_0(t\mid a,X)G_0(t-\mid a,X)}\right|_0^{\tau\wedge\widetilde{T}} + \int_0^{\tau\wedge\widetilde{T}} \frac{1}{S_0(t\mid a,X)}\mathrm{d}\left\{\frac{\mathrm{RMTL}_{j,0}(t\mid a,X)}{G_0(t-\mid a,X)}\right\}$$

$$= \frac{\Delta I(\widetilde{T}\le\tau)\mathrm{RMTL}_{j,0}(\tau\wedge\widetilde{T}\mid a,X)}{S_0(\tau\wedge\widetilde{T}\mid a,X)G_0(\tau\wedge\widetilde{T}-\mid a,X)} - \frac{\mathrm{RMTL}_{j,0}(\tau\wedge\widetilde{T}\mid a,X)}{S_0(\tau\wedge\widetilde{T}\mid a,X)G_0(\tau\wedge\widetilde{T}-\mid a,X)} + \int_0^{\tau\wedge\widetilde{T}} \frac{F_{j,0}(t\mid a,X)\mathrm{d}t}{S_0(t\mid a,X)G_0(t-\mid a,X)}$$

$$+ \int_0^{\tau\wedge\widetilde{T}} \frac{\mathrm{RMTL}_{j,0}(t\mid a,X)\mathrm{d}\Lambda_0^C(t\mid a,X)}{S_0(t\mid a,X)G_0(t-\mid a,X)}$$

$$= \frac{\Delta I(\widetilde{T}\le\tau)\mathrm{RMTL}_{j,0}(\tau\wedge\widetilde{T}\mid a,X)}{S_0(\tau\wedge\widetilde{T}\mid a,X)G_0(\tau\wedge\widetilde{T}-\mid a,X)} - \frac{\mathrm{RMTL}_{j,0}(\tau\wedge\widetilde{T}\mid a,X)}{S_0(\tau\wedge\widetilde{T}\mid a,X)G_0(\tau\wedge\widetilde{T}-\mid a,X)} + \int_0^{\tau\wedge\widetilde{T}} \frac{F_{j,0}(t\mid a,X)\mathrm{d}t}{S_0(t\mid a,X)G_0(t-\mid a,X)}$$

$$- \int_0^{\tau\wedge\widetilde{T}} \frac{\mathrm{RMTL}_{j,0}(t\mid a,X)\mathrm{d}M_0^C(t\mid a,X)}{S_0(t\mid a,X)G_0(t-\mid a,X)} + \frac{(1-\Delta)I(\widetilde{T}\le\tau)\mathrm{RMTL}_{j,0}(\tau\wedge\widetilde{T}\mid a,X)}{S_0(\tau\wedge\widetilde{T}\mid a,X)G_0(\tau\wedge\widetilde{T}-\mid a,X)}$$

$$= -\frac{I(\widetilde{T}>\tau)\mathrm{RMTL}_{j,0}(\tau\wedge\widetilde{T}\mid a,X)}{S_0(\tau\wedge\widetilde{T}\mid a,X)G_0(\tau\wedge\widetilde{T}-\mid a,X)} + \int_0^{\tau\wedge\widetilde{T}} \frac{F_{j,0}(t\mid a,X)\mathrm{d}t}{S_0(t\mid a,X)G_0(t-\mid a,X)}$$

$$- \int_0^{\tau\wedge\widetilde{T}} \frac{\mathrm{RMTL}_{j,0}(t\mid a,X)\mathrm{d}M_0^C(t\mid a,X)}{S_0(t\mid a,X)G_0(t-\mid a,X)}$$

and

$$\int_0^{\tau\wedge\widetilde{T}} \frac{tF_{j,0}(t\mid a,X)\mathrm{d}M_0(t\mid a,X)}{S_0(t\mid a,X)G_0(t-\mid a,X)}$$

$$= \int_0^{\tau\wedge\widetilde{T}} \frac{tF_{j,0}(t\mid a,X)\mathrm{d}N(t)}{S_0(t\mid a,X)G_0(t-\mid a,X)} - \int_0^{\tau\wedge\widetilde{T}} \frac{tF_{j,0}(t\mid a,X)}{G_0(t-\mid a,X)}\mathrm{d}\left\{\frac{1}{S_0(t\mid a,X)}\right\}$$

$$= \frac{\Delta I(\widetilde{T}\le\tau)(\tau\wedge\widetilde{T})F_{j,0}(\tau\wedge\widetilde{T}\mid a,X)}{S_0(\tau\wedge\widetilde{T}\mid a,X)G_0(\tau\wedge\widetilde{T}-\mid a,X)} - \left.\frac{tF_{j,0}(t\mid a,X)}{S_0(t\mid a,X)G_0(t-\mid a,X)}\right|_0^{\tau\wedge\widetilde{T}} + \int_0^{\tau\wedge\widetilde{T}} \frac{1}{S_0(t\mid a,X)}\mathrm{d}\left\{\frac{tF_{j,0}(t\mid a,X)}{G_{j,0}(t-\mid a,X)}\right\}$$

$$= \frac{\Delta I(\widetilde{T}\le\tau)(\tau\wedge\widetilde{T})F_{j,0}(\tau\wedge\widetilde{T}\mid a,X)}{S_0(\tau\wedge\widetilde{T}\mid a,X)G_0(\tau\wedge\widetilde{T}-\mid a,X)} - \frac{(\tau\wedge\widetilde{T})F_{j,0}(\tau\wedge\widetilde{T}\mid a,X)}{S_0(\tau\wedge\widetilde{T}\mid a,X)G_0(\tau\wedge\widetilde{T}-\mid a,X)}$$

$$+ \int_0^{\tau\wedge\widetilde{T}} \frac{1}{S_0(t\mid a,X)}\mathrm{d}\left\{\frac{tF_{j,0}(t\mid a,X)}{G_0(t-\mid a,X)}\right\}$$

$$= \frac{\Delta I(\widetilde{T}\le\tau)(\tau\wedge\widetilde{T})F_{j,0}(\tau\wedge\widetilde{T}\mid a,X)}{S_0(\tau\wedge\widetilde{T}\mid a,X)G_0(\tau\wedge\widetilde{T}-\mid a,X)} - \frac{(\tau\wedge\widetilde{T})F_{j,0}(\tau\wedge\widetilde{T}\mid a,X)}{S_0(\tau\wedge\widetilde{T}\mid a,X)G_0(\tau\wedge\widetilde{T}-\mid a,X)} + \int_0^{\tau\wedge\widetilde{T}} \frac{F_{j,0}(t\mid a,X)\mathrm{d}t}{S_0(t\mid a,X)G_0(t-\mid a,X)}$$

$$+ \int_0^{\tau\wedge\widetilde{T}} \frac{t\mathrm{d}\Lambda_{j,0}(t\mid a,X)}{G_0(t-\mid a,X)} - \int_0^{\tau\wedge\widetilde{T}} \frac{tF_{j,0}(t\mid a,X)\mathrm{d}M_0^C(t-\mid a,X)}{S_0(t\mid a,X)G_0(t-\mid a,X)} + \frac{(1-\Delta)I(\widetilde{T}\le\tau)(\tau\wedge\widetilde{T})F_{j,0}(\tau\wedge\widetilde{T}\mid a,X)}{S_0(\tau\wedge\widetilde{T}\mid a,X)G_0(\tau\wedge\widetilde{T}-\mid a,X)}$$

$$= -\frac{I(\widetilde{T}>\tau)(\tau\wedge\widetilde{T})F_{j,0}(\tau\wedge\widetilde{T}\mid a,X)}{S_0(\tau\wedge\widetilde{T}\mid a,X)G_0(\tau\wedge\widetilde{T}-\mid a,X)} + \int_0^{\tau\wedge\widetilde{T}} \frac{F_{j,0}(t\mid a,X)\mathrm{d}t}{S_0(t\mid a,X)G_0(t-\mid a,X)} + \int_0^{\tau\wedge\widetilde{T}} \frac{t\mathrm{d}\Lambda_{j,0}(t\mid a,X)}{G_0(t-\mid a,X)}$$

$$- \int_0^{\tau\wedge\widetilde{T}} \frac{tF_{j,0}(t\mid a,X)\mathrm{d}M_0^C(t-\mid a,X)}{S_0(t\mid a,X)G_0(t-\mid a,X)}$$

and

$$d\left\{\frac{tF_{j,0}(t \mid a,X)}{G_0(t- \mid a,X)}\right\}$$

$$=\frac{G_0(t- \mid a,X)\{F_{j,0}(t \mid a,X) + tS_0(t \mid a,X)d\Lambda_{j,0}(t \mid a,X)\} + tF_{j,0}(t \mid a,X)G_0(t- \mid a,X)d\Lambda_0^C(t- \mid a,X)}{G_0^2(t- \mid a,X)}$$

$$=\frac{F_{j,0}(t \mid a,X)}{G_0(t- \mid a,X)} + \frac{tS_0(t \mid a,X)d\Lambda_{j,0}(t \mid a,X)}{G_0(t- \mid a,X)} + \frac{tF_{j,0}(t \mid a,X)d\Lambda_0^C(t- \mid a,X)}{G_0(t- \mid a,X)}$$

Substitute all preparation equalities into the RHS of the second equation in Lemma 1 gives,

$$\frac{I(A=a)}{\pi_0(a \mid X)}\int_0^{\tau \wedge \widetilde{T}} \frac{(\tau - t)dM_{j,0}(t \mid a,X)}{G_0(t- \mid a,X)} - \frac{I(A=a)}{\pi_0(a \mid X)}$$

$$\times \int_0^{\tau \wedge \widetilde{T}} \frac{\{\text{RMTL}_{j,0}(\tau \mid a,X) - \text{RMTL}_{j,0}(t \mid a,X)\}dM_0(t \mid a,X)}{S_0(t \mid a,X)G_0(t- \mid a,X)} + \frac{I(A=a)}{\pi_0(a \mid X)}$$

$$\times \int_0^{\tau \wedge \widetilde{T}} \frac{(\tau - t)F_{j,0}(t \mid a,X)dM_0(t \mid a,X)}{S_0(t \mid a,X)G_0(t- \mid a,X)} + \text{RMTL}_{j,0}(\tau \mid a,X)$$

$$=\frac{I(A=a)\{\tau - \min(\widetilde{T},\tau)\}I(\widetilde{J}=j)}{\pi_0(a \mid X)G_0(\widetilde{T}- \mid a,X)} - \frac{I(A=a)\tau}{\pi_0(a \mid X)}\int_0^{\tau \wedge \widetilde{T}} \frac{d\Lambda_{j,0}(t \mid a,X)}{G_0(t- \mid a,X)}$$

$$+ \frac{I(A=a)}{\pi_0(a \mid X)}\int_0^{\tau \wedge \widetilde{T}} \frac{td\Lambda_{j,0}(t \mid a,X)}{G_0(t- \mid a,X)} + \frac{I(A=a)I(\widetilde{T}>\tau)\text{RMTL}_{j,0}(\tau \mid a,X)}{\pi_0(a \mid X)S_0(\tau \wedge \widetilde{T}- \mid a,X)G_0(\tau \wedge \widetilde{T}- \mid a,X)}$$

$$- \frac{I(A=a)\text{RMTL}_{j,0}(\tau \mid a,X)}{\pi_0(a \mid X)} + \frac{I(A=a)\text{RMTL}_{j,0}(\tau \mid a,X)}{\pi_0(a \mid X)}$$

$$\times \int_0^{\tau \wedge \widetilde{T}} \frac{dM_0^C(t \mid a,X)}{S_0(t \mid a,X)G_0(t- \mid a,X)} - \frac{I(A=a)I(\widetilde{T}>\tau)\text{RMTL}_{j,0}(\tau \wedge \widetilde{T} \mid a,X)}{\pi_0(a \mid X)S_0(\tau \wedge \widetilde{T} \mid a,X)G_0(\tau \wedge \widetilde{T}- \mid a,X)}$$

$$+ \frac{I(A=a)}{\pi_0(a \mid X)}\int_0^{\tau \wedge \widetilde{T}} \frac{F_{j,0}(t \mid a,X)dt}{S_0(t \mid a,X)G_0(t- \mid a,X)} - \frac{I(A=a)}{\pi_0(a \mid X)}  \qquad \blacksquare$$

$$\times \int_0^{\tau \wedge \widetilde{T}} \frac{\text{RMTL}_{j,0}(t \mid a,X)dM_0^C(t \mid a,X)}{S_0(t \mid a,X)G_0(t- \mid a,X)} - \frac{I(A=a)\tau I(\widetilde{T}>\tau)F_{j,0}(\tau \mid a,X)}{\pi_0(a \mid X)S_0(\tau \wedge \widetilde{T} \mid a,X)G_0(\tau \wedge \widetilde{T}- \mid a,X)}$$

$$+ \frac{I(A=a)\tau}{\pi_0(a \mid X)}\int_0^{\tau \wedge \widetilde{T}} \frac{d\Lambda_{j,0}(t \mid a,X)}{G_0(t- \mid a,X)} - \frac{I(A=a)\tau}{\pi_0(a \mid X)}\int_0^{\tau \wedge \widetilde{T}} \frac{F_{j,0}(t \mid a,X)dM_0^C(t \mid a,X)}{S_0(t \mid a,X)G_0(t- \mid a,X)}$$

$$+ \frac{I(A=a)I(\widetilde{T}>\tau)(\tau \wedge \widetilde{T})F_{j,0}(\tau \wedge \widetilde{T} \mid a,X)}{\pi_0(a \mid X)S_0(\tau \wedge \widetilde{T} \mid a,X)G_0(\tau \wedge \widetilde{T}- \mid a,X)} - \frac{I(A=a)}{\pi_0(a \mid X)}\int_0^{\tau \wedge \widetilde{T}} \frac{F_{j,0}(t \mid a,X)dt}{S_0(t \mid a,X)G_0(t- \mid a,X)}$$

$$- \frac{I(A=a)}{\pi_0(a \mid X)}\int_0^{\tau \wedge \widetilde{T}} \frac{td\Lambda_{j,0}(t \mid a,X)}{G_0(t- \mid a,X)} + \frac{I(A=a)}{\pi_0(a \mid X)}\int_0^{\tau \wedge \widetilde{T}} \frac{tF_{j,0}(t \mid a,X)dM_0^C(t- \mid a,X)}{S_0(t \mid a,X)G_0(t- \mid a,X)}$$

$$+ \text{RMTL}_{j,0}(\tau \mid a,X)$$

$$=\frac{I(A=a)\{\tau - \min(\widetilde{T},\tau)\}I(\widetilde{J}=j)}{\pi_0(a \mid X)G_0(\widetilde{T}- \mid a,X)} + \left\{1 - \frac{I(A=a)}{\pi_0(a \mid X)}\right\}\text{RMTL}_{j,0}(\tau \mid a,X) + \frac{I(A=a)}{\pi_0(a \mid X)}$$

$$\times \int_0^{\tau \wedge \widetilde{T}} \frac{\{(t-\tau)F_{j,0}(t \mid a,X) + \text{RMTL}_{j,0}(\tau \mid a,X) - \text{RMTL}_{j,0}(t \mid a,X)\}dM_0^C(t \mid a,X)}{S_0(t \mid a,X)G_0(t- \mid a,X)}$$

$$=\phi_{j,0}^{\text{RMTL},h}(\tau,a;\eta_{j,0}^\Lambda)$$

## D.4    Lemma 2

**Lemma 2**

$$
P_0\{\widehat{\phi}^{\mathrm{RMST},h}(\tau,a;\widehat{\eta}^{\Lambda,h})\} - \psi_0^{\mathrm{RMST},h}(\tau,a)
$$

$$
=[P_n\{\widehat{w}^h(a,X)\}]^{-1}\left(E_0\left[\widehat{h}(X)\widehat{\mathrm{RMST}}(\tau\mid a,X)\int_0^\tau \frac{S_0(t\mid a,X)}{\widehat{S}(t\mid a,X)}\left\{\frac{\pi_0(a\mid X)G_0(t-\mid a,X)}{\widehat{\pi}(a\mid X)\widehat{G}(t-\mid a,X)}-1\right\}\right.\right.
$$

$$
\times\{\mathrm{d}\widehat{\Lambda}(t\mid a,X)-\mathrm{d}\Lambda_0(t\mid a,X)\}\Big]
$$

$$
-E_0\left[\widehat{h}(X)\int_0^\tau \frac{\widehat{\mathrm{RMST}}(t\mid a,X)S_0(t\mid a,X)}{\widehat{S}(t\mid a,X)}\left\{\frac{\pi_0(a\mid X)G_0(t-\mid a,X)}{\widehat{\pi}(a\mid X)\widehat{G}(t-\mid a,X)}-1\right\}\{\mathrm{d}\widehat{\Lambda}(t\mid a,X)-\mathrm{d}\Lambda_0(t\mid a,X)\}\right]\Bigg)
$$

$$
+([P_n\{\widehat{h}(X)\}]^{-1}-[P_n\{\widehat{w}^h(a,X)\}]^{-1})E_0\{\widehat{h}(X)\widehat{\mathrm{RMST}}(\tau\mid a,X)\}
$$

$$
+([P_n\{\widehat{w}^h(a,X)\}]^{-1}-[E_0\{h_0(X)\}]^{-1})E_0\{\widehat{h}(X)\mathrm{RMST}_0(\tau\mid a,X)\}
$$

$$
+[E\{h_0(x)\}]^{-1}E_0[\{\widehat{h}(X)-h_0(X)\}\mathrm{RMST}_0(\tau\mid a,X)]
$$

*and*

$$
P\{\widehat{\phi}_j^{\mathrm{RMTL},h}(\tau,a;\widehat{\eta}_j^{\Lambda,h})\} - \psi_{j,0}^{\mathrm{RMTL},h}(\tau,a)
$$

$$
=[P_n\{\widehat{w}^h(a,X)\}]^{-1}\left(-E_0\left[\widehat{h}(X)\tau\int_0^\tau S_0(t\mid a,X)\left\{\frac{\pi_0(a\mid X)G_0(t-\mid a,X)}{\widehat{\pi}(a\mid X)\widehat{G}(t-\mid a,X)}-1\right\}\right.\right.
$$

$$
\times\{\mathrm{d}\widehat{\Lambda}_j(t\mid a,X)-\mathrm{d}\Lambda_{j,0}(t\mid a,X)\}\Big]+E_0\left[\widehat{h}(X)\int_0^\tau tS_0(t\mid a,X)\left\{\frac{\pi_0(a\mid X)G_0(t-\mid a,X)}{\widehat{\pi}(a\mid X)\widehat{G}(t-\mid a,X)}-1\right\}\right.
$$

$$
\times\{\mathrm{d}\widehat{\Lambda}_j(t\mid a,X)-\mathrm{d}\Lambda_{j,0}(t\mid a,X)\}\Big]+E_0\left[\widehat{h}(X)\widehat{\mathrm{RMTL}}_j(\tau\mid a,X)\int_0^\tau \frac{S_0(t\mid a,X)}{\widehat{S}(t\mid a,X)}\right.
$$

$$
\times\left\{\frac{\pi(a\mid X)G_0(t-\mid a,X)}{\widehat{\pi}(a\mid X)\widehat{G}(t-\mid a,X)}-1\right\}\{\mathrm{d}\widehat{\Lambda}(t\mid a,X)-\mathrm{d}\Lambda_0(t\mid a,X)\}\Big]
$$

$$
-E_0\left[\widehat{h}(X)\int_0^\tau \frac{\widehat{\mathrm{RMTL}}_j(t\mid a,X)S_0(t\mid a,X)}{\widehat{S}(t\mid a,X)}\left\{\frac{\pi_0(a\mid X)G_0(t-\mid a,X)}{\widehat{\pi}(a\mid X)\widehat{G}(t-\mid a,X)}-1\right\}\{\mathrm{d}\widehat{\Lambda}(t\mid a,X)-\mathrm{d}\Lambda_0(t\mid a,X)\}\right]
$$

$$
-E_0\left[\widehat{h}(X)\tau\int_0^\tau \frac{\widehat{F}_j(t\mid a,X)S_0(t\mid a,X)}{\widehat{S}(t\mid a,X)}\left\{\frac{\pi_0(a\mid X)G_0(t-\mid a,X)}{\widehat{\pi}(a\mid X)\widehat{G}(t-\mid a,X)}-1\right\}\{\mathrm{d}\widehat{\Lambda}(t\mid a,X)-\mathrm{d}\Lambda_0(t\mid a,X)\}\right]
$$

$$
+E_0\left[\widehat{h}(X)\int_0^\tau \frac{t\widehat{F}_j(t\mid a,X)S_0(t\mid a,X)}{\widehat{S}(t\mid a,X)}\left\{\frac{\pi_0(a\mid X)G_0(t-\mid a,X)}{\widehat{\pi}(a\mid X)\widehat{G}(t-\mid a,X)}-1\right\}\{\mathrm{d}\widehat{\Lambda}(t\mid a,X)-\mathrm{d}\Lambda_0(t\mid a,X)\}\right]\Bigg)
$$

$$
+([P_n\{\widehat{h}(X)\}]^{-1}-[P_n\{\widehat{w}^h(a,X)\}]^{-1})E_0\{\widehat{h}(X)\widehat{\mathrm{RMTL}}_j(\tau\mid a,X)\}
$$

$$
+([P_n\{\widehat{w}^h(a,X)\}]^{-1}-[E_0\{h_0(X)\}]^{-1})E_0\{\widehat{h}(X)\mathrm{RMTL}_{j,0}(\tau\mid a,X)\}
$$

$$
+[E\{h_0(X)\}]^{-1}E_0[\{\widehat{h}(X)-h_0(X)\}\mathrm{RMTL}_{j,0}(\tau\mid a,X)]
$$

Proof of Lemma 2:

$$
P\{\widehat{\phi}^{\mathrm{RMST},h}(\tau,a;\widehat{\eta}^{\Lambda,h})\} - \psi_0^{\mathrm{RMST},h}(\tau,a)
$$

$$
=[P_n\{\widehat{w}^h(a,X)\}]^{-1}E_0\left[\frac{\widehat{h}(X)\pi_0(a\mid x)}{\widehat{\pi}(a\mid X)}E_0\left\{-\widehat{\mathrm{RMST}}(\tau\mid a,X)\int_0^{\tau\wedge\widetilde{T}}\frac{\mathrm{d}\widehat{M}(t\mid a,X)}{\widehat{S}(t\mid a,X)\widehat{G}(t-\mid a,X)}\right.\right.
$$

$$
+\int_0^{\tau\wedge\widetilde{T}}\frac{\widehat{\mathrm{RMST}}(t\mid a,X)\mathrm{d}\widehat{M}(t\mid a,X)}{\widehat{S}(t\mid a,X)\widehat{G}(t-\mid a,X)}\mid A=a,X\Bigg\}\Bigg]
$$

$$
+[P_n\{\widehat{h}(X)\}]^{-1}E_0[\widehat{h}(X)\{\widehat{\mathrm{RMST}}(\tau\mid a,X)\}]-[E_0\{h_0(X)\}]^{-1}E_0[h_0(X)\mathrm{RMST}_0(\tau\mid a,X)]
$$

$$
=[P_n\{\widehat{w}^h(a,X)\}]^{-1}E_0\left[\frac{\widehat{h}(X)\pi_0(a\mid X)}{\widehat{\pi}(a\mid X)}E_0\left\{-\widehat{\mathrm{RMST}}(\tau\mid a,X)\int_0^{\tau\wedge\widetilde{T}}\frac{\mathrm{d}\widehat{M}(t\mid a,X)}{\widehat{S}(t\mid a,X)\widehat{G}(t-\mid a,X)}\right.\right.
$$

$$
\left.\left.+\int_0^{\tau\wedge\widetilde{T}}\frac{\widehat{\mathrm{RMST}}(t\mid a,X)\mathrm{d}\widehat{M}(t\mid a,X)}{\widehat{S}(t\mid a,X)\widehat{G}(t-\mid a,X)}\mid A=a,X\right\}+\widehat{h}(X)\{\widehat{\mathrm{RMST}}(\tau\mid a,X)-\mathrm{RMST}_0(\tau\mid a,X)\}\right]
$$

$$
+[P_n\{\widehat{h}(X)\}]^{-1}E_0[\widehat{h}(X)\{\widehat{\mathrm{RMST}}(\tau\mid a,X)\}]-[E_0\{h_0(X)\}]^{-1}E_0[h_0(X)\mathrm{RMST}_0(\tau\mid a,X)]
$$

$$
+([P_n\{\widehat{h}(X)\}]^{-1}-[P_n\{\widehat{w}^h(a,X)\}]^{-1})E_0\{\widehat{h}(X)\widehat{\mathrm{RMST}}(\tau\mid a,X)\}
$$

$$
+([P_n\{\widehat{w}^h(a,X)\}]^{-1}-[E_0\{h_0(X)\}]^{-1})E_0\{\widehat{h}(X)\mathrm{RMST}_0(\tau\mid a,X)\}
$$

$$
+[E_0\{h_0(X)\}]^{-1}E_0[\{\widehat{h}(X)-h_0(X)\}\mathrm{RMST}_0(\tau\mid a,X)]
$$

For the first conditional event martingale integral term,

$$
E_0\left\{\int_0^{\tau\wedge\widetilde{T}}\frac{\mathrm{d}\widehat{M}(t\mid a,X)}{\widehat{S}(t\mid a,X)\widehat{G}(t-\mid a,X)}\mid A=a,X\right\}
$$

$$
=-\int_0^\tau\frac{S_0(t-\mid a,X)G_0(t-\mid a,X)\{\mathrm{d}\widehat{\Lambda}(t\mid a,X)-\mathrm{d}\Lambda_0(t\mid a,X)\}}{\widehat{S}(t\mid a,X)\widehat{G}(t-\mid a,X)}
$$

and the second conditional event martingale integral term

$$
E_0\left\{\int_0^{\tau\wedge\widetilde{T}}\frac{\widehat{\mathrm{RMST}}(t\mid a,X)\mathrm{d}\widehat{M}(t\mid a,X)}{\widehat{S}(t\mid a,X)\widehat{G}(t-\mid a,X)}\mid A=a,X\right\}
$$

$$
=E_0\left\{\frac{\Delta I(\widetilde{T}\le\tau)\widehat{\mathrm{RMST}}(\widetilde{T}\mid a,X)}{\widehat{S}(\widetilde{T}\mid a,X)\widehat{G}(\widetilde{T}-\mid a,X)}-\int_0^{\tau\wedge\widetilde{T}}\frac{\widehat{\mathrm{RMST}}(t\mid a,X)\mathrm{d}\widehat{\Lambda}(t\mid a,X)}{\widehat{S}(t\mid a,X)\widehat{G}(t-\mid a,X)}\mid A=a,X\right\}
$$

$$
=-\int_0^\tau\frac{S_0(t-\mid a,X)G_0(t-\mid a,X)\widehat{\mathrm{RMST}}(t\mid a,X)\{\mathrm{d}\widehat{\Lambda}(t\mid a,X)-\mathrm{d}\Lambda_0(t\mid a,X)\}}{\widehat{S}(t\mid a,X)\widehat{G}(t-\mid a,X)}
$$

The Duhamel equation (Gill & Johansen, 1990) gives

$$
S_0(t\mid a,X)-\widehat{S}(t\mid a,X)=\widehat{S}(t\mid a,X)\int_0^t\frac{S_0(u-\mid a,X)}{\widehat{S}(u\mid a,X)}\{\mathrm{d}\widehat{\Lambda}(u\mid a,X)-\mathrm{d}\Lambda_0(u\mid a,X)\}
$$

such that

$$
\widehat{\mathrm{RMST}}(t\mid a,X)-\mathrm{RMST}_0(t\mid a,X)=\int_0^\tau\{\widehat{S}(t\mid a,X)-S_0(t\mid a,X)\}\mathrm{d}t
$$

$$
=-\int_0^\tau\left[\widehat{S}(t\mid a,X)\int_0^t\frac{S_0(u-\mid a,X)}{\widehat{S}(u\mid a,X)}\{\mathrm{d}\widehat{\Lambda}(u\mid a,X)-\mathrm{d}\Lambda_0(u\mid a,X)\}\right]\mathrm{d}t
$$

$$
=-\widehat{\mathrm{RMST}}(\tau\mid a,X)\int_0^\tau\frac{S_0(t-\mid a,X)}{\widehat{S}(t\mid a,X)}\{\mathrm{d}\widehat{\Lambda}(t\mid a,X)-\mathrm{d}\Lambda_0(t\mid a,X)\}
$$

$$
+\int_0^\tau\frac{\widehat{\mathrm{RMST}}(t\mid a,X)S_0(t-\mid a,X)}{\widehat{S}(t\mid a,X)}\{\mathrm{d}\widehat{\Lambda}(t\mid a,X)-\mathrm{d}\Lambda_0(t\mid a,X)\}
$$

Substitute the conditional event martingale integral terms and $\widehat{\mathrm{RMST}}(t\mid a,X)-\mathrm{RMST}_0(t\mid a,X)$ into $P_0\{\widehat{\phi}^{\mathrm{RMST},h}(\tau,a;\widehat{\eta}^{\Lambda,h})\}-\psi_0^{\mathrm{RMST},h}(\tau,a)$ yields the equality as presented in Lemma 2.

Then, we consider the competing risks setting

$$P_0\{\widehat{\phi}_j^{\mathrm{RMTL},h}(\tau,a;\widehat{\eta}_j^{\Lambda,h})\} - \psi_{j,0}^{\mathrm{RMTL},h}(\tau,a)$$

$$=[P_n\{\widehat{w}^h(a,X)\}]^{-1}E_0\left[\frac{\widehat{h}(X)\pi_0(a\mid X)}{\widehat{\pi}(a\mid X)}E_0\left\{\tau\int_0^{\tau\wedge\widetilde{T}}\frac{\mathrm{d}\widehat{M}_j(t\mid a,X)}{\widehat{G}(t-\mid a,X)} - \int_0^{\tau\wedge\widetilde{T}}\frac{t\mathrm{d}\widehat{M}_j(t\mid a,X)}{\widehat{G}(t-\mid a,X)}\right.\right.$$

$$-\mathrm{R\widehat{MTL}}_j(\tau\mid a,X)\int_0^{\tau\wedge\widetilde{T}}\frac{\mathrm{d}\widehat{M}(t\mid a,X)}{\widehat{S}(t\mid a,X)\widehat{G}(t-\mid a,X)} + \int_0^{\tau\wedge\widetilde{T}}\frac{\mathrm{R\widehat{MTL}}_j(t\mid a,X)\mathrm{d}\widehat{M}(t\mid a,X)}{\widehat{S}(t\mid a,X)\widehat{G}(t-\mid a,X)}$$

$$+\tau\int_0^{\tau\wedge\widetilde{T}}\frac{\widehat{F}_j(t\mid a,X)\mathrm{d}\widehat{M}(t\mid a,X)}{\widehat{S}(t\mid a,X)\widehat{G}(t-\mid a,X)} - \int_0^{\tau\wedge\widetilde{T}}\frac{t\widehat{F}_j(t\mid a,X)\mathrm{d}\widehat{M}(t\mid a,X)}{\widehat{S}(t\mid a,X)\widehat{G}(t-\mid a,X)}\mid A=a,X\left.\right\}\right]$$

$$+[P_n\{\widehat{h}(X)\}]^{-1}E[\widehat{h}(X)\{\mathrm{R\widehat{MTL}}_j(\tau\mid a,X)\}] - [E_0\{h_0(X)\}]^{-1}E_0[h_0(X)\mathrm{RMTL}_{j,0}(\tau\mid a,X)]$$

$$=[P_n\{\widehat{w}^h(a,X)\}]^{-1}E_0\left[\frac{\widehat{h}(X)\pi_0(a\mid X)}{\widehat{\pi}(a\mid X)}E_0\left\{\tau\int_0^{\tau\wedge\widetilde{T}}\frac{\mathrm{d}\widehat{M}_j(t\mid a,X)}{\widehat{G}(t-\mid a,X)} - \int_0^{\tau\wedge\widetilde{T}}\frac{t\mathrm{d}\widehat{M}_j(t\mid a,X)}{\widehat{G}(t-\mid a,X)}\right.\right.$$

$$-\mathrm{R\widehat{MTL}}_j(\tau\mid a,X)\int_0^{\tau\wedge\widetilde{T}}\frac{\mathrm{d}\widehat{M}(t\mid a,X)}{\widehat{S}(t\mid a,X)\widehat{G}(t-\mid a,X)} + \int_0^{\tau\wedge\widetilde{T}}\frac{\mathrm{R\widehat{MTL}}_j(t\mid a,X)\mathrm{d}\widehat{M}(t\mid a,X)}{\widehat{S}(t\mid a,X)\widehat{G}(t-\mid a,X)}$$

$$+\tau\int_0^{\tau\wedge\widetilde{T}}\frac{\widehat{F}_j(t\mid a,X)\mathrm{d}\widehat{M}(t\mid a,X)}{\widehat{S}(t\mid a,X)\widehat{G}(t-\mid a,X)} - \int_0^{\tau\wedge\widetilde{T}}\frac{t\widehat{F}_j(t\mid a,X)\mathrm{d}\widehat{M}(t\mid a,X)}{\widehat{S}(t\mid a,X)\widehat{G}(t-\mid a,X)}\mid A=a,X\left.\right\}$$

$$+\widehat{h}(X)\{\mathrm{R\widehat{MST}}(\tau\mid a,X) - \mathrm{RMST}(\tau\mid a,X)\}\right]$$

$$+([P_n\{\widehat{h}(X)\}]^{-1} - [P_n\{\widehat{w}^h(a,X)\}]^{-1})E_0\{\widehat{h}(X)\mathrm{R\widehat{MTL}}_j(\tau\mid a,X)\}$$

$$+([P_n\{\widehat{w}^h(a,X)\}]^{-1} - [E_0\{h_0(X)\}]^{-1})E_0\{\widehat{h}(X)\mathrm{RMTL}_{j,0}(\tau\mid a,X)\}$$

$$+[E_0\{h(X)\}]^{-1}E_0[\{\widehat{h}(X) - h_0(X)\}\mathrm{RMTL}_{j,0}(\tau\mid a,X)]$$

We derive equalities on conditional cause-specific martingale integrals

$$E_0\left\{\int_0^{\tau\wedge\widetilde{T}}\frac{\mathrm{d}\widehat{M}_j(t\mid a,X)}{\widehat{G}(t-\mid a,X)}\mid A=a,X\right\} = -\int_0^\tau\frac{S_0(t\mid a,X)G_0(t-\mid a,X)\{\mathrm{d}\widehat{\Lambda}_j(t\mid a,X) - \mathrm{d}\Lambda_{j,0}(t\mid a,X)\}}{\widehat{G}(t-\mid a,X)}$$

$$E_0\left\{\int_0^{\tau\wedge\widetilde{T}}\frac{t\mathrm{d}\widehat{M}_j(t\mid a,X)}{\widehat{G}(t-\mid a,X)}\mid A=a,X\right\} = -\int_0^\tau\frac{tS_0(t\mid a,X)G_0(t-\mid a,X)\{\mathrm{d}\widehat{\Lambda}_j(t\mid a,X) - \mathrm{d}\Lambda_{j,0}(t\mid a,X)\}}{\widehat{G}(t-\mid a,X)}$$

Similar for the conditional event martingale integrals,

$$E_0\left\{\int_0^{\tau\wedge\widetilde{T}}\frac{\mathrm{R\widehat{MTL}}_j(t\mid a,X)\mathrm{d}\widehat{M}(t\mid a,X)}{\widehat{S}(t\mid a,X)\widehat{G}(t-\mid a,X)}\mid A=a,X\right\}$$

$$=-\int_0^\tau\frac{S_0(t\mid a,X)G_0(t-\mid a,X)\mathrm{R\widehat{MTL}}_j(t\mid a,X)\{\mathrm{d}\widehat{\Lambda}(t\mid a,X) - \mathrm{d}\Lambda_0(t\mid a,X)\}}{\widehat{S}(t\mid a,X)\widehat{G}(t-\mid a,X)}$$

$$E_0\left\{\int_0^{\tau\wedge\widetilde{T}}\frac{\widehat{F}_j(t\mid a,X)\mathrm{d}\widehat{M}(t\mid a,X)}{\widehat{S}(t\mid a,X)\widehat{G}(t-\mid a,X)}\mid A=a,X\right\}$$

$$=-\int_0^\tau\frac{S_0(t\mid a,X)G_0(t-\mid a,X)\widehat{F}_j(t\mid a,X)\{\mathrm{d}\widehat{\Lambda}(t\mid a,X) - \mathrm{d}\Lambda_0(t\mid a,X)\}}{\widehat{S}(t\mid a,X)\widehat{G}(t-\mid a,X)}$$

$$E_0\left\{\int_0^{\tau\wedge\widetilde{T}}\frac{t\widehat{F}_j(t\mid a,X)\mathrm{d}\widehat{M}(t\mid a,X)}{\widehat{S}(t\mid a,X)\widehat{G}(t-\mid a,X)}\mid A=a,X\right\}$$

$$=-\int_0^\tau\frac{tS_0(t\mid a,X)G_0(t-\mid a,X)\widehat{F}_j(t\mid a,X)\{\mathrm{d}\widehat{\Lambda}(t\mid a,X) - \mathrm{d}\Lambda_0(t\mid a,X)\}}{\widehat{S}(t\mid a,X)\widehat{G}(t-\mid a,X)}$$

We wish to derive $\widehat{\mathrm{RMTL}}_{j,0}(\tau \mid a, X) - \mathrm{RMTL}_j(\tau \mid a, X)$ in terms of conditional hazards differences. To begin with, we consider

$$\widehat{F}_j(t \mid a, X) - F_{j,0}(t \mid a, X) = \int_0^t \widehat{S}(u- \mid a, X)\mathrm{d}\widehat{\Lambda}_j(u \mid a, X) - \int_0^t S_0(u- \mid a, X)\mathrm{d}\Lambda_{j,0}(u \mid a, X)$$

$$= \int_0^t \{\widehat{S}(u- \mid a, X) - S_0(u- \mid a, X)\}\mathrm{d}\widehat{\Lambda}_j(u \mid a, X) + \int_0^t S_0(u- \mid a, X)\{\mathrm{d}\widehat{\Lambda}_j(u \mid a, X) - \mathrm{d}\Lambda_{j,0}(u \mid a, X)\}$$

$$= \int_0^t \left[-\widehat{S}(u- \mid a, X)\int_0^u \frac{S_0(s- \mid a, X)}{\widehat{S}(s- \mid a, X)}\{\mathrm{d}\widehat{\Lambda}(s- \mid a, X) - \mathrm{d}\Lambda_0(s- \mid a, X)\}\right]\mathrm{d}\widehat{\Lambda}_j(u \mid a, X)$$

$$+ \int_0^t S_0(u- \mid a, X)\{\mathrm{d}\widehat{\Lambda}_j(u \mid a, X) - \mathrm{d}\Lambda_{j,0}(u \mid a, X)\}$$

$$= \int_0^t \left[-\int_0^u \frac{S_0(s- \mid a, X)}{\widehat{S}(s- \mid a, X)}\{\mathrm{d}\widehat{\Lambda}(s \mid a, X) - \mathrm{d}\Lambda_0(s \mid a, X)\}\right]\mathrm{d}\widehat{F}_j(u \mid a, X)$$

$$+ \int_0^t S_0(u- \mid a, X)\{\mathrm{d}\widehat{\Lambda}_j(u \mid a, X) - \mathrm{d}\Lambda_{j,0}(u \mid a, X)\}$$

$$= -\widehat{F}_j(t \mid a, X)\int_0^t \frac{S_0(u- \mid a, X)}{\widehat{S}(u- \mid a, X)}\{\mathrm{d}\widehat{\Lambda}(u \mid a, X) - \mathrm{d}\Lambda_0(u \mid a, X)\} + \int_0^t \frac{\widehat{F}_j(u \mid a, X)S_0(u- \mid a, X)}{\widehat{S}(u- \mid a, X)}$$

$$\times \{\mathrm{d}\widehat{\Lambda}(u \mid a, X) - \mathrm{d}\Lambda_0(u \mid a, X)\} + \int_0^t S_0(u- \mid a, X)\{\mathrm{d}\widehat{\Lambda}_j(u \mid a, X) - \mathrm{d}\Lambda_{j,0}(u \mid a, X)\}$$

Since the integral of the first term

$$\int_0^\tau \left[\widehat{F}_j(t \mid a, X)\int_0^t \frac{S_0(u- \mid a, X)}{\widehat{S}(u- \mid a, X)}\{\mathrm{d}\widehat{\Lambda}(u \mid a, X) - \mathrm{d}\Lambda_0(u \mid a, X)\}\right]\mathrm{d}t$$

$$= \widehat{\mathrm{RMTL}}_j(\tau \mid a, X)\int_0^\tau \frac{S_0(t- \mid a, X)}{\widehat{S}(t- \mid a, X)}\{\mathrm{d}\widehat{\Lambda}(t \mid a, X) - \mathrm{d}\Lambda_0(t \mid a, X)\}$$

$$- \int_0^\tau \frac{\widehat{\mathrm{RMTL}}_j(t \mid a, X)S_0(t- \mid a, X)}{\widehat{S}(t- \mid a, X)}\{\mathrm{d}\widehat{\Lambda}(t \mid a, X) - \mathrm{d}\Lambda_0(t \mid a, X)\}$$

the integral of the second term

$$\int_0^\tau \left[\int_0^t \frac{\widehat{F}_j(u \mid a, X)S_0(u- \mid a, X)}{\widehat{S}(u- \mid a, X)}\{\mathrm{d}\widehat{\Lambda}(u \mid a, X) - \mathrm{d}\Lambda_0(u \mid a, X)\}\right]\mathrm{d}t$$

$$= \tau\int_0^\tau \frac{\widehat{F}_j(t \mid a, X)S_0(t- \mid a, X)}{\widehat{S}(t- \mid a, X)}\{\mathrm{d}\widehat{\Lambda}(t \mid a, X) - \mathrm{d}\Lambda_0(t \mid a, X)\}$$

$$- \int_0^\tau \frac{t\widehat{F}_j(t \mid a, X)S_0(t- \mid a, X)}{\widehat{S}(t- \mid a, X)}\{\mathrm{d}\widehat{\Lambda}(t \mid a, X) - \mathrm{d}\Lambda_0(t \mid a, X)\}$$

the integral of the third term

$$\int_0^\tau \left[\int_0^t S_0(u- \mid a, X)\{\mathrm{d}\widehat{\Lambda}_j(u \mid a, X) - \mathrm{d}\Lambda_{j,0}(u \mid a, X)\}\right]\mathrm{d}t$$

$$= \int_0^\tau tS_0(t- \mid a, X)\{\mathrm{d}\Lambda_{j,0}(t \mid a, X) - \mathrm{d}\widehat{\Lambda}_j(t \mid a, X)\} - \tau\int_0^\tau S_0(t- \mid a, X)\{\mathrm{d}\Lambda_{j,0}(t \mid a, X) - \mathrm{d}\widehat{\Lambda}_j(t \mid a, X)\}$$

To make use of the Duhamel equation, we also assume $\widehat{S}$ is continuous, i.e. $\widehat{S}(t- \mid a, X) = \widehat{S}(t \mid a, X)$, we have

$$\widehat{\mathrm{RMTL}}_j(\tau \mid a, X) - \mathrm{RMTL}_{j,0}(\tau \mid a, X) = \int_0^\tau \{\widehat{F}_j(t \mid a, X) - F_{j,0}(t \mid a, X)\}\mathrm{d}t$$

$$= -\int_0^\tau \frac{S_0(t- \mid a, X)\{\widehat{\mathrm{RMTL}}_j(\tau \mid a, X) - \widehat{\mathrm{RMTL}}_j(t \mid a, X)\}}{\widehat{S}(t \mid a, X)}\{\mathrm{d}\widehat{\Lambda}(t \mid a, X) - \mathrm{d}\Lambda_0(t \mid a, X)\}$$

$$+ \int_0^\tau \frac{(\tau - t)\widehat{F}_j(t \mid a, X)S_0(t- \mid a, X)}{\widehat{S}(t \mid a, X)}\{\mathrm{d}\widehat{\Lambda}(t \mid a, X) - \mathrm{d}\Lambda_0(t \mid a, X)\}$$

$$+ \int_0^\tau (\tau - t)S_0(t- \mid a, X)\{\mathrm{d}\widehat{\Lambda}_j(t \mid a, X) - \mathrm{d}\Lambda_{j,0}(t \mid a, X)\} \quad \blacksquare$$

After replacing conditional martingale integral terms and $\widehat{\mathrm{RMTL}}_j(t \mid a, X) - \mathrm{RMTL}_{j,0}(t \mid a, X)$ by conditional hazards differences, we have the equality for $P_0\{\widehat{\phi}_j^{\mathrm{RMTL},h}(\tau, a; \widehat{\eta}_j^{\Lambda,h})\} - \psi_{j,0}^{\mathrm{RMTL},h}(\tau, a)$ as presented.

## D.5 Lemma 3

**Lemma 3** *If Condition 2 holds, then for every $a$, $\tau$, and $k$, two universal constant vectors $C_\iota^{\mathrm{RMST},h}$ and $C_{j,\iota}^{\mathrm{RMTL},h}$ ensure that*

$$
[P_0\{\widehat{\phi}_k^{\mathrm{RMST},h}(\tau, a; \widehat{\eta}_k^{\Lambda,h}) - \phi_\infty^{\mathrm{RMST},h}(\tau, a; \eta_\infty^{\Lambda,h})\}^2]^{1/2} \leq \sum_{\iota=1}^{6} C_\iota^{\mathrm{RMST},h} \mathcal{B}_{\iota,k}^{\mathrm{RMST},h}(\widehat{\eta}_k^{\Lambda,h}, \eta_\infty^{\Lambda,h})
$$

$$
[P_0\{\sup_{t\in[0,\tau]} |\widehat{\phi}_k^{\mathrm{RMST},h}(t, a; \widehat{\eta}_k^{\Lambda,h}) - \phi_\infty^{\mathrm{RMST},h}(t, a; \eta_\infty^{\Lambda,h})|\}^2]^{1/2}
$$

$$
\leq C_1^{\mathrm{RMST},h} \widetilde{\mathcal{B}}_{1,k}^{\mathrm{RMST},h}(\widehat{\eta}_k^{\Lambda,h}, \eta_\infty^{\Lambda,h}) + \sum_{\iota=2}^{6} C_\iota^{\mathrm{RMST},h} \mathcal{B}_{\iota,k}^{\mathrm{RMST},h}(\widehat{\eta}_k^{\Lambda,h}, \eta_\infty^{\Lambda,h})
$$

$$
[P_0\{\widehat{\phi}_{j,k}^{\mathrm{RMTL},h}(\tau, a; \widehat{\eta}_{j,k}^{\Lambda,h}) - \phi_{j,\infty}^{\mathrm{RMTL},h}(\tau, a; \eta_\infty^{\Lambda,h})\}^2]^{1/2} \leq \sum_{\iota=1}^{9} C_{j,\iota}^{\mathrm{RMTL},h} \mathcal{B}_{j,\iota,k}^{\mathrm{RMTL},h}(\widehat{\eta}_{j,k}^{\Lambda,h}, \eta_{j,\infty}^{\Lambda,h})
$$

$$
[P_0\{\sup_{t\in[0,\tau]} |\widehat{\phi}_{j,k}^{\mathrm{RMTL},h}(t, a; \widehat{\eta}_{j,k}^{\Lambda,h}) - \phi_{j,\infty}^{\mathrm{RMTL},h}(t, a; \eta_\infty^{\Lambda,h})|\}^2]^{1/2}
$$

$$
\leq C_{j,1}^{\mathrm{RMTL},h} \widetilde{\mathcal{B}}_{j,1,k}^{\mathrm{RMTL},h}(\widehat{\eta}_{j,k}^{\Lambda,h}, \eta_{j,\infty}^{\Lambda,h}) + \sum_{\iota=2}^{9} C_{j,\iota}^{\mathrm{RMTL},h} \mathcal{B}_{j,\iota,k}^{\mathrm{RMTL},h}(\widehat{\eta}_{j,k}^{\Lambda,h}, \eta_{j,\infty}^{\Lambda,h})
$$

*where*

$$
\{\mathcal{B}_{1,k}^{\mathrm{RMST},h}(\widehat{\eta}_k^{\Lambda,h}, \eta_\infty^{\Lambda,h})\}^2 = E_0\left[\sup_{t\in[0,\tau]} \left|\frac{\widehat{\mathrm{RMST}}_k(\tau \mid a, X)}{\widehat{S}_k(t \mid a, X)} - \frac{\mathrm{RMST}_\infty(\tau \mid a, X)}{S_\infty(t \mid a, X)}\right|\right]^2
$$

$$
\{\widetilde{\mathcal{B}}_{1,k}^{\mathrm{RMST},h}(\widehat{\eta}_k^{\Lambda,h}, \eta_\infty^{\Lambda,h})\}^2 = E_0\left[\sup_{t\in[0,\tau]} \sup_{u\in[0,t]} \left|\frac{\widehat{\mathrm{RMST}}_k(t \mid a, X)}{\widehat{S}_k(u \mid a, X)} - \frac{\mathrm{RMST}_\infty(t \mid a, X)}{S_\infty(u \mid a, X)}\right|\right]^2
$$

$$
\{\mathcal{B}_{2,k}^{\mathrm{RMST},h}(\widehat{\eta}_k^{\Lambda,h}, \eta_\infty^{\Lambda,h})\}^2 = E_0|[P_{n,k}\{\widehat{h}_k(X)\}]^{-1}\widehat{h}_k(X) - [E_0\{h_\infty(X)\}]^{-1}h_\infty(X)|^2
$$

$$
\{\mathcal{B}_{3,k}^{\mathrm{RMST},h}(\widehat{\eta}_k^{\Lambda,h}, \eta_\infty^{\Lambda,h})\}^2 = ([P_{n,k}\{\widehat{w}_k^h(a, X)\}]^{-1} - [E_0\{h_\infty(X)\}]^{-1})^2
$$

$$
\{\mathcal{B}_{4,k}^{\mathrm{RMST},h}(\widehat{\eta}_k^{\Lambda,h}, \eta_\infty^{\Lambda,h})\}^2 = E_0\left|\widehat{w}_k^h(a, X) - w_\infty^h(a, X)\right|^2
$$

$$
\{\mathcal{B}_{5,k}^{\mathrm{RMST},h}(\widehat{\eta}_k^{\Lambda,h}, \eta_\infty^{\Lambda,h})\}^2 = E_0\left[\sup_{t\in[0,\tau]} \left|\frac{1}{\widehat{G}_k(t- \mid a, X)} - \frac{1}{G_\infty(t- \mid a, X)}\right|\right]^2
$$

$$
\{\mathcal{B}_{6,k}^{\mathrm{RMST},h}(\widehat{\eta}_k^{\Lambda,h}, \eta_\infty^{\Lambda,h})\}^2 = E_0\left[\sup_{t\in[0,\tau]} \left|\frac{\widehat{\mathrm{RMST}}_k(t \mid a, X)}{\widehat{S}_k(t \mid a, X)} - \frac{\mathrm{RMST}_\infty(t \mid a, X)}{S_\infty(t \mid a, X)}\right|\right]^2
$$

*and*

$$\{\mathcal{B}_{j,1,k}^{\mathrm{RMTL},h}(\widehat{\eta}_{j,k}^{\Lambda,h},\eta_{j,\infty}^{\Lambda,h})\}^2 = \max_k E_0\left[\sup_{t\in[0,\tau]}\left|\frac{\widehat{\mathrm{RMTL}}_{j,k}(\tau\mid a,X)}{\widehat{S}_k(t\mid a,X)} - \frac{\mathrm{RMTL}_{j,\infty}(\tau\mid a,X)}{S_\infty(t\mid a,X)}\right|\right]^2 \xrightarrow{P} 0$$

$$\{\mathcal{B}_{j,2,k}^{\mathrm{RMTL},h}(\widehat{\eta}_{j,k}^{\Lambda,h},\eta_{j,\infty}^{\Lambda,h})\}^2 = \max_k E_0|[P_{n,k}\{\widehat{h}_k(X)\}]^{-1}\widehat{h}_k(X) - [E_0\{h_\infty(X)\}]^{-1}h_\infty(X))|^2 \xrightarrow{P} 0$$

$$\{\mathcal{B}_{j,3,k}^{\mathrm{RMTL},h}(\widehat{\eta}_{j,k}^{\Lambda,h},\eta_{j,\infty}^{\Lambda,h})\}^2 = \max_k([P_{n,k}\{\widehat{w}_k^h(a,X)\}]^{-1} - [E_0\{h_\infty(X)\}]^{-1})^2 \xrightarrow{P} 0$$

$$\{\mathcal{B}_{j,4,k}^{\mathrm{RMTL},h}(\widehat{\eta}_{j,k}^{\Lambda,h},\eta_{j,\infty}^{\Lambda,h})\}^2 = \max_k E_0\left|\widehat{w}_k^h(a,X) - w_\infty^h(a,X)\right|^2 \xrightarrow{P} 0$$

$$\{\mathcal{B}_{j,5,k}^{\mathrm{RMTL},h}(\widehat{\eta}_{j,k}^{\Lambda,h},\eta_{j,\infty}^{\Lambda,h})\}^2 = \max_k E_0\left[\sup_{t\in[0,\tau]}\left|\frac{1}{\widehat{G}_k(t-\mid a,X)} - \frac{1}{G_\infty(t-\mid a,X)}\right|\right]^2 \xrightarrow{P} 0$$

$$\{\mathcal{B}_{j,6,k}^{\mathrm{RMTL},h}(\widehat{\eta}_{j,k}^{\Lambda,h},\eta_{j,\infty}^{\Lambda,h})\}^2 = \max_k E_0\left[\sup_{t\in[0,\tau]}\left|\widehat{\Lambda}_{j,k}(t\mid a,X) - \Lambda_{j,\infty}(t\mid a,X)\right|\right]^2 \xrightarrow{P} 0$$

$$\{\mathcal{B}_{j,7,k}^{\mathrm{RMTL},h}(\widehat{\eta}_{j,k}^{\Lambda,h},\eta_{j,\infty}^{\Lambda,h})\}^2 = \max_k E_0\left[\sup_{t\in[0,\tau]}\left|\frac{\widehat{\mathrm{RMTL}}_{j,k}(t\mid a,X)}{\widehat{S}_k(t\mid a,X)} - \frac{\mathrm{RMTL}_{j,\infty}(t\mid a,X)}{S_\infty(t\mid a,X)}\right|\right]^2 \xrightarrow{P} 0$$

$$\{\mathcal{B}_{j,8,k}^{\mathrm{RMTL},h}(\widehat{\eta}_{j,k}^{\Lambda,h},\eta_{j,\infty}^{\Lambda,h})\}^2 = \max_k E_0\left[\sup_{t\in[0,\tau]}\left|\frac{\widehat{F}_{j,k}(t\mid a,X)}{\widehat{S}_k(t\mid a,X)} - \frac{F_{j,\infty}(t\mid a,X)}{S_\infty(t\mid a,X)}\right|\right]^2 \xrightarrow{P} 0$$

$$\{\mathcal{B}_{j,9,k}^{\mathrm{RMTL},h}(\widehat{\eta}_{j,k}^{\Lambda,h},\eta_{j,\infty}^{\Lambda,h})\}^2 = \max_k E_0\left[\sup_{t\in[0,\tau]}\left|\widehat{\Lambda}_k(t\mid a,X) - \Lambda_\infty(t\mid a,X)\right|\right]^2 \xrightarrow{P} 0$$

*Proof of Lemma 3:* We expand $\widehat{\phi}_k^{\mathrm{RMST},h}(\tau,a;\widehat{\eta}_k^{\Lambda,h}) - \phi_\infty^{\mathrm{RMST},h}(\tau,a;\eta_\infty^{\Lambda,h}) = \sum_{\iota=1}^{14}\mathcal{Q}_{\iota,k}^{\mathrm{RMST},h}(\tau,a;\widehat{\eta}_k^{\Lambda,h},\eta_\infty^{\Lambda,h})$ *into terms which can be squared-bounded in expectation, where*

$$\mathcal{Q}_{1,k}^{\mathrm{RMST},h}(\tau,a;\widehat{\eta}_k^{\Lambda,h},\eta_\infty^{\Lambda,h}) = [P_{n,k}\{\widehat{h}_k(X)\}]^{-1}\widehat{h}_k(X)\{\widehat{\mathrm{RMST}}_k(\tau\mid a,X) - \mathrm{RMST}_\infty(\tau\mid a,X)\}$$

$$\mathcal{Q}_{2,k}^{\mathrm{RMST},h}(\tau,a;\widehat{\eta}_k^{\Lambda,h},\eta_\infty^{\Lambda,h}) = ([P_{n,k}\{\widehat{h}_k(X)\}]^{-1}\widehat{h}_k(X) - [E_0\{h_\infty(X)\}]^{-1}h_\infty(X))\mathrm{RMST}_\infty(\tau\mid a,X)$$

$$\mathcal{Q}_{3,k}^{\mathrm{RMST},h}(\tau,a;\widehat{\eta}_k^{\Lambda,h},\eta_\infty^{\Lambda,h}) = -([P_{n,k}\{\widehat{w}_k^h(a,X)\}]^{-1} - [E_0\{h_\infty(X)\}]^{-1})w_\infty^h(a,X)\mathrm{RMST}_\infty(\tau\mid a,X)$$
$$\times \int_0^{\tau\wedge\widetilde{T}}\frac{\mathrm{d}M_\infty(t\mid a,X)}{S_\infty(t\mid a,X)G_\infty(t-\mid a,X)}$$

$$\mathcal{Q}_{4,k}^{\mathrm{RMST},h}(\tau,a;\widehat{\eta}_k^{\Lambda,h},\eta_\infty^{\Lambda,h}) = -[P_{n,k}\{\widehat{w}_k^h(a,X)\}]^{-1}\{\widehat{w}_k^h(a,X) - w_\infty^h(a,X)\}\mathrm{RMST}_\infty(\tau\mid a,X)$$
$$\times \int_0^{\tau\wedge\widetilde{T}}\frac{\mathrm{d}M_\infty(t\mid a,X)}{S_\infty(t\mid a,X)G_\infty(t-\mid a,X)}$$

$$\mathcal{Q}_{5,k}^{\mathrm{RMST},h}(\tau,a;\widehat{\eta}_k^{\Lambda,h},\eta_\infty^{\Lambda,h}) = -[P_{n,k}\{\widehat{w}_k^h(a,X)\}]^{-1}\frac{\widehat{w}_k^h(a,X)I(\widetilde{T}\le\tau,\Delta=1)}{G_\infty(\widetilde{T}-\mid a,X)}$$
$$\times \left\{\frac{\widehat{\mathrm{RMST}}_k(\tau\mid a,X)}{\widehat{S}_k(\widetilde{T}\mid a,X)} - \frac{\mathrm{RMST}_\infty(\tau\mid a,X)}{S_\infty(\widetilde{T}\mid a,X)}\right\}$$

$$\mathcal{Q}_{6,k}^{\mathrm{RMST},h}(\tau,a;\widehat{\eta}_k^{\Lambda,h},\eta_\infty^{\Lambda,h}) = -[P_{n,k}\{\widehat{w}_k^h(a,X)\}]^{-1}\frac{\widehat{w}_k^h(a,X)I(\widetilde{T}\le\tau,\Delta=1)\widehat{\mathrm{RMST}}_k(\tau\mid a,X)}{\widehat{S}_k(\widetilde{T}\mid a,X)}$$
$$\times \left\{\frac{1}{\widehat{G}_k(\widetilde{T}-\mid a,X)} - \frac{1}{G_\infty(\widetilde{T}-\mid a,X)}\right\}$$

$$\mathcal{Q}_{7,k}^{\mathrm{RMST},h}(\tau,a;\widehat{\eta}_k^{\Lambda,h},\eta_\infty^{\Lambda,h}) = [P_{n,k}\{\widehat{w}_k^h(a,X)\}]^{-1}\widehat{w}_k^h(a,X)\mathrm{RMST}_\infty(\tau\mid a,X)\int_0^\tau\frac{\mathrm{d}\Lambda_\infty(t\mid a,X)}{S_\infty(t\mid a,X)}$$
$$\times \left\{\frac{1}{\widehat{G}_k(t-\mid a,X)} - \frac{1}{G_\infty(t-\mid a,X)}\right\}$$

$$\mathcal{Q}_{8,k}^{\mathrm{RMST},h}(\tau,a;\widehat{\eta}_k^{\Lambda,h},\eta_\infty^{\Lambda,h}) = [P_{n,k}\{\widehat{w}_k^h(a,X)\}]^{-1}\widehat{w}_k^h(a,X)\left\{\widehat{\mathrm{RMST}}_k(\tau\mid a,X)\right.$$

$$\left.\times\int_0^\tau\frac{\mathrm{d}\widehat{\Lambda}_k(t\mid a,X)}{\widehat{S}_k(t\mid a,X)\widehat{G}_k(t-\mid a,X)} - \mathrm{RMST}_\infty(\tau\mid a,X)\int_0^\tau\frac{\mathrm{d}\Lambda_\infty(t\mid a,X)}{S_\infty(t\mid a,X)\widehat{G}_k(t-\mid a,X)}\right\}$$

$$\mathcal{Q}_{9,k}^{\mathrm{RMST},h}(\tau,a;\widehat{\eta}_k^{\Lambda,h},\eta_\infty^{\Lambda,h}) = ([P_{n,k}\{\widehat{w}_k^h(a,X)\}]^{-1} - [E\{h_\infty(X)\}]^{-1})w_\infty^h(a,X)$$

$$\times\int_0^{\tau\wedge\widetilde{T}}\frac{\mathrm{RMST}_\infty(t\mid a,X)\mathrm{d}M_\infty(t\mid a,X)}{S_\infty(t\mid a,X)G_\infty(t-\mid a,X)}$$

$$\mathcal{Q}_{10,k}^{\mathrm{RMST},h}(\tau,a;\widehat{\eta}_k^{\Lambda,h},\eta_\infty^{\Lambda,h}) = [P_{n,k}\{\widehat{w}_k^h(a,X)\}]^{-1}\{\widehat{w}_k^h(a,X) - w_\infty^h(a,X)\}\int_0^{\tau\wedge\widetilde{T}}\frac{\mathrm{RMST}_\infty(t\mid a,X)\mathrm{d}M_\infty(t\mid a,X)}{S_\infty(t\mid a,X)G_\infty(t-\mid a,X)}$$

$$\mathcal{Q}_{11,k}^{\mathrm{RMST},h}(\tau,a;\widehat{\eta}_k^{\Lambda,h},\eta_\infty^{\Lambda,h}) = [P_{n,k}\{\widehat{w}_k^h(a,X)\}]^{-1}\frac{\widehat{w}_k^h(a,X)I(\widetilde{T}\le\tau,\Delta=1)}{G_\infty(\widetilde{T}-\mid a,X)}$$

$$\times\left\{\frac{\widehat{\mathrm{RMST}}_k(\widetilde{T}\mid a,X)}{\widehat{S}_k(\widetilde{T}\mid a,X)} - \frac{\mathrm{RMST}_\infty(\widetilde{T}\mid a,X)}{S_\infty(\widetilde{T}\mid a,X)}\right\}$$

$$\mathcal{Q}_{12,k}^{\mathrm{RMST},h}(\tau,a;\widehat{\eta}_k^{\Lambda,h},\eta_\infty^{\Lambda,h}) = [P_{n,k}\{\widehat{w}_k^h(a,X)\}]^{-1}\frac{\widehat{w}_k^h(a,X)I(\widetilde{T}\le\tau,\Delta=1)\widehat{\mathrm{RMST}}_k(\widetilde{T}\mid a,X)}{\widehat{S}_k(\widetilde{T}\mid a,X)}$$

$$\times\left\{\frac{1}{\widehat{G}_k(\widetilde{T}-\mid a,X)} - \frac{1}{G_\infty(\widetilde{T}-\mid a,X)}\right\}$$

$$\mathcal{Q}_{13,k}^{\mathrm{RMST},h}(\tau,a;\widehat{\eta}_k^{\Lambda,h},\eta_\infty^{\Lambda,h}) = -[P_{n,k}\{\widehat{w}_k^h(a,X)\}]^{-1}\widehat{w}_k^h(a,X)\int_0^\tau\frac{\mathrm{RMST}_\infty(t\mid a,X)\mathrm{d}\Lambda_\infty(t\mid a,X)}{S_\infty(t\mid a,X)}$$

$$\times\left\{\frac{1}{\widehat{G}_k(t-\mid a,X)} - \frac{1}{G_\infty(t-\mid a,X)}\right\}$$

$$\mathcal{Q}_{14,k}^{\mathrm{RMST},h}(\tau,a;\widehat{\eta}_k^{\Lambda,h},\eta_\infty^{\Lambda,h}) = -[P_{n,k}\{\widehat{w}_k^h(a,X)\}]^{-1}\widehat{w}_k^h(a,X)\left\{\int_0^\tau\frac{\widehat{\mathrm{RMST}}_k(t\mid a,X)\mathrm{d}\widehat{\Lambda}_k(t\mid a,X)}{\widehat{S}_k(t\mid a,X)\widehat{G}_k(t-\mid a,X)}\right.$$

$$\left.-\int_0^\tau\frac{\mathrm{RMST}_\infty(t\mid a,X)\mathrm{d}\Lambda_\infty(t\mid a,X)}{S_\infty(t\mid a,X)\widehat{G}_k(t-\mid a,X)}\right\}$$

Following the triangle inequality,

$$P_0\{\widehat{\phi}^{\mathrm{RMST},h}(\tau,a;\widehat{\eta}_k^{\Lambda,h},\eta_\infty^{\Lambda,h}) - \phi_\infty^{\mathrm{RMST},h}(\tau,a;\widehat{\eta}_k^{\Lambda,h},\eta_\infty^{\Lambda,h})\}^2 \le \left(\sum_{\iota=1}^{14}[P_0\{\mathcal{Q}_{\iota,k}^{\mathrm{RMST},h}(\tau,a;\widehat{\eta}_k^{\Lambda,h},\eta_\infty^{\Lambda,h})\}^2]^{1/2}\right)^2$$

Thus we need to bound $P_0\{\mathcal{Q}_{\iota,k}^{\mathrm{RMST},h}(\tau,a;\widehat{\eta}_k^{\Lambda,h},\eta_\infty^{\Lambda,h})\}^2, \forall\iota$.

$$P_0\{\mathcal{Q}_{1,k}^{\mathrm{RMST},h}(\tau,a;\widehat{\eta}_k^{\Lambda,h},\eta_\infty^{\Lambda,h})\}^2 = E_0\left|[P_{n,k}\{\widehat{h}_k(X)\}]^{-1}\widehat{h}_k(X)\{\widehat{\mathrm{RMST}}_k(\tau\mid a,X) - \mathrm{RMST}_\infty(\tau\mid a,X)\}\right|^2$$

$$\le \epsilon^2 h_{\max}^2 E_0\left|\frac{\widehat{\mathrm{RMST}}_k(\tau\mid a,X)}{\widehat{S}_k(\tau\mid a,X)} - \frac{\mathrm{RMST}_\infty(\tau\mid a,X)}{S_\infty(\tau\mid a,X)}\right|^2$$

$$\le \epsilon^2 h_{\max}^2 E_0\left[\sup_{t\in[0,\tau]}\left|\frac{\widehat{\mathrm{RMST}}_k(\tau\mid a,X)}{\widehat{S}_k(t\mid a,X)} - \frac{\mathrm{RMST}_\infty(\tau\mid a,X)}{S_\infty(t\mid a,X)}\right|\right]^2$$

$$P_0\{\mathcal{Q}_{2,k}^{\mathrm{RMST},h}(\tau,a;\widehat{\eta}_k^{\Lambda,h},\eta_\infty^{\Lambda,h})\}^2 = E_0\left|([P_{n,k}\{\widehat{h}_k(X)\}]^{-1}\widehat{h}_k(X) - [E_0\{h_\infty(X)\}]^{-1}h_\infty(X))\mathrm{RMST}_\infty(\tau\mid a,X)\right|^2$$

$$\le \tau^2 E_0\left|([P_{n,k}\{\widehat{h}_k(X)\}]^{-1}\widehat{h}_k(X) - [E_0\{h_\infty(X)\}]^{-1}h_\infty(X))\right|^2$$

The backwards equation from Theorem 5 of Gill & Johansen (1990) suggests that $\int_0^t \frac{S(t|a,X)}{S(u|,a,X)} d\Lambda(u \mid a, X) = 1 - S(t \mid a, X)$ and

$$E_0 \left| \int_0^{\tau \wedge \widetilde{T}} \frac{dM_\infty(t \mid a, X)}{S_\infty(t \mid a, X) G_\infty(t- \mid a, X)} \right|^2 = E_0 \left| \frac{I(\widetilde{T} \leq \tau, \Delta = 1)}{S_\infty(\widetilde{T} \mid a, X) G_\infty(\widetilde{T}- \mid a, X)} - \int_0^\tau \frac{d\Lambda_\infty(t \mid a, X)}{S_\infty(t \mid a, X) G_\infty(t- \mid a, X)} \right|^2$$

$$\leq E_0 \left| \epsilon^2 + \epsilon \int_0^\tau d\left\{ \frac{1}{S_\infty(t \mid a, X)} \right\} \right|^2 \leq E_0 \left| \epsilon^2 + \epsilon(\epsilon - 1) \right|^2 \leq 4\epsilon^4$$

$$P_0\{\mathcal{Q}_{3,k}^{\mathrm{RMST},h}(\tau, a; \widehat{\eta}_k^{\Lambda,h}, \eta_\infty^{\Lambda,h})\}^2 = E_0 \left| ([P_{n,k}\{\widehat{w}_k^h(a, X)\}]^{-1} - [E_0\{h_\infty(X)\}]^{-1}) \right.$$

$$\left. \frac{h_\infty(X) I(A = a) \mathrm{RMST}_\infty(\tau \mid a, X)}{\pi_\infty(a \mid X)} \int_0^{\tau \wedge \widetilde{T}} \frac{dM_\infty(t \mid a, X)}{S_\infty(t \mid a, X) G_\infty(t- \mid a, X)} \right|^2$$

$$\leq ([P_{n,k}\{\widehat{w}_k^h(a, X)\}]^{-1} - [E_0\{h_\infty(X)\}]^{-1})^2 \epsilon^2 h_{\max}^2 \tau^2 \times 4\epsilon^4$$

Since

$$P_{n,k}\{\widehat{w}_k^h(a, X)\} = P_{n,k}\left\{ \frac{\widehat{h}_k(X) I(A = a)}{\widehat{\pi}_k(a \mid X)} \right\} \geq P_{n,k}\{\widehat{h}_k(X) I(A = a)\} \geq P_{n,k}\{\widehat{h}_k(X)\} \geq h_{\min}$$

We have

$$P_0\{\mathcal{Q}_{4,k}^{\mathrm{RMST},h}(\tau, a; \widehat{\eta}_k^{\Lambda,h}, \eta_\infty^{\Lambda,h})\}^2 = E_0 \left| [P_{n,k}\{\widehat{w}_k^h(a, X)\}]^{-1}\{\widehat{w}_k^h(a, X) - w_\infty^h(a, X)\} \mathrm{RMST}_\infty(\tau \mid a, X) \right.$$

$$\left. \times \int_0^{\tau \wedge \widetilde{T}} \frac{dM_\infty(t \mid a, X)}{S_\infty(t \mid a, X) G_\infty(t- \mid a, X)} \right|^2 \leq h_{\min}^{-2} \tau^2 \times 4\epsilon^4 E_0 \left| \widehat{w}_k^h(a, X) - w_\infty^h(a, X) \right|^2$$

$$P_0\{\mathcal{Q}_{5,k}^{\mathrm{RMST},h}(\tau, a; \widehat{\eta}_k^{\Lambda,h}, \eta_\infty^{\Lambda,h})\}^2 = E_0 \left| [P_{n,k}\{\widehat{w}_k^h(a, X)\}]^{-1} \frac{\widehat{w}_k^h(a, X) I(\widetilde{T} \leq \tau, \Delta = 1)}{G_\infty(\widetilde{T}- \mid a, X)} \right.$$

$$\left. \times \left\{ \frac{\widehat{\mathrm{RMST}}_k(\tau \mid a, X)}{\widehat{S}_k(\widetilde{T} \mid a, X)} - \frac{\mathrm{RMST}_\infty(\tau \mid a, X)}{S_\infty(\widetilde{T} \mid a, X)} \right\} \right|^2$$

$$\leq h_{\min}^{-2} h_{\max}^2 \epsilon^4 E_0 \left[ \sup_{t \in [0,\tau]} \left| \frac{\widehat{\mathrm{RMST}}_k(\tau \mid a, X)}{\widehat{S}_k(t \mid a, X)} - \frac{\mathrm{RMST}_\infty(\tau \mid a, X)}{S_\infty(t \mid a, X)} \right| \right]^2$$

$$P_0\{\mathcal{Q}_{6,k}^{\mathrm{RMST},h}(\tau, a; \widehat{\eta}_k^{\Lambda,h}, \eta_\infty^{\Lambda,h})\}^2 = E_0 \left| [P_{n,k}\{\widehat{w}_k^h(a, X)\}]^{-1} \frac{\widehat{w}_k^h(a, X) I(\widetilde{T} \leq \tau, \Delta = 1) \widehat{\mathrm{RMST}}_k(\tau \mid a, X)}{\widehat{S}_k(\widetilde{T} \mid a, X)} \right.$$

$$\left. \times \left\{ \frac{1}{\widehat{G}_k(\widetilde{T}- \mid a, X)} - \frac{1}{G_\infty(\widetilde{T}- \mid a, X)} \right\} \right|^2$$

$$\leq h_{\min}^{-2} h_{\max}^2 \tau^2 \epsilon^4 E_0 \left[ \sup_{t \in [0,\tau]} \left| \frac{1}{\widehat{G}_k(t- \mid a, X)} - \frac{1}{G_\infty(t- \mid a, X)} \right| \right]^2$$

$$P_0\{\mathcal{Q}_{7,k}^{\mathrm{RMST},h}(\tau, a; \widehat{\eta}_k^{\Lambda,h}, \eta_\infty^{\Lambda,h})\}^2 = E_0 \left| [P_{n,k}\{\widehat{w}_k^h(a, X)\}]^{-1} \widehat{w}_k^h(a, X) \mathrm{RMST}_\infty(\tau \mid a, X) \int_0^\tau \frac{d\Lambda_\infty(t \mid a, X)}{S_\infty(t \mid a, X)} \right.$$

$$\left. \times \left\{ \frac{1}{\widehat{G}_k(t- \mid a, X)} - \frac{1}{G_\infty(t- \mid a, X)} \right\} \right|^2$$

$$\leq h_{\min}^{-2} h_{\max}^2 \tau^2 \epsilon^2 E_0 \left[ \sup_{t \in [0,\tau]} \left| \frac{1}{\widehat{G}_k(t- \mid a, X)} - \frac{1}{G_\infty(t- \mid a, X)} \right| \int_0^\tau \frac{d\Lambda_\infty(t \mid a, X)}{S_\infty(t \mid a, X)} \right]^2$$

$$\leq h_{\min}^{-2} h_{\max}^2 \tau^2 \epsilon^2 (\epsilon - 1)^2 E_0 \left[ \sup_{t \in [0,\tau]} \left| \frac{1}{\widehat{G}_k(t- \mid a, X)} - \frac{1}{G_\infty(t- \mid a, X)} \right| \right]^2$$

As

$$
E_0 \left| \widehat{\mathrm{RMST}}_k(\tau \mid a, X) \int_0^\tau \frac{\mathrm{d}\widehat{\Lambda}_k(t \mid a, X)}{\widehat{S}_k(t \mid a, X)\widehat{G}_k(t- \mid a, X)} - \mathrm{RMST}_\infty(\tau \mid a, X) \int_0^\tau \frac{\mathrm{d}\Lambda_\infty(t \mid a, X)}{S_\infty(t \mid a, X)\widehat{G}_k(t- \mid a, X)} \right|^2
$$

$$
= E_0 \left| \widehat{\mathrm{RMST}}_k(\tau \mid a, X) \left[ \left. \frac{1}{\widehat{S}_k(t \mid a, X)\widehat{G}_k(t- \mid a, X)} \right|_0^\tau - \int_0^\tau \frac{1}{\widehat{S}_k(t \mid a, X)} \mathrm{d}\left\{ \frac{1}{\widehat{G}_k(t- \mid a, X)} \right\} \right] \right.
$$

$$
\left. - \mathrm{RMST}_\infty(\tau \mid a, X) \left[ \left. \frac{1}{S_\infty(t \mid a, X)\widehat{G}_k(t- \mid a, X)} \right|_0^\tau - \int_0^\tau \frac{1}{S_\infty(t \mid a, X)} \mathrm{d}\left\{ \frac{1}{\widehat{G}_k(t- \mid a, X)} \right\} \right] \right|^2
$$

$$
= E_0 \left| \left\{ \frac{\widehat{\mathrm{RMST}}_k(\tau \mid a, X)}{\widehat{S}_k(\tau \mid a, X)} - \frac{\mathrm{RMST}_\infty(\tau \mid a, X)}{S_\infty(\tau \mid a, X)} \right\} \frac{1}{\widehat{G}_k(\tau- \mid a, X)} \right.
$$

$$
\left. - \int_0^\tau \left\{ \frac{\widehat{\mathrm{RMST}}_k(\tau \mid a, X)}{\widehat{S}_k(t \mid a, X)} - \frac{\mathrm{RMST}_\infty(\tau \mid a, X)}{S_\infty(t \mid a, X)} \right\} \mathrm{d}\left\{ \frac{1}{\widehat{G}_k(t- \mid a, X)} \right\} \right|^2
$$

$$
\leq \epsilon^2 E_0 \left\{ \sup_{t \in [0,\tau]} \left| \frac{\widehat{\mathrm{RMST}}_k(\tau \mid a, X)}{\widehat{S}_k(t \mid a, X)} - \frac{\mathrm{RMST}_\infty(\tau \mid a, X)}{S_\infty(t \mid a, X)} \right| \right\}^2 + E_0 \left\{ \left. \frac{1}{\widehat{G}_k(t- \mid a, X)} \right|_0^\tau \right.
$$

$$
\times \sup_{t \in [0,\tau]} \left| \frac{\widehat{\mathrm{RMST}}_k(\tau \mid a, X)}{\widehat{S}_k(t \mid a, X)} - \frac{\mathrm{RMST}_\infty(\tau \mid a, X)}{S_\infty(t \mid a, X)} \right| \right\}^2
$$

$$
\leq \{\epsilon^2 + (\epsilon - 1)^2\} E_0 \left\{ \sup_{t \in [0,\tau]} \left| \frac{\widehat{\mathrm{RMST}}_k(\tau \mid a, X)}{\widehat{S}_k(t \mid a, X)} - \frac{\mathrm{RMST}_\infty(\tau \mid a, X)}{S_\infty(t \mid a, X)} \right| \right\}^2
$$

We have

$$
P_0\{\mathcal{Q}_{8,k}^{\mathrm{RMST},h}(\tau, a; \widehat{\eta}_k^{\Lambda,h}, \eta_\infty^{\Lambda,h})\}^2
$$

$$
= E_0 \left| [P_{n,k}\{\widehat{w}_k^h(a, X)\}]^{-1} \widehat{w}_k^h(a, X) \left\{ \widehat{\mathrm{RMST}}_k(\tau \mid a, X) \right. \right.
$$

$$
\left. \left. \times \int_0^\tau \frac{\mathrm{d}\widehat{\Lambda}_k(t \mid a, X)}{\widehat{S}_k(t \mid a, X)\widehat{G}_k(t- \mid a, X)} - \mathrm{RMST}_\infty(\tau \mid a, X) \int_0^\tau \frac{\mathrm{d}\Lambda_\infty(t \mid a, X)}{S_\infty(t \mid a, X)\widehat{G}_k(t- \mid a, X)} \right\} \right|^2
$$

$$
\leq h_{\min}^{-2} h_{\max}^2 \epsilon^2 \{\epsilon^2 + (\epsilon - 1)^2\} E_0 \left\{ \sup_{t \in [0,\tau]} \left| \frac{\widehat{\mathrm{RMST}}_k(\tau \mid a, X)}{\widehat{S}_k(t \mid a, X)} - \frac{\mathrm{RMST}_\infty(\tau \mid a, X)}{S_\infty(t \mid a, X)} \right| \right\}^2
$$

Similar to previous terms,

$$
P_0\{\mathcal{Q}_{9,k}^{\mathrm{RMST},h}(\tau, a; \widehat{\eta}_k^{\Lambda,h}, \eta_\infty^{\Lambda,h})\}^2 \leq ([P_{n,k}\{\widehat{w}_k^h(a, X)\}]^{-1} - [E_0\{h_\infty(X)\}]^{-1})^2 \epsilon^2 h_{\max}^2 \tau^2 \times 4\epsilon^4
$$

$$
P_0\{\mathcal{Q}_{10,k}^{\mathrm{RMST},h}(\tau, a; \widehat{\eta}_k^{\Lambda,h}, \eta_\infty^{\Lambda,h})\}^2 \leq h_{\min}^{-2} \tau^2 \times 4\epsilon^4 E_0 \left| \widehat{w}_k^h(a, X) - w_\infty^h(a, X) \right|^2
$$

$$
P_0\{\mathcal{Q}_{11,k}^{\mathrm{RMST},h}(\tau, a; \widehat{\eta}_k^{\Lambda,h}, \eta_\infty^{\Lambda,h})\}^2 \leq h_{\min}^{-2} h_{\max}^2 \epsilon^4 E_0 \left[ \sup_{t \in [0,\tau]} \left| \frac{\widehat{\mathrm{RMST}}_k(t \mid a, X)}{\widehat{S}_k(t \mid a, X)} - \frac{\mathrm{RMST}_\infty(t \mid a, X)}{S_\infty(t \mid a, X)} \right| \right]^2
$$

$$
P_0\{\mathcal{Q}_{12,k}^{\mathrm{RMST},h}(\tau, a; \widehat{\eta}_k^{\Lambda,h}, \eta_\infty^{\Lambda,h})\}^2 \leq h_{\min}^{-2} h_{\max}^2 \tau^2 \epsilon^4 E_0 \left[ \sup_{t \in [0,\tau]} \left| \frac{1}{\widehat{G}_k(t- \mid a, X)} - \frac{1}{G_\infty(t- \mid a, X)} \right| \right]^2
$$

$$
P_0\{\mathcal{Q}_{13,k}^{\mathrm{RMST},h}(\tau, a; \widehat{\eta}_k^{\Lambda,h}, \eta_\infty^{\Lambda,h})\}^2 \leq h_{\min}^{-2} h_{\max}^2 \tau^2 \epsilon^2 (\epsilon - 1)^2 E_0 \left[ \sup_{t \in [0,\tau]} \left| \frac{1}{\widehat{G}_k(t- \mid a, X)} - \frac{1}{G_\infty(t- \mid a, X)} \right| \right]^2
$$

As

$$
E_0 \left| \int_0^\tau \frac{\widehat{\mathrm{RMST}}_k(t \mid a, X) \mathrm{d}\widehat{\Lambda}_k(t \mid a, X)}{\widehat{S}_k(t \mid a, X)\widehat{G}_k(t- \mid a, X)} - \int_0^\tau \frac{\mathrm{RMST}_\infty(t \mid a, X)\mathrm{d}\Lambda_\infty(t \mid a, X)}{S_\infty(t \mid a, X)\widehat{G}_k(t- \mid a, X)} \right|^2
$$

$$
= E_0 \left| \frac{\widehat{\mathrm{RMST}}_k(t \mid a, X)}{\widehat{S}_k(t \mid a, X)\widehat{G}_k(t- \mid a, X)} \right|_0^\tau - \int_0^\tau \frac{1}{\widehat{S}_k(t \mid a, X)} \mathrm{d} \left\{ \frac{\widehat{\mathrm{RMST}}_k(t \mid a, X)}{\widehat{G}_k(t- \mid a, X)} \right\}
$$

$$
- \frac{\mathrm{RMST}_\infty(t \mid a, X)}{S_\infty(t \mid a, X)\widehat{G}_k(t- \mid a, X)} \bigg|_0^\tau + \int_0^\tau \frac{1}{S_\infty(t \mid a, X)} \mathrm{d} \left\{ \frac{\mathrm{RMST}_\infty(t \mid a, X)}{\widehat{G}_k(t- \mid a, X)} \right\} \bigg|^2
$$

$$
\leq \left\{ \epsilon^2 + (\epsilon - 1)^2 \right\} E_0 \left\{ \sup_{t \in [0,\tau]} \left| \frac{\widehat{\mathrm{RMST}}_k(t \mid a, X)}{\widehat{S}_k(t \mid a, X)} - \frac{\mathrm{RMST}_\infty(t \mid a, X)}{S_\infty(t \mid a, X)} \right| \right\}^2
$$

Thus

$$
P_0 \{ \mathcal{Q}_{14,k}^{\mathrm{RMST},h}(\tau, a; \widehat{\eta}_k^{\Lambda,h}, \eta_\infty^{\Lambda,h}) \}^2 \leq h_{\min}^{-2} h_{\max}^2 \epsilon^2 \{ \epsilon^2 + (\epsilon-1)^2 \} E_0 \left\{ \sup_{t \in [0,\tau]} \left| \frac{\widehat{\mathrm{RMST}}_k(t \mid a, X)}{\widehat{S}_k(t \mid a, X)} - \frac{\mathrm{RMST}_\infty(t \mid a, X)}{S_\infty(t \mid a, X)} \right| \right\}^2
$$

In the competing risks setting, we expand $\widehat{\phi}_{j,k}^{\mathrm{RMTL},h}(\tau, a; \widehat{\eta}_{j,k}^{\Lambda,h}, \eta_{j,\infty}^{\Lambda,h}) - \phi_{j,\infty}^{\mathrm{RMTL},h}(\tau, a; \widehat{\eta}_{j,k}^{\Lambda,h}, \eta_{j,\infty}^{\Lambda,h}) = \sum_{\iota=1}^{36} \mathcal{Q}_{j,\iota,k}^{\mathrm{RMTL},h}(\tau, a; \widehat{\eta}_{j,k}^{\Lambda,h}, \eta_{j,\infty}^{\Lambda,h})$ which can be squared-bounded in expectation, where

$$
\mathcal{Q}_{j,1,k}^{\mathrm{RMTL},h}(\tau, a; \widehat{\eta}_{j,k}^{\Lambda,h}, \eta_{j,\infty}^{\Lambda,h}) = [P_{n,k}\{\widehat{h}_k(X)\}]^{-1} \widehat{h}_k(X) \{ \widehat{\mathrm{RMTL}}_{j,k}(\tau \mid a, X) - \mathrm{RMTL}_{j,\infty}(\tau \mid a, X) \}
$$

$$
\mathcal{Q}_{j,2,k}^{\mathrm{RMTL},h}(\tau, a; \widehat{\eta}_{j,k}^{\Lambda,h}, \eta_{j,\infty}^{\Lambda,h}) = ([P_{n,k}\{\widehat{h}_k(X)\}]^{-1} \widehat{h}_k(X) - [E_0\{h_\infty(X)\}]^{-1} h_\infty(X)) \mathrm{RMTL}_{j,\infty}(\tau \mid a, X)
$$

$$
\mathcal{Q}_{j,3,k}^{\mathrm{RMTL},h}(\tau, a; \widehat{\eta}_{j,k}^{\Lambda,h}, \eta_{j,\infty}^{\Lambda,h}) = ([P_{n,k}\{\widehat{w}_k^h(a, X)\}]^{-1} - [E_0\{h_\infty(X)\}]^{-1}) w_\infty^h(a, X)\tau \int_0^{\tau \wedge \widetilde{T}} \frac{\mathrm{d}M_{j,\infty}(t \mid a, X)}{G_\infty(t- \mid a, X)}
$$

$$
\mathcal{Q}_{j,4,k}^{\mathrm{RMTL},h}(\tau, a; \widehat{\eta}_{j,k}^{\Lambda,h}, \eta_{j,\infty}^{\Lambda,h}) = [P_{n,k}\{\widehat{w}_k^h(a, X)\}]^{-1} \{ \widehat{w}_k^h(a, X) - w_\infty^h(a, X) \}\tau \int_0^{\tau \wedge \widetilde{T}} \frac{\mathrm{d}M_{j,\infty}(t \mid a, X)}{G_\infty(t- \mid a, X)}
$$

$$
\mathcal{Q}_{j,5,k}^{\mathrm{RMTL},h}(\tau, a; \widehat{\eta}_{j,k}^{\Lambda,h}, \eta_{j,\infty}^{\Lambda,h}) = [P_{n,k}\{\widehat{w}_k^h(a, X)\}]^{-1} \widehat{w}_k^h(a, X) I(\widetilde{T} \leq \tau, \widetilde{J} = j)\tau \left\{ \frac{1}{\widehat{G}_k(\widetilde{T}- \mid a, X)} - \frac{1}{G_\infty(\widetilde{T}- \mid a, X)} \right\}
$$

$$
\mathcal{Q}_{j,6,k}^{\mathrm{RMTL},h}(\tau, a; \widehat{\eta}_{j,k}^{\Lambda,h}, \eta_{j,\infty}^{\Lambda,h}) = -[P_{n,k}\{\widehat{w}_k^h(a, X)\}]^{-1} \widehat{w}_k^h(a, X)\tau \int_0^\tau \left\{ \frac{1}{\widehat{G}_k(t- \mid a, X)} - \frac{1}{G_\infty(t- \mid a, X)} \right\} \mathrm{d}\Lambda_{j,\infty}(t \mid a, X)
$$

$$
\mathcal{Q}_{j,7,k}^{\mathrm{RMTL},h}(\tau, a; \widehat{\eta}_{j,k}^{\Lambda,h}, \eta_{j,\infty}^{\Lambda,h}) = -[P_{n,k}\{\widehat{w}_k^h(a, X)\}]^{-1} \widehat{w}_k^h(a, X)\tau \int_0^\tau \frac{\mathrm{d}\widehat{\Lambda}_{j,k}(t \mid a, X) - \mathrm{d}\Lambda_{j,\infty}(t \mid a, X)}{\widehat{G}_k(t- \mid a, X)}
$$

$$
\mathcal{Q}_{j,8,k}^{\mathrm{RMTL},h}(\tau, a; \widehat{\eta}_{j,k}^{\Lambda,h}, \eta_{j,\infty}^{\Lambda,h}) = -([P_{n,k}\{\widehat{w}_k^h(a, X)\}]^{-1} - [E_0\{h_\infty(X)\}]^{-1}) w_\infty^h(a, X) \int_0^{\tau \wedge \widetilde{T}} \frac{t\mathrm{d}M_{j,\infty}(t \mid a, X)}{G_\infty(t- \mid a, X)}
$$

$$
\mathcal{Q}_{j,9,k}^{\mathrm{RMTL},h}(\tau, a; \widehat{\eta}_{j,k}^{\Lambda,h}, \eta_{j,\infty}^{\Lambda,h}) = -[P_{n,k}\{\widehat{w}_k^h(a, X)\}]^{-1} \{ \widehat{w}_k^h(a, X) - w_\infty^h(a, X) \} \int_0^{\tau \wedge \widetilde{T}} \frac{t\mathrm{d}M_{j,\infty}(t \mid a, X)}{G_\infty(t- \mid a, X)}
$$

$$
\mathcal{Q}_{j,10,k}^{\mathrm{RMTL},h}(\tau, a; \widehat{\eta}_{j,k}^{\Lambda,h}, \eta_{j,\infty}^{\Lambda,h}) = -[P_{n,k}\{\widehat{w}_k^h(a, X)\}]^{-1} \widehat{w}_k^h(a, X) I(\widetilde{T} \leq \tau, \widetilde{J} = j)\widetilde{T} \left\{ \frac{1}{\widehat{G}_k(\widetilde{T}- \mid a, X)} - \frac{1}{G_\infty(\widetilde{T}- \mid a, X)} \right\}
$$

$$
\mathcal{Q}_{j,11,k}^{\mathrm{RMTL},h}(\tau, a; \widehat{\eta}_{j,k}^{\Lambda,h}, \eta_{j,\infty}^{\Lambda,h}) = [P_{n,k}\{\widehat{w}_k^h(a, X)\}]^{-1} \widehat{w}_k^h(a, X) \int_0^\tau \left\{ \frac{1}{\widehat{G}_k(t- \mid a, X)} - \frac{1}{G_\infty(t- \mid a, X)} \right\} t\mathrm{d}\Lambda_{j,\infty}(t \mid a, X)
$$

$$
\mathcal{Q}_{j,12,k}^{\mathrm{RMTL},h}(\tau, a; \widehat{\eta}_{j,k}^{\Lambda,h}, \eta_{j,\infty}^{\Lambda,h}) = [P_{n,k}\{\widehat{w}_k^h(a, X)\}]^{-1} \widehat{w}_k^h(a, X) \int_0^\tau \frac{t\{\mathrm{d}\widehat{\Lambda}_{j,k}(t \mid a, X) - \mathrm{d}\Lambda_{j,\infty}(t \mid a, X)\}}{\widehat{G}_k(t- \mid a, X)}
$$

$$
\mathcal{Q}_{j,13,k}^{\mathrm{RMTL},h}(\tau, a; \widehat{\eta}_{j,k}^{\Lambda,h}, \eta_{j,\infty}^{\Lambda,h}) = -([P_{n,k}\{\widehat{w}_k^h(a, X)\}]^{-1} - [E_0\{h_\infty(X)\}]^{-1}) w_\infty^h(a, X)\mathrm{RMTL}_{j,\infty}(\tau \mid a, X)
$$

$$
\times \int_0^{\tau \wedge \widetilde{T}} \frac{\mathrm{d}M_\infty(t \mid a, X)}{S_\infty(t \mid a, X)G_\infty(t- \mid a, X)}
$$

$$\mathcal{Q}_{j,14,k}^{\mathrm{RMTL},h}(\tau,a;\widehat{\eta}_{j,k}^{\Lambda,h},\eta_{j,\infty}^{\Lambda,h}) = -[P_{n,k}\{\widehat{w}_k^h(a,X)\}]^{-1}\{\widehat{w}_k^h(a,X)-w_\infty^h(a,X)\}\mathrm{RMTL}_{j,\infty}(\tau\mid a,X)$$
$$\times \int_0^{\tau\wedge\widetilde{T}} \frac{\mathrm{d}M_\infty(t\mid a,X)}{S_\infty(t\mid a,X)G_\infty(t-\mid a,X)}$$

$$\mathcal{Q}_{j,15,k}^{\mathrm{RMTL},h}(\tau,a;\widehat{\eta}_{j,k}^{\Lambda,h},\eta_{j,\infty}^{\Lambda,h}) = -[P_{n,k}\{\widehat{w}_k^h(a,X)\}]^{-1}\frac{\widehat{w}_k^h(a,X)I(\widetilde{T}\le\tau,\Delta=1)}{G_\infty(\widetilde{T}-\mid a,X)}$$
$$\times \left\{\frac{\widehat{\mathrm{RMTL}}_{j,k}(\tau\mid a,X)}{\widehat{S}_k(\widetilde{T}\mid a,X)} - \frac{\mathrm{RMTL}_{j,\infty}(\tau\mid a,X)}{S_\infty(\widetilde{T}\mid a,X)}\right\}$$

$$\mathcal{Q}_{j,16,k}^{\mathrm{RMTL},h}(\tau,a;\widehat{\eta}_{j,k}^{\Lambda,h},\eta_{j,\infty}^{\Lambda,h}) = -[P_{n,k}\{\widehat{w}_k^h(a,X)\}]^{-1}\frac{\widehat{w}_k^h(a,X)I(\widetilde{T}\le\tau,\Delta=1)\widehat{\mathrm{RMTL}}_{j,k}(\tau\mid a,X)}{\widehat{S}_k(\widetilde{T}\mid a,X)}$$
$$\times \left\{\frac{1}{\widehat{G}_k(\widetilde{T}-\mid a,X)} - \frac{1}{G_\infty(\widetilde{T}-\mid a,X)}\right\}$$

$$\mathcal{Q}_{j,17,k}^{\mathrm{RMTL},h}(\tau,a;\widehat{\eta}_{j,k}^{\Lambda,h},\eta_{j,\infty}^{\Lambda,h}) = [P_{n,k}\{\widehat{w}_k^h(a,X)\}]^{-1}\widehat{w}_k^h(a,X)\mathrm{RMTL}_{j,\infty}(\tau\mid a,X)\int_0^\tau \frac{\mathrm{d}\Lambda_\infty(t\mid a,X)}{S_\infty(t\mid a,X)}$$
$$\times \left\{\frac{1}{\widehat{G}_k(t-\mid a,X)} - \frac{1}{G_\infty(t-\mid a,X)}\right\}$$

$$\mathcal{Q}_{j,18,k}^{\mathrm{RMTL},h}(\tau,a;\widehat{\eta}_{j,k}^{\Lambda,h},\eta_{j,\infty}^{\Lambda,h}) = [P_{n,k}\{\widehat{w}_k^h(a,X)\}]^{-1}\widehat{w}_k^h(a,X)\left\{\widehat{\mathrm{RMTL}}_{j,k}(\tau\mid a,X)\int_0^\tau \frac{\mathrm{d}\widehat{\Lambda}_k(t\mid a,X)}{\widehat{S}_k(t\mid a,X)\widehat{G}_k(t-\mid a,X)}\right.$$
$$\left.-\mathrm{RMTL}_{j,\infty}(\tau\mid a,X)\int_0^\tau \frac{\mathrm{d}\Lambda_\infty(t\mid a,X)}{S_\infty(t\mid a,X)\widehat{G}_k(t-\mid a,X)}\right\}$$

$$\mathcal{Q}_{j,19,k}^{\mathrm{RMTL},h}(\tau,a;\widehat{\eta}_{j,k}^{\Lambda,h},\eta_{j,\infty}^{\Lambda,h}) = ([P_{n,k}\{\widehat{w}_k^h(a,X)\}]^{-1} - [E_0\{h_\infty(X)\}]^{-1})w_\infty^h(a,X)\int_0^{\tau\wedge\widetilde{T}} \frac{\mathrm{RMTL}_{j,\infty}(t\mid a,X)\mathrm{d}M_\infty(t\mid a,X)}{S_\infty(t\mid a,X)G_\infty(t-\mid a,X)}$$

$$\mathcal{Q}_{j,20,k}^{\mathrm{RMTL},h}(\tau,a;\widehat{\eta}_{j,k}^{\Lambda,h},\eta_{j,\infty}^{\Lambda,h}) = [P_{n,k}\{\widehat{w}_k^h(a,X)\}]^{-1}\{\widehat{w}_k^h(a,X) - w_\infty^h(a,X)\}\int_0^{\tau\wedge\widetilde{T}} \frac{\mathrm{RMTL}_{j,\infty}(t\mid a,X)\mathrm{d}M_\infty(t\mid a,X)}{S_\infty(t\mid a,X)G_\infty(t-\mid a,X)}$$

$$\mathcal{Q}_{j,21,k}^{\mathrm{RMTL},h}(\tau,a;\widehat{\eta}_{j,k}^{\Lambda,h},\eta_{j,\infty}^{\Lambda,h}) = [P_{n,k}\{\widehat{w}_k^h(a,X)\}]^{-1}\frac{\widehat{w}_k^h(a,X)I(\widetilde{T}\le\tau,\Delta=1)}{G_\infty(\widetilde{T}-\mid a,X)}$$
$$\times \left\{\frac{\widehat{\mathrm{RMTL}}_{j,k}(\widetilde{T}\mid a,X)}{\widehat{S}_k(\widetilde{T}\mid a,X)} - \frac{\mathrm{RMTL}_{j,\infty}(\widetilde{T}\mid a,X)}{S_\infty(\widetilde{T}\mid a,X)}\right\}$$

$$\mathcal{Q}_{j,22,k}^{\mathrm{RMTL},h}(\tau,a;\widehat{\eta}_{j,k}^{\Lambda,h},\eta_{j,\infty}^{\Lambda,h}) = [P_{n,k}\{\widehat{w}_k^h(a,X)\}]^{-1}\frac{\widehat{w}_k^h(a,X)I(\widetilde{T}\le\tau,\Delta=1)\widehat{\mathrm{RMTL}}_{j,k}(\widetilde{T}\mid a,X)}{\widehat{S}_k(\widetilde{T}\mid a,X)}$$
$$\times \left\{\frac{1}{\widehat{G}_k(\widetilde{T}-\mid a,X)} - \frac{1}{G_\infty(\widetilde{T}-\mid a,X)}\right\}$$

$$\mathcal{Q}_{j,23,k}^{\mathrm{RMTL},h}(\tau,a;\widehat{\eta}_{j,k}^{\Lambda,h},\eta_{j,\infty}^{\Lambda,h}) = -[P_{n,k}\{\widehat{w}_k^h(a,X)\}]^{-1}\widehat{w}_k^h(a,X)\int_0^\tau \frac{\mathrm{RMTL}_{j,\infty}(t\mid a,X)\mathrm{d}\Lambda_\infty(t\mid a,X)}{S_\infty(t\mid a,X)}$$
$$\times \left\{\frac{1}{\widehat{G}_k(t-\mid a,X)} - \frac{1}{G_\infty(t-\mid a,X)}\right\}$$

$$\mathcal{Q}_{j,24,k}^{\mathrm{RMTL},h}(\tau,a;\widehat{\eta}_{j,k}^{\Lambda,h},\eta_{j,\infty}^{\Lambda,h}) = -[P_{n,k}\{\widehat{w}_k^h(a,X)\}]^{-1}\widehat{w}_k^h(a,X)\left\{\int_0^\tau \frac{\widehat{\mathrm{RMTL}}_{j,k}(t\mid a,X)\mathrm{d}\widehat{\Lambda}_k(t\mid a,X)}{\widehat{S}_k(t\mid a,X)\widehat{G}_k(t-\mid a,X)}\right.$$
$$\left.- \int_0^\tau \frac{\mathrm{RMTL}_{j,\infty}(t\mid a,X)\mathrm{d}\Lambda_\infty(t\mid a,X)}{S_\infty(t\mid a,X)\widehat{G}_k(t-\mid a,X)}\right\}$$

$$\mathcal{Q}_{j,25,k}^{\mathrm{RMTL},h}(\tau,a;\widehat{\eta}_{j,k}^{\Lambda,h},\eta_{j,\infty}^{\Lambda,h}) = ([P_{n,k}\{\widehat{w}_k^h(a,X)\}]^{-1} - [E_0\{h_\infty(X)\}]^{-1})w_\infty^h(a,X)\tau\int_0^{\tau\wedge\widetilde{T}} \frac{F_{j,\infty}(t\mid a,X)\mathrm{d}M_\infty(t\mid a,X)}{S_\infty(t\mid a,X)G_\infty(t-\mid a,X)}$$

$$\mathcal{Q}_{j,26,k}^{\mathrm{RMTL},h}(\tau,a;\widehat{\eta}_{j,k}^{\Lambda,h},\eta_{j,\infty}^{\Lambda,h}) = [P_{n,k}\{\widehat{w}_k^h(a,X)\}]^{-1}\{\widehat{w}_k^h(a,X) - w_\infty^h(a,X)\}\tau\int_0^{\tau\wedge\widetilde{T}}\frac{F_{j,\infty}(t\mid a,X)\mathrm{d}M_\infty(t\mid a,X)}{S_\infty(t\mid a,X)G_\infty(t-\mid a,X)}$$

$$\mathcal{Q}_{j,27,k}^{\mathrm{RMTL},h}(\tau,a;\widehat{\eta}_{j,k}^{\Lambda,h},\eta_{j,\infty}^{\Lambda,h}) = [P_{n,k}\{\widehat{w}_k^h(a,X)\}]^{-1}\frac{\widehat{w}_k^h(a,X)I(\widetilde{T}\le\tau,\Delta=1)\tau}{G_\infty(\widetilde{T}-\mid a,X)}\left\{\frac{\widehat{F}_{j,k}(\widetilde{T}\mid a,X)}{\widehat{S}_k(\widetilde{T}\mid a,X)} - \frac{F_{j,\infty}(\widetilde{T}\mid a,X)}{S_\infty(\widetilde{T}\mid a,X)}\right\}$$

$$\mathcal{Q}_{j,28,k}^{\mathrm{RMTL},h}(\tau,a;\widehat{\eta}_{j,k}^{\Lambda,h},\eta_{j,\infty}^{\Lambda,h}) = [P_{n,k}\{\widehat{w}_k^h(a,X)\}]^{-1}\frac{\widehat{w}_k^h(a,X)I(\widetilde{T}\le\tau,\Delta=1)\tau F_{j,\infty}(\widetilde{T}\mid a,X)}{\widehat{S}_k(\widetilde{T}\mid a,X)}$$

$$\times\left\{\frac{1}{\widehat{G}_k(\widetilde{T}-\mid a,X)} - \frac{1}{G_\infty(\widetilde{T}-\mid a,X)}\right\}$$

$$\mathcal{Q}_{j,29,k}^{\mathrm{RMTL},h}(\tau,a;\widehat{\eta}_{j,k}^{\Lambda,h},\eta_{j,\infty}^{\Lambda,h}) = -[P_{n,k}\{\widehat{w}_k^h(a,X)\}]^{-1}\widehat{w}_k^h(a,X)\tau\int_0^\tau\frac{F_{j,\infty}(t\mid a,X)\mathrm{d}\Lambda_\infty(t\mid a,X)}{S_\infty(t\mid a,X)}$$

$$\times\left\{\frac{1}{\widehat{G}_k(t-\mid a,X)} - \frac{1}{G_\infty(t-\mid a,X)}\right\}$$

$$\mathcal{Q}_{j,30,k}^{\mathrm{RMTL},h}(\tau,a;\widehat{\eta}_{j,k}^{\Lambda,h},\eta_{j,\infty}^{\Lambda,h}) = -[P_{n,k}\{\widehat{w}_k^h(a,X)\}]^{-1}\widehat{w}_k^h(a,X)\tau\left\{\int_0^\tau\frac{\widehat{F}_{j,k}(t\mid a,X)\mathrm{d}\widehat{\Lambda}_k(t\mid a,X)}{\widehat{S}_k(t\mid a,X)\widehat{G}_k(t-\mid a,X)}\right.$$

$$\left. - \int_0^\tau\frac{F_{j,\infty}(t\mid a,X)\mathrm{d}\Lambda_\infty(t\mid a,X)}{S_\infty(t\mid a,X)\widehat{G}_k(t-\mid a,X)}\right\}$$

$$\mathcal{Q}_{j,31,k}^{\mathrm{RMTL},h}(\tau,a;\widehat{\eta}_{j,k}^{\Lambda,h},\eta_{j,\infty}^{\Lambda,h}) = -([P_{n,k}\{\widehat{w}_k^h(a,X)\}]^{-1} - [E_0\{h_\infty(X)\}]^{-1})w_\infty^h(a,X)\int_0^{\tau\wedge\widetilde{T}}\frac{tF_{j,\infty}(t\mid a,X)\mathrm{d}M_\infty(t\mid a,X)}{S_\infty(t\mid a,X)G_\infty(t-\mid a,X)}$$

$$\mathcal{Q}_{j,32,k}^{\mathrm{RMTL},h}(\tau,a;\widehat{\eta}_{j,k}^{\Lambda,h},\eta_{j,\infty}^{\Lambda,h}) = -[P_{n,k}\{\widehat{w}_k^h(a,X)\}]^{-1}\{\widehat{w}_k^h(a,X) - w_\infty^h(a,X)\}\int_0^{\tau\wedge\widetilde{T}}\frac{tF_{j,\infty}(t\mid a,X)\mathrm{d}M_\infty(t\mid a,X)}{S_\infty(t\mid a,X)G_\infty(t-\mid a,X)}$$

$$\mathcal{Q}_{j,33,k}^{\mathrm{RMTL},h}(\tau,a;\widehat{\eta}_{j,k}^{\Lambda,h},\eta_{j,\infty}^{\Lambda,h}) = -[P_{n,k}\{\widehat{w}_k^h(a,X)\}]^{-1}\frac{\widehat{w}_k^h(a,X)I(\widetilde{T}\le\tau,\Delta=1)\widetilde{T}}{G_\infty(\widetilde{T}-\mid a,X)}\left\{\frac{\widehat{F}_{j,k}(\widetilde{T}\mid a,X)}{\widehat{S}_k(\widetilde{T}\mid a,X)} - \frac{F_{j,\infty}(\widetilde{T}\mid a,X)}{S_\infty(\widetilde{T}\mid a,X)}\right\}$$

$$\mathcal{Q}_{j,34,k}^{\mathrm{RMTL},h}(\tau,a;\widehat{\eta}_{j,k}^{\Lambda,h},\eta_{j,\infty}^{\Lambda,h}) = -[P_{n,k}\{\widehat{w}_k^h(a,X)\}]^{-1}\frac{\widehat{w}_k^h(a,X)I(\widetilde{T}\le\tau,\Delta=1)\widetilde{T}F_{j,\infty}(\widetilde{T}\mid a,X)}{\widehat{S}_k(\widetilde{T}\mid a,X)}$$

$$\times\left\{\frac{1}{\widehat{G}_k(\widetilde{T}-\mid a,X)} - \frac{1}{G_\infty(\widetilde{T}-\mid a,X)}\right\}$$

$$\mathcal{Q}_{j,35,k}^{\mathrm{RMTL},h}(\tau,a;\widehat{\eta}_{j,k}^{\Lambda,h},\eta_{j,\infty}^{\Lambda,h}) = [P_{n,k}\{\widehat{w}_k^h(a,X)\}]^{-1}\widehat{w}_k^h(a,X)\int_0^\tau\frac{tF_{j,\infty}(t\mid a,X)\mathrm{d}\Lambda_\infty(t\mid a,X)}{S_\infty(t\mid a,X)}$$

$$\times\left\{\frac{1}{\widehat{G}_k(t-\mid a,X)} - \frac{1}{G_\infty(t-\mid a,X)}\right\}$$

$$\mathcal{Q}_{j,36,k}^{\mathrm{RMTL},h}(\tau,a;\widehat{\eta}_{j,k}^{\Lambda,h},\eta_{j,\infty}^{\Lambda,h}) = [P_{n,k}\{\widehat{w}_k^h(a,X)\}]^{-1}\widehat{w}_k^h(a,X)\left\{\int_0^\tau\frac{t\widehat{F}_{j,k}(t\mid a,X)\mathrm{d}\widehat{\Lambda}_k(t\mid a,X)}{\widehat{S}_k(t\mid a,X)\widehat{G}_k(t-\mid a,X)}\right.$$

$$\left. - \int_0^\tau\frac{tF_{j,\infty}(t\mid a,X)\mathrm{d}\Lambda_\infty(t\mid a,X)}{S_\infty(t\mid a,X)\widehat{G}_k(t-\mid a,X)}\right\}$$

Similar to $P_0\{\mathcal{Q}_{\iota,k}^{\mathrm{RMST},h}(\tau,a;\widehat{\eta}_{j,k}^{\Lambda,h},\eta_{j,\infty}^{\Lambda,h})\}^2$

$$P_0\{\mathcal{Q}_{j,1,k}^{\mathrm{RMTL},h}(\tau,a;\widehat{\eta}_{j,k}^{\Lambda,h},\eta_{j,\infty}^{\Lambda,h})\}^2 \le \epsilon^2 h_{\max}^2 E_0\left[\sup_{t\in[0,\tau]}\left|\frac{\widehat{\mathrm{RMTL}}_{j,k}(\tau\mid a,X)}{\widehat{S}_k(t\mid a,X)} - \frac{\mathrm{RMTL}_{j,\infty}(\tau\mid a,X)}{S_\infty(t\mid a,X)}\right|\right]^2$$

$$P_0\{\mathcal{Q}_{j,2,k}^{\mathrm{RMTL},h}(\tau,a;\widehat{\eta}_{j,k}^{\Lambda,h},\eta_{j,\infty}^{\Lambda,h})\}^2 \le \tau^2 E_0\left|([P_{n,k}\{\widehat{h}_k(X)\}]^{-1}\widehat{h}_k(X) - [E_0\{h_\infty(X)\}]^{-1}h_\infty(X))\right|^2$$

Since

$$E_0\left|\int_0^{\tau\wedge\widetilde{T}}\frac{\mathrm{d}M_{j,\infty}(t\mid a,X)}{G_\infty(t-\mid a,X)}\right|^2 = E_0\left|\frac{I(\widetilde{T}\le\tau,\widetilde{J}=j)}{G_\infty(\widetilde{T}-\mid a,X)} - \int_0^\tau\frac{\mathrm{d}F_{j,\infty}(t\mid a,X)}{S_\infty(t-\mid a,X)G_\infty(t-\mid a,X)}\right|^2$$

$$\le \epsilon^2 + \epsilon^4 E_0\left|F_{j,\infty}(\tau\mid a,X)\right|^2 \le \epsilon^2 + \epsilon^4$$

we have

$$P_0\{\mathcal{Q}_{j,3,k}^{\text{RMTL},h}(\tau,a;\widehat{\eta}_{j,k}^{\Lambda,h},\eta_{j,\infty}^{\Lambda,h})\}^2 \leq ([P_{n,k}\{\widehat{w}_k^h(a,X)\}]^{-1} - [E_0\{h_\infty(X)\}]^{-1})^2\epsilon^2 h_{\max}^2\tau^2(\epsilon^2+\epsilon^4)$$

$$P_0\{\mathcal{Q}_{j,4,k}^{\text{RMTL},h}(\tau,a;\widehat{\eta}_{j,k}^{\Lambda,h},\eta_{j,\infty}^{\Lambda,h})\}^2 \leq h_{\min}^{-2}\tau^2(\epsilon^2+\epsilon^4)E_0\left|\widehat{w}_k^h(a,X)-w_\infty^h(a,X)\right|^2$$

$$P_0\{\mathcal{Q}_{j,5,k}^{\text{RMTL},h}(\tau,a;\widehat{\eta}_{j,k}^{\Lambda,h},\eta_{j,\infty}^{\Lambda,h})\}^2 \leq h_{\min}^{-2}h_{\max}^2\tau^2\epsilon^2 E_0\left[\sup_{t\in[0,\tau]}\left|\frac{1}{\widehat{G}_k(t-\mid a,X)}-\frac{1}{G_\infty(t-\mid a,X)}\right|\right]^2$$

$$P_0\{\mathcal{Q}_{j,6,k}^{\text{RMTL},h}(\tau,a;\widehat{\eta}_{j,k}^{\Lambda,h},\eta_{j,\infty}^{\Lambda,h})\}^2 \leq h_{\min}^{-2}h_{\max}^2\tau^2\epsilon^2 E_0\left[\sup_{t\in[0,\tau]}\left|\frac{1}{\widehat{G}_k(t-\mid a,X)}-\frac{1}{G_\infty(t-\mid a,X)}\right|\right]^2 E_0\left|\int_0^\tau\frac{\mathrm{d}F_{j,\infty}(t\mid a,X)}{S_\infty(t-\mid a,X)}\right|^2$$

$$\leq h_{\min}^{-2}h_{\max}^2\tau^2\epsilon^2\epsilon^2 E_0\left[\sup_{t\in[0,\tau]}\left|\frac{1}{\widehat{G}_k(t-\mid a,X)}-\frac{1}{G_\infty(t-\mid a,X)}\right|\right]^2$$

$$P_0\{\mathcal{Q}_{j,7,k}^{\text{RMTL},h}(\tau,a;\widehat{\eta}_{j,k}^{\Lambda,h},\eta_{j,\infty}^{\Lambda,h})\}^2 \leq h_{\min}^{-2}h_{\max}^2\tau^2\epsilon^2\epsilon^2 E_0\left[\sup_{t\in[0,\tau]}\left|\widehat{\Lambda}_{j,k}(t\mid a,X)-\Lambda_{j,\infty}(t\mid a,X)\right|\right]^2$$

$$P_0\{\mathcal{Q}_{j,8,k}^{\text{RMTL},h}(\tau,a;\widehat{\eta}_{j,k}^{\Lambda,h},\eta_{j,\infty}^{\Lambda,h})\}^2 \leq ([P_{n,k}\{\widehat{w}_k^h(a,X)\}]^{-1} - [E_0\{h_\infty(X)\}]^{-1})^2\epsilon^2 h_{\max}^2\tau^2(\epsilon^2+\epsilon^4)$$

$$P_0\{\mathcal{Q}_{j,9,k}^{\text{RMTL},h}(\tau,a;\widehat{\eta}_{j,k}^{\Lambda,h},\eta_{j,\infty}^{\Lambda,h})\}^2 \leq h_{\min}^{-2}\tau^2(\epsilon^2+\epsilon^4)E_0\left|\widehat{w}_k^h(a,X)-w_\infty^h(a,X)\right|^2$$

$$P_0\{\mathcal{Q}_{j,10,k}^{\text{RMTL},h}(\tau,a;\widehat{\eta}_{j,k}^{\Lambda,h},\eta_{j,\infty}^{\Lambda,h})\}^2 \leq h_{\min}^{-2}h_{\max}^2\tau^2\epsilon^2 E_0\left[\sup_{t\in[0,\tau]}\left|\frac{1}{\widehat{G}_k(t-\mid a,X)}-\frac{1}{G_\infty(t-\mid a,X)}\right|\right]^2$$

$$P_0\{\mathcal{Q}_{j,11,k}^{\text{RMTL},h}(\tau,a;\widehat{\eta}_{j,k}^{\Lambda,h},\eta_{j,\infty}^{\Lambda,h})\}^2 \leq h_{\min}^{-2}h_{\max}^2\tau^2\epsilon^2\epsilon^2 E_0\left[\sup_{t\in[0,\tau]}\left|\frac{1}{\widehat{G}_k(t-\mid a,X)}-\frac{1}{G_\infty(t-\mid a,X)}\right|\right]^2$$

$$P_0\{\mathcal{Q}_{j,12,k}^{\text{RMTL},h}(\tau,a;\widehat{\eta}_{j,k}^{\Lambda,h},\eta_{j,\infty}^{\Lambda,h})\}^2 \leq h_{\min}^{-2}h_{\max}^2\tau^2\epsilon^2\epsilon^2 E_0\left[\sup_{t\in[0,\tau]}\left|\widehat{\Lambda}_{j,k}(t\mid a,X)-\Lambda_{j,\infty}(t\mid a,X)\right|\right]^2$$

$$P_0\{\mathcal{Q}_{j,13,k}^{\text{RMTL},h}(\tau,a;\widehat{\eta}_{j,k}^{\Lambda,h},\eta_{j,\infty}^{\Lambda,h})\}^2 \leq ([P_{n,k}\{\widehat{w}_k^h(a,X)\}]^{-1} - [E_0\{h_\infty(X)\}]^{-1})^2\epsilon^2 h_{\max}^2\tau^2\times 4\epsilon^4$$

$$P_0\{\mathcal{Q}_{j,14,k}^{\text{RMTL},h}(\tau,a;\widehat{\eta}_{j,k}^{\Lambda,h},\eta_{j,\infty}^{\Lambda,h})\}^2 \leq h_{\min}^{-2}\tau^2\times 4\epsilon^4 E_0\left|\widehat{w}_k^h(a,X)-w_\infty^h(a,X)\right|^2$$

$$P_0\{\mathcal{Q}_{j,15,k}^{\text{RMTL},h}(\tau,a;\widehat{\eta}_{j,k}^{\Lambda,h},\eta_{j,\infty}^{\Lambda,h})\}^2 \leq h_{\min}^{-2}h_{\max}^2\epsilon^4 E_0\left[\sup_{t\in[0,\tau]}\left|\frac{\widehat{\text{RMTL}}_{j,k}(\tau\mid a,X)}{\widehat{S}_k(t\mid a,X)}-\frac{\text{RMTL}_{j,\infty}(\tau\mid a,X)}{S_\infty(t\mid a,X)}\right|\right]^2$$

$$P_0\{\mathcal{Q}_{j,16,k}^{\text{RMTL},h}(\tau,a;\widehat{\eta}_{j,k}^{\Lambda,h},\eta_{j,\infty}^{\Lambda,h})\}^2 \leq h_{\min}^{-2}h_{\max}^2\tau^2\epsilon^4 E_0\left[\sup_{t\in[0,\tau]}\left|\frac{1}{\widehat{G}_k(t-\mid a,X)}-\frac{1}{G_\infty(t-\mid a,X)}\right|\right]^2$$

$$P_0\{\mathcal{Q}_{j,17,k}^{\text{RMTL},h}(\tau,a;\widehat{\eta}_{j,k}^{\Lambda,h},\eta_{j,\infty}^{\Lambda,h})\}^2 \leq h_{\min}^{-2}h_{\max}^2\tau^2\epsilon^2(\epsilon-1)^2 E_0\left[\sup_{t\in[0,\tau]}\left|\frac{1}{\widehat{G}_k(t-\mid a,X)}-\frac{1}{G_\infty(t-\mid a,X)}\right|\right]^2$$

$$P_0\{\mathcal{Q}_{j,18,k}^{\text{RMTL},h}(\tau,a;\widehat{\eta}_{j,k}^{\Lambda,h},\eta_{j,\infty}^{\Lambda,h})\}^2 \leq h_{\min}^{-2}h_{\max}^2\epsilon^2\{\epsilon^2+(\epsilon-1)^2\} E_0\left\{\sup_{t\in[0,\tau]}\left|\frac{\widehat{\text{RMTL}}_{j,k}(\tau\mid a,X)}{\widehat{S}_k(t\mid a,X)}-\frac{\text{RMTL}_{j,\infty}(\tau\mid a,X)}{S_\infty(t\mid a,X)}\right|\right\}^2$$

$$P_0\{\mathcal{Q}_{j,19,k}^{\text{RMTL},h}(\tau,a;\widehat{\eta}_{j,k}^{\Lambda,h},\eta_{j,\infty}^{\Lambda,h})\}^2 \leq ([P_{n,k}\{\widehat{w}_k^h(a,X)\}]^{-1} - [E_0\{h_\infty(X)\}]^{-1})^2\epsilon^2 h_{\max}^2\tau^2\times 4\epsilon^4$$

$$P_0\{\mathcal{Q}_{j,20,k}^{\text{RMTL},h}(\tau,a;\widehat{\eta}_{j,k}^{\Lambda,h},\eta_{j,\infty}^{\Lambda,h})\}^2 \leq h_{\min}^{-2}\tau^2\times 4\epsilon^4 E_0\left|\widehat{w}_k^h(a,X)-w_\infty^h(a,X)\right|^2$$

$$P_0\{\mathcal{Q}_{j,21,k}^{\text{RMTL},h}(\tau,a;\widehat{\eta}_{j,k}^{\Lambda,h},\eta_{j,\infty}^{\Lambda,h})\}^2 \leq h_{\min}^{-2}h_{\max}^2\epsilon^4 E_0\left[\sup_{t\in[0,\tau]}\left|\frac{\widehat{\text{RMTL}}_{j,k}(t\mid a,X)}{\widehat{S}_k(t\mid a,X)}-\frac{\text{RMTL}_{j,\infty}(t\mid a,X)}{S_\infty(t\mid a,X)}\right|\right]^2$$

$$P_0\{\mathcal{Q}_{j,22,k}^{\text{RMTL},h}(\tau,a;\widehat{\eta}_{j,k}^{\Lambda,h},\eta_{j,\infty}^{\Lambda,h})\}^2 \leq h_{\min}^{-2}h_{\max}^2\tau^2\epsilon^4 E_0\left[\sup_{t\in[0,\tau]}\left|\frac{1}{\widehat{G}_k(t-\mid a,X)}-\frac{1}{G_\infty(t-\mid a,X)}\right|\right]^2$$

$$P_0\{\mathcal{Q}_{j,23,k}^{\text{RMTL},h}(\tau,a;\widehat{\eta}_{j,k}^{\Lambda,h},\eta_{j,\infty}^{\Lambda,h})\}^2 \leq h_{\min}^{-2}h_{\max}^2\tau^2\epsilon^2(\epsilon-1)^2 E_0\left[\sup_{t\in[0,\tau]}\left|\frac{1}{\widehat{G}_k(t-\mid a,X)}-\frac{1}{G_\infty(t-\mid a,X)}\right|\right]^2$$

As

$$
E_0 \left| \int_0^\tau \frac{\widehat{\mathrm{RMTL}}_{j,k}(t \mid a, X) \mathrm{d}\widehat{\Lambda}_k(t \mid a, X)}{\widehat{S}_k(t \mid a, X) \widehat{G}_k(t- \mid a, X)} - \int_0^\tau \frac{\mathrm{RMTL}_{j,\infty}(t \mid a, X) \mathrm{d}\Lambda_\infty(t \mid a, X)}{S_\infty(t \mid a, X) \widehat{G}_k(t- \mid a, X)} \right|^2
$$

$$
= E_0 \left| \left. \frac{\widehat{\mathrm{RMTL}}_{j,k}(t \mid a, X)}{\widehat{S}_k(t \mid a, X) \widehat{G}_k(t- \mid a, X)} \right|_0^\tau - \int_0^\tau \frac{1}{\widehat{S}_k(t \mid a, X)} \mathrm{d} \left\{ \frac{\widehat{\mathrm{RMTL}}_{j,k}(t \mid a, X)}{\widehat{G}_k(t- \mid a, X)} \right\} \right.
$$

$$
\left. - \left. \frac{\mathrm{RMTL}_{j,\infty}(t \mid a, X)}{S_\infty(t \mid a, X) \widehat{G}_k(t- \mid a, X)} \right|_0^\tau + \int_0^\tau \frac{1}{S_\infty(t \mid a, X)} \mathrm{d} \left\{ \frac{\mathrm{RMTL}_{j,\infty}(t \mid a, X)}{\widehat{G}_k(t- \mid a, X)} \right\} \right|^2
$$

$$
= E_0 \left| \left\{ \frac{\widehat{\mathrm{RMTL}}_{j,k}(\tau \mid a, X)}{\widehat{S}_k(\tau \mid a, X)} - \frac{\mathrm{RMTL}_{j,\infty}(\tau \mid a, X)}{S_\infty(\tau \mid a, X)} \right\} \frac{1}{\widehat{G}_k(\tau- \mid a, X)} - \int_0^\tau \left\{ \frac{\widehat{F}_{j,k}(t \mid a, X)}{\widehat{S}_k(t \mid a, X)} - \frac{F_{j,\infty}(t \mid a, X)}{S_\infty(t \mid a, X)} \right\} \right.
$$

$$
\left. \frac{\mathrm{d}t}{\widehat{G}_k(t- \mid a, X)} - \int_0^\tau \left\{ \frac{\widehat{\mathrm{RMTL}}_{j,k}(t \mid a, X)}{\widehat{S}_k(t \mid a, X)} - \frac{\mathrm{RMTL}_{j,\infty}(t \mid a, X)}{S_\infty(t \mid a, X)} \right\} \mathrm{d} \left\{ \frac{1}{\widehat{G}_k(t- \mid a, X)} \right\} \right|^2
$$

$$
\leq \left\{ \epsilon^2 + (\epsilon - 1)^2 \right\} E_0 \left\{ \sup_{t \in [0,\tau]} \left| \frac{\widehat{\mathrm{RMTL}}_{j,k}(t \mid a, X)}{\widehat{S}_k(t \mid a, X)} - \frac{\mathrm{RMTL}_{j,\infty}(t \mid a, X)}{S_\infty(t \mid a, X)} \right| \right\}^2
$$

$$
+ \epsilon^2 \tau^2 E_0 \left\{ \sup_{t \in [0,\tau]} \left| \frac{\widehat{F}_{j,k}(t \mid a, X)}{\widehat{S}_k(t \mid a, X)} - \frac{F_{j,\infty}(t \mid a, X)}{S_\infty(t \mid a, X)} \right| \right\}^2
$$

Thus

$$
P_0 \{ \mathcal{Q}_{j,24,k}^{\mathrm{RMTL},h}(\tau, a; \widehat{\eta}_{j,k}^{\Lambda,h}, \eta_{j,\infty}^{\Lambda,h}) \}^2 \leq h_{\min}^{-2} h_{\max}^2 \epsilon^2 \{ \epsilon^2 + (\epsilon - 1)^2 \} E_0 \left\{ \sup_{t \in [0,\tau]} \left| \frac{\widehat{\mathrm{RMTL}}_{j,k}(t \mid a, X)}{\widehat{S}_k(t \mid a, X)} - \frac{\mathrm{RMTL}_{j,\infty}(t \mid a, X)}{S_\infty(t \mid a, X)} \right| \right\}^2
$$

$$
+ h_{\min}^{-2} h_{\max}^2 \epsilon^2 \epsilon^2 \tau^2 E_0 \left\{ \sup_{t \in [0,\tau]} \left| \frac{\widehat{F}_{j,k}(t \mid a, X)}{\widehat{S}_k(t \mid a, X)} - \frac{F_{j,\infty}(t \mid a, X)}{S_\infty(t \mid a, X)} \right| \right\}^2
$$

$$
P_0 \{ \mathcal{Q}_{j,25,k}^{\mathrm{RMTL},h}(\tau, a; \widehat{\eta}_{j,k}^{\Lambda,h}, \eta_{j,\infty}^{\Lambda,h}) \}^2 \leq ([P_{n,k}\{\widehat{w}_k^h(a, X)\}]^{-1} - [E_0\{h_\infty(X)\}]^{-1})^2 \epsilon^2 h_{\max}^2 \tau^2 \times 4\epsilon^4
$$

$$
P_0 \{ \mathcal{Q}_{j,26,k}^{\mathrm{RMTL},h}(\tau, a; \widehat{\eta}_{j,k}^{\Lambda,h}, \eta_{j,\infty}^{\Lambda,h}) \}^2 \leq h_{\min}^{-2} \tau^2 \times 4\epsilon^4 E_0 \left| \widehat{w}_k^h(a, X) - w_\infty^h(a, X) \right|^2
$$

$$
P_0 \{ \mathcal{Q}_{j,27,k}^{\mathrm{RMTL},h}(\tau, a; \widehat{\eta}_{j,k}^{\Lambda,h}, \eta_{j,\infty}^{\Lambda,h}) \}^2 \leq h_{\min}^{-2} h_{\max}^2 \tau^2 \epsilon^4 E_0 \left[ \sup_{t \in [0,\tau]} \left| \frac{\widehat{F}_{j,k}(t \mid a, X)}{\widehat{S}_k(t \mid a, X)} - \frac{F_{j,\infty}(t \mid a, X)}{S_\infty(t \mid a, X)} \right| \right]^2
$$

$$
P_0 \{ \mathcal{Q}_{j,28,k}^{\mathrm{RMTL},h}(\tau, a; \widehat{\eta}_{j,k}^{\Lambda,h}, \eta_{j,\infty}^{\Lambda,h}) \}^2 \leq h_{\min}^{-2} h_{\max}^2 \tau^2 \epsilon^4 E_0 \left[ \sup_{t \in [0,\tau]} \left| \frac{1}{\widehat{G}_k(t- \mid a, X)} - \frac{1}{G_\infty(t- \mid a, X)} \right| \right]^2
$$

$$
P_0 \{ \mathcal{Q}_{j,29,k}^{\mathrm{RMTL},h}(\tau, a; \widehat{\eta}_{j,k}^{\Lambda,h}, \eta_{j,\infty}^{\Lambda,h}) \}^2 \leq h_{\min}^{-2} h_{\max}^2 \tau^2 \epsilon^2 (\epsilon - 1)^2 E_0 \left[ \sup_{t \in [0,\tau]} \left| \frac{1}{\widehat{G}_k(t- \mid a, X)} - \frac{1}{G_\infty(t- \mid a, X)} \right| \right]^2
$$

Note that

$$
E_0 \left| \int_0^\tau \left\{ \frac{\mathrm{d}\widehat{F}_{j,k}(t \mid a, X)}{\widehat{S}_k(t \mid a, X)} - \frac{\mathrm{d}F_{j,\infty}(t \mid a, X)}{S_\infty(t \mid a, X)} \right\} \right|^2
$$

$$
= E_0 \left| \left\{ \frac{\widehat{F}_{j,k}(t \mid a, X)}{\widehat{S}_k(t \mid a, X)} \right\} \Big|_0^\tau - \int_0^\tau \frac{\widehat{F}_{j,k}(t \mid a, X)}{\widehat{S}_k(t \mid a, X)} \mathrm{d}\widehat{\Lambda}_k(t \mid a, X) - \frac{F_{j,\infty}(t \mid a, X)}{S_\infty(t \mid a, X)} \Big|_0^\tau + \int_0^\tau \frac{F_{j,\infty}(t \mid a, X)}{S_\infty(t \mid a, X)} \mathrm{d}\Lambda_\infty(t \mid a, X) \right|^2
$$

$$
= E_0 \left| \frac{\widehat{F}_{j,k}(\tau \mid a, X)}{\widehat{S}_k(\tau \mid a, X)} - \frac{F_{j,\infty}(\tau \mid a, X)}{S_\infty(\tau \mid a, X)} - \int_0^\tau \frac{\widehat{F}_{j,k}(t \mid a, X)}{\widehat{S}_k(t \mid a, X)} \{ \mathrm{d}\widehat{\Lambda}_k(t \mid a, X) - \mathrm{d}\Lambda_\infty(t \mid a, X) \} \right.
$$

$$
\left. - \int_0^\tau \left\{ \frac{\widehat{F}_{j,k}(t \mid a, X)}{\widehat{S}_k(t \mid a, X)} - \frac{F_{j,\infty}(t \mid a, X)}{S_\infty(t \mid a, X)} \right\} \mathrm{d}\Lambda_\infty(t \mid a, X) \right|^2
$$

$$
\leq E_0 \left\{ \sup_{t \in [0,\tau]} \left| \frac{\widehat{F}_{j,k}(t \mid a, X)}{\widehat{S}_k(t \mid a, X)} - \frac{F_{j,\infty}(t \mid a, X)}{S_\infty(t \mid a, X)} \right| \right\}^2 + \epsilon^2 E_0 \left\{ \sup_{t \in [0,\tau]} \left| \widehat{\Lambda}_k(t \mid a, X) - \Lambda_\infty(t \mid a, X) \right| \right\}^2
$$

$$
+ \log^2 \epsilon E_0 \left\{ \sup_{t \in [0,\tau]} \left| \frac{\widehat{F}_{j,k}(t \mid a, X)}{\widehat{S}_k(t \mid a, X)} - \frac{F_{j,\infty}(t \mid a, X)}{S_\infty(t \mid a, X)} \right| \right\}^2
$$

such that

$$
E_0 \left| \int_0^\tau \frac{\widehat{F}_{j,k}(t \mid a, X)\mathrm{d}\widehat{\Lambda}_k(t \mid a, X)}{\widehat{S}_k(t \mid a, X)\widehat{G}_k(t- \mid a, X)} - \int_0^\tau \frac{F_{j,\infty}(t \mid a, X)\mathrm{d}\Lambda_\infty(t \mid a, X)}{S_\infty(t \mid a, X)\widehat{G}_k(t- \mid a, X)} \right|^2
$$

$$
= E_0 \left| \left\{ \frac{\widehat{F}_{j,k}(t \mid a, X)}{\widehat{S}_k(t \mid a, X)\widehat{G}_k(t- \mid a, X)} \right\} \Big|_0^\tau - \int_0^\tau \frac{1}{\widehat{S}_k(t \mid a, X)} \mathrm{d} \left\{ \frac{\widehat{F}_{j,k}(t \mid a, X)}{\widehat{G}_k(t- \mid a, X)} \right\} - \frac{F_{j,\infty}(t \mid a, X)}{S_\infty(t \mid a, X)\widehat{G}_k(t- \mid a, X)} \Big|_0^\tau \right.
$$

$$
\left. + \int_0^\tau \frac{1}{S_\infty(t \mid a, X)} \mathrm{d} \left\{ \frac{F_{j,\infty}(t \mid a, X)}{\widehat{G}_k(t- \mid a, X)} \right\} \right|^2
$$

$$
= E_0 \left| \left\{ \frac{\widehat{F}_{j,k}(\tau \mid a, X)}{\widehat{S}_k(\tau \mid a, X)} - \frac{F_{j,\infty}(\tau \mid a, X)}{S_\infty(\tau \mid a, X)} \right\} \frac{1}{\widehat{G}_k(\tau- \mid a, X)} - \int_0^\tau \left\{ \frac{\mathrm{d}\widehat{F}_{j,k}(t \mid a, X)}{\widehat{S}_k(t \mid a, X)} - \frac{\mathrm{d}F_{j,\infty}(t \mid a, X)}{S_\infty(t \mid a, X)} \right\} \right.
$$

$$
\left. \times \frac{1}{\widehat{G}_k(t- \mid a, X)} - \int_0^\tau \left\{ \frac{\widehat{F}_{j,k}(t \mid a, X)}{\widehat{S}_k(t \mid a, X)} - \frac{F_{j,\infty}(t \mid a, X)}{S_\infty(t \mid a, X)} \right\} \mathrm{d} \left\{ \frac{1}{\widehat{G}_k(t- \mid a, X)} \right\} \right|^2
$$

$$
\leq \{ \epsilon^2 + (\epsilon - 1)^2 \} E_0 \left\{ \sup_{t \in [0,\tau]} \left| \frac{\widehat{F}_{j,k}(t \mid a, X)}{\widehat{S}_k(t \mid a, X)} - \frac{F_{j,\infty}(t \mid a, X)}{S_\infty(t \mid a, X)} \right| \right\}^2 + \epsilon^2 E_0 \left[ \int_0^\tau \left\{ \frac{\mathrm{d}\widehat{F}_{j,k}(t \mid a, X)}{\widehat{S}_k(t \mid a, X)} - \frac{\mathrm{d}F_{j,\infty}(t \mid a, X)}{S_\infty(t \mid a, X)} \right\} \right]^2
$$

$$
= \{ 2\epsilon^2 + (\epsilon - 1)^2 + \epsilon^2 \log^2 \epsilon \} E_0 \left\{ \sup_{t \in [0,\tau]} \left| \frac{\widehat{F}_{j,k}(t \mid a, X)}{\widehat{S}_k(t \mid a, X)} - \frac{F_{j,\infty}(t \mid a, X)}{S_\infty(t \mid a, X)} \right| \right\}^2
$$

$$
+ \epsilon^4 E_0 \left\{ \sup_{t \in [0,\tau]} \left| \widehat{\Lambda}_k(t \mid a, X) - \Lambda_\infty(t \mid a, X) \right| \right\}^2
$$

$$P_0\{\mathcal{Q}_{j,30,k}^{\mathrm{RMTL},h}(\tau,a;\widehat{\eta}_{j,k}^{\Lambda,h},\eta_{j,\infty}^{\Lambda,h})\}^2 \le h_{\min}^{-2}h_{\max}^2\epsilon^2\tau^2\left[\{2\epsilon^2+(\epsilon-1)^2+\epsilon^2\log^2\epsilon\}E_0\left\{\sup_{t\in[0,\tau]}\left|\frac{\widehat{F}_{j,k}(t\mid a,X)}{\widehat{S}_k(t\mid a,X)}\right.\right.\right.$$

$$\left.\left.\left.-\frac{F_{j,\infty}(t\mid a,X)}{S_\infty(t\mid a,X)}\right|\right\}^2+\epsilon^4 E_0\left\{\sup_{t\in[0,\tau]}\left|\widehat{\Lambda}_k(t\mid a,X)-\Lambda_\infty(t\mid a,X)\right|\right\}^2\right]$$

$$P_0\{\mathcal{Q}_{j,31,k}^{\mathrm{RMTL},h}(\tau,a;\widehat{\eta}_{j,k}^{\Lambda,h},\eta_{j,\infty}^{\Lambda,h})\}^2 \le([P_{n,k}\{\widehat{w}_k^h(a,X)\}]^{-1}-[E_0\{h_\infty(X)\}]^{-1})^2\epsilon^2 h_{\max}^2\tau^2\times 4\epsilon^4$$

$$P_0\{\mathcal{Q}_{j,32,k}^{\mathrm{RMTL},h}(\tau,a;\widehat{\eta}_{j,k}^{\Lambda,h},\eta_{j,\infty}^{\Lambda,h})\}^2 \le h_{\min}^{-2}\tau^2\times 4\epsilon^4 E_0\left|\widehat{w}_k^h(a,X)-w_\infty^h(a,X)\right|^2$$

$$P_0\{\mathcal{Q}_{j,33,k}^{\mathrm{RMTL},h}(\tau,a;\widehat{\eta}_{j,k}^{\Lambda,h},\eta_{j,\infty}^{\Lambda,h})\}^2 \le h_{\min}^{-2}h_{\max}^2\tau^2\epsilon^4 E_0\left[\sup_{t\in[0,\tau]}\left|\frac{\widehat{F}_{j,k}(t\mid a,X)}{\widehat{S}_k(t\mid a,X)}-\frac{F_{j,\infty}(t\mid a,X)}{S_\infty(t\mid a,X)}\right|\right]^2$$

$$P_0\{\mathcal{Q}_{j,34,k}^{\mathrm{RMTL},h}(\tau,a;\widehat{\eta}_{j,k}^{\Lambda,h},\eta_{j,\infty}^{\Lambda,h})\}^2 \le h_{\min}^{-2}h_{\max}^2\tau^2\epsilon^4 E_0\left[\sup_{t\in[0,\tau]}\left|\frac{1}{\widehat{G}_k(t-\mid a,X)}-\frac{1}{G_\infty(t-\mid a,X)}\right|\right]^2$$

$$P_0\{\mathcal{Q}_{j,35,k}^{\mathrm{RMTL},h}(\tau,a;\widehat{\eta}_{j,k}^{\Lambda,h},\eta_{j,\infty}^{\Lambda,h})\}^2 \le h_{\min}^{-2}h_{\max}^2\tau^2\epsilon^2(\epsilon-1)^2 E_0\left[\sup_{t\in[0,\tau]}\left|\frac{1}{\widehat{G}_k(t-\mid a,X)}-\frac{1}{G_\infty(t-\mid a,X)}\right|\right]^2$$

Since

$$E_0\left|\int_0^\tau\frac{t\widehat{F}_{j,k}(t\mid a,X)\mathrm{d}\widehat{\Lambda}_k(t\mid a,X)}{\widehat{S}_k(t\mid a,X)\widehat{G}_k(t-\mid a,X)}-\int_0^\tau\frac{tF_{j,\infty}(t\mid a,X)\mathrm{d}\Lambda_\infty(t\mid a,X)}{S_\infty(t\mid a,X)\widehat{G}_k(t-\mid a,X)}\right|^2$$

$$=E_0\left|\frac{\tau}{\widehat{G}_k(\tau-\mid a,X)}\left\{\frac{\widehat{F}_{j,k}(\tau\mid a,X)}{\widehat{S}_k(\tau\mid a,X)}-\frac{F_{j,\infty}(\tau\mid a,X)}{S_\infty(\tau\mid a,X)}\right\}-\int_0^\tau\frac{t}{\widehat{G}_k(t-\mid a,X)}\right.$$

$$\times\left\{\frac{\mathrm{d}\widehat{F}_{j,k}(t\mid a,X)}{\widehat{S}_k(t\mid a,X)}-\frac{\mathrm{d}F_{j,\infty}(t\mid a,X)}{S_\infty(t\mid a,X)}\right\}-\int_0^\tau\left\{\frac{\widehat{F}_{j,k}(t\mid a,X)}{\widehat{S}_k(t\mid a,X)}-\frac{F_{j,\infty}(t\mid a,X)}{S_\infty(t\mid a,X)}\right\}\frac{\mathrm{d}t}{\widehat{G}_k(t-\mid a,X)}$$

$$\left.-\int_0^\tau\left\{\frac{\widehat{F}_{j,k}(t\mid a,X)}{\widehat{S}_k(t\mid a,X)}-\frac{F_{j,\infty}(t\mid a,X)}{S_\infty(t\mid a,X)}\right\}t\mathrm{d}\left\{\frac{1}{\widehat{G}_k(t-\mid a,X)}\right\}\right|^2$$

$$\le\tau^2\epsilon^2 E_0\left\{\sup_{t\in[0,\tau]}\left|\frac{\widehat{F}_{j,k}(t\mid a,X)}{\widehat{S}_k(t\mid a,X)}-\frac{F_{j,\infty}(t\mid a,X)}{S_\infty(t\mid a,X)}\right|\right\}^2+\tau^2\epsilon^2 E_0\left|\int_0^\tau\left\{\frac{\mathrm{d}\widehat{F}_{j,k}(t\mid a,X)}{\widehat{S}_k(t\mid a,X)}-\frac{\mathrm{d}F_{j,\infty}(t\mid a,X)}{S_\infty(t\mid a,X)}\right\}\right|^2$$

$$+\tau^2\epsilon^2 E_0\left\{\sup_{t\in[0,\tau]}\left|\frac{\widehat{F}_{j,k}(t\mid a,X)}{\widehat{S}_k(t\mid a,X)}-\frac{F_{j,\infty}(t\mid a,X)}{S_\infty(t\mid a,X)}\right|\right\}^2+\tau^2(\epsilon-1)^2 E_0\left\{\sup_{t\in[0,\tau]}\left|\frac{\widehat{F}_{j,k}(t\mid a,X)}{\widehat{S}_k(t\mid a,X)}-\frac{F_{j,\infty}(t\mid a,X)}{S_\infty(t\mid a,X)}\right|\right\}^2$$

$$=\tau^2(3\epsilon^2+(\epsilon-1)^2+\epsilon^2\log^2\epsilon)E_0\left\{\sup_{t\in[0,\tau]}\left|\frac{\widehat{F}_{j,k}(t\mid a,X)}{\widehat{S}_k(t\mid a,X)}-\frac{F_{j,\infty}(t\mid a,X)}{S_\infty(t\mid a,X)}\right|\right\}^2$$

$$+\tau^2\epsilon^4 E_0\left\{\sup_{t\in[0,\tau]}\left|\widehat{\Lambda}_k(t\mid a,X)-\Lambda_\infty(t\mid a,X)\right|\right\}^2$$

Therefore

$$P_0\{\mathcal{Q}_{j,36,k}^{\mathrm{RMTL},h}(\tau,a;\widehat{\eta}_{j,k}^{\Lambda,h},\eta_{j,\infty}^{\Lambda,h})\}^2$$

$$\le h_{\min}^{-2}h_{\max}^2\epsilon^2\tau^2\left[\{3\epsilon^2+(\epsilon-1)^2+\epsilon^2\log^2\epsilon\}E_0\left\{\sup_{t\in[0,\tau]}\left|\frac{\widehat{F}_{j,k}(t\mid a,X)}{\widehat{S}_k(t\mid a,X)}-\frac{F_{j,\infty}(t\mid a,X)}{S_\infty(t\mid a,X)}\right|\right\}^2\right.$$

$$\left.+\epsilon^4 E_0\left\{\sup_{t\in[0,\tau]}\left|\widehat{\Lambda}_k(t\mid a,X)-\Lambda_\infty(t\mid a,X)\right|\right\}^2\right] \quad\blacksquare$$

### D.6 LEMMA 4

**Lemma 4** *Under Condition 1-2,*

$$\frac{1}{K}\sum_{k=1}^K\frac{Kn_k^{1/2}}{n}\mathcal{G}_{n,k}\{\widehat{\phi}_k^{\mathrm{RMST},h}(\tau,a;\widehat{\eta}_k^{\Lambda,h})-\phi_\infty^{\mathrm{RMST},h}(\tau,a;\eta_\infty^{\Lambda,h})\}=o_P(n^{-1/2})$$

*If Condition 4 holds additionally,*

$$\frac{1}{K} \sum_{k=1}^{K} \frac{K n_k^{1/2}}{n} \sup_{t \in [0,\tau]} \left| \mathcal{G}_{n,k} \{ \widehat{\phi}_k^{\mathrm{RMST},h}(t, a; \widehat{\eta}_k^{\Lambda,h}) - \phi_\infty^{\mathrm{RMST},h}(t, a; \eta_\infty^{\Lambda,h}) \} \right| = o_P(n^{-1/2})$$

*Under Condition 2 and 5,*

$$\frac{1}{K} \sum_{k=1}^{K} \frac{K n_k^{1/2}}{n} \mathcal{G}_{n,k} \{ \widehat{\phi}_{j,k}^{\mathrm{RMTL},h}(\tau, a; \widehat{\eta}_{j,k}^{\Lambda,h}) - \phi_{j,\infty}^{\mathrm{RMTL},h}(\tau, a; \eta_{j,\infty}^{\Lambda,h}) \} = o_P(n^{-1/2})$$

*If Condition 7 holds additionally,*

$$\frac{1}{K} \sum_{k=1}^{K} \frac{K n_k^{1/2}}{n} \sup_{t \in [0,\tau]} \left| \mathcal{G}_{n,k} \{ \widehat{\phi}_{j,k}^{\mathrm{RMTL},h}(t, a; \widehat{\eta}_{j,k}^{\Lambda,h}) - \phi_{j,\infty}^{\mathrm{RMTL},h}(t, a; \eta_{j,\infty}^{\Lambda,h}) \} \right| = o_P(n^{-1/2})$$

Proof of Lemma 4: Following Lemma 6 of Westling et al. (2023), the empirical process term is bounded by

$$\left| \frac{1}{K} \sum_{k=1}^{K} \frac{K n_k^{1/2}}{n} \mathcal{G}_{n,k} \{ \widehat{\phi}_k^{\mathrm{RMST},h}(\tau, a; \widehat{\eta}_k^{\Lambda,h}) - \phi_\infty^{\mathrm{RMST},h}(\tau, a; \eta_\infty^{\Lambda,h}) \} \right|$$

$$\leq O_P(n^{-1/2}) \frac{1}{K} \sum_{k=1}^{K} \left| \mathcal{G}_{n,k} \{ \widehat{\phi}_k^{\mathrm{RMST},h}(\tau, a; \widehat{\eta}_k^{\Lambda,h}) - \phi_\infty^{\mathrm{RMST},h}(\tau, a; \eta_\infty^{\Lambda,h}) \} \right|$$

By Lemma 3,

$$\frac{1}{K} \sum_{k=1}^{K} \left| \mathcal{G}_{n,k} \{ \widehat{\phi}_k^{\mathrm{RMST},h}(\tau, a; \widehat{\eta}_k^{\Lambda,h}) - \phi_\infty^{\mathrm{RMST},h}(\tau, a; \eta_\infty^{\Lambda,h}) \} \right| \leq \max_k \sum_{\iota=1}^{6} C_\iota^{\mathrm{RMST},h} \mathcal{B}_{\iota,k}^{\mathrm{RMST},h}(\widehat{\eta}_k^{\Lambda,h}, \eta_\infty^{\Lambda,h})$$

where $\mathcal{B}_{\iota,k}^{\mathrm{RMST},h}(\widehat{\eta}_k^{\Lambda,h}, \eta_\infty^{\Lambda,h})$ is defined in Lemma 3.

$$[P_0 \{ \widehat{\phi}_k^{\mathrm{RMST},h}(\tau, a; \widehat{\eta}_k^{\Lambda,h}) - \phi_\infty^{\mathrm{RMST},h}(\tau, a; \eta_\infty^{\Lambda,h}) \}^2]^{1/2} \leq \sum_{\iota=1}^{6} C_\iota^{\mathrm{RMST},h} \mathcal{B}_{\iota,k}^{\mathrm{RMST},h}(\widehat{\eta}_k^{\Lambda,h}, \eta_\infty^{\Lambda,h})$$

$$[P_0 \{ \sup_{t \in [0,\tau]} |\widehat{\phi}_k^{\mathrm{RMST},h}(t, a; \widehat{\eta}_k^{\Lambda,h}) - \phi_\infty^{\mathrm{RMST},h}(t, a; \eta_\infty^{\Lambda,h})| \}^2]^{1/2}$$

$$\leq C_1^{\mathrm{RMST},h} \widetilde{\mathcal{B}}_{1,k}^{\mathrm{RMST},h}(\widehat{\eta}_k^{\Lambda,h}, \eta_\infty^{\Lambda,h}) + \sum_{\iota=2}^{6} C_\iota^{\mathrm{RMST},h} \mathcal{B}_{\iota,k}^{\mathrm{RMST},h}(\widehat{\eta}_k^{\Lambda,h}, \eta_\infty^{\Lambda,h})$$

$$[P_0 \{ \widehat{\phi}_{j,k}^{\mathrm{RMTL},h}(\tau, a; \widehat{\eta}_{j,k}^{\Lambda,h}) - \phi_{j,\infty}^{\mathrm{RMTL},h}(\tau, a; \eta_{j,\infty}^{\Lambda,h}) \}^2]^{1/2} \leq \sum_{\iota=1}^{9} C_{j,\iota}^{\mathrm{RMTL},h} \mathcal{B}_{j,\iota,k}^{\mathrm{RMTL},h}(\widehat{\eta}_{j,k}^{\Lambda,h}, \eta_{j,\infty}^{\Lambda,h})$$

$$[P_0 \{ \sup_{t \in [0,\tau]} |\widehat{\phi}_{j,k}^{\mathrm{RMTL},h}(t, a; \widehat{\eta}_{j,k}^{\Lambda,h}) - \phi_{j,\infty}^{\mathrm{RMTL},h}(t, a; \eta_{j,\infty}^{\Lambda,h})| \}^2]^{1/2}$$

$$\leq C_{j,1}^{\mathrm{RMTL},h} \widetilde{\mathcal{B}}_{j,1,k}^{\mathrm{RMTL},h}(\widehat{\eta}_{j,k}^{\Lambda,h}, \eta_{j,\infty}^{\Lambda,h}) + \sum_{\iota=2}^{9} C_{j,\iota}^{\mathrm{RMTL},h} \mathcal{B}_{j,\iota,k}^{\mathrm{RMTL},h}(\widehat{\eta}_{j,k}^{\Lambda,h}, \eta_{j,\infty}^{\Lambda,h})$$

Based on Condition 1, $\max_k \sum_{\iota=1}^{6} C_\iota^{\mathrm{RMST},h} \mathcal{B}_{\iota,k}^{\mathrm{RMST},h}(\widehat{\eta}_k^{\Lambda,h}, \eta_\infty^{\Lambda,h}) \xrightarrow{P} 0$, such that

$$\left| \frac{1}{K} \sum_{k=1}^{K} \frac{K n_k^{1/2}}{n} \mathcal{G}_{n,k} \{ \widehat{\phi}_k^{\mathrm{RMST},h}(\tau, a; \widehat{\eta}_k^{\Lambda,h}) - \phi_\infty^{\mathrm{RMST},h}(\tau, a; \eta_\infty^{\Lambda,h}) \} \right| = o_P(n^{-1/2})$$

For simultaneous inference, we bound the empirical stochastic process

$$\left| \frac{1}{K} \sum_{k=1}^{K} \frac{K n_k^{1/2}}{n} \sup_{t \in [0,\tau]} \mathcal{G}_{n,k} \{ \widehat{\phi}_k^{\mathrm{RMST},h}(t, a; \widehat{\eta}_k^{\Lambda,h}) - \phi_\infty^{\mathrm{RMST},h}(t, a; \eta_\infty^{\Lambda,h}) \} \right|$$

$$\leq o_P(n^{-1/2}) \frac{1}{K} \sum_{k=1}^{K} \left| \sup_{t \in [0,\tau]} \mathcal{G}_{n,k} \{ \widehat{\phi}_k^{\mathrm{RMST},h}(t, a; \widehat{\eta}_k^{\Lambda,h}) - \phi_\infty^{\mathrm{RMST},h}(t, a; \eta_\infty^{\Lambda,h}) \} \right|$$

Using Theorem 2.14.2 from van der Vaart & Wellner (1996), we have

$$
\frac{1}{K} \sum_{k=1}^{K} \left| \sup_{t \in [0,\tau]} \mathcal{G}_{n,k} \{ \widehat{\phi}_k^{\mathrm{RMST},h}(t,a;\widehat{\eta}_k^{\Lambda,h}) - \phi_\infty^{\mathrm{RMST},h}(t,a;\eta_\infty^{\Lambda,h}) \} \right|
$$

$$
\leq \max_k \left\{ C_1^{\mathrm{RMST},h} \widetilde{\mathcal{B}}_{1,k}^{\mathrm{RMST},h}(\widehat{\eta}_k^{\Lambda,h}, \eta_\infty^{\Lambda,h}) + \sum_{\iota=2}^{6} C_\iota^{\mathrm{RMST},h} \mathcal{B}_{\iota,k}^{\mathrm{RMST},h}(\widehat{\eta}_k^{\Lambda,h}, \eta_\infty^{\Lambda,h}) \right\}
$$

Based on Condition 1 and 4, $\max_k \left\{ C_1^{\mathrm{RMST},h} \widetilde{\mathcal{B}}_{1,k}^{\mathrm{RMST},h}(\widehat{\eta}_k^{\Lambda,h}, \eta_\infty^{\Lambda,h}) + \sum_{\iota=2}^{6} C_\iota^{\mathrm{RMST},h} \mathcal{B}_{\iota,k}^{\mathrm{RMST},h}(\widehat{\eta}_k^{\Lambda,h}, \eta_\infty^{\Lambda,h}) \right\} \xrightarrow{P}$ 0, such that

$$
\frac{1}{K} \sum_{k=1}^{K} \left| \sup_{t \in [0,\tau]} \mathcal{G}_{n,k} \{ \widehat{\phi}_k^{\mathrm{RMST},h}(t,a;\widehat{\eta}_k^{\Lambda,h}) - \phi_\infty^{\mathrm{RMST},h}(t,a;\eta_\infty^{\Lambda,h}) \} \right| = o_P(n^{-1/2})
$$

Similar arguments also applies to $1/K \sum_{k=1}^{K} \left| \mathcal{G}_{n,k} \{ \widehat{\phi}_{j,k}^{\mathrm{RMTL},h}(\tau,a;\widehat{\eta}_{j,k}^{\Lambda,h}) - \phi_{j,\infty}^{\mathrm{RMTL},h}(\tau,a;\eta_{j,\infty}^{\Lambda,h}) \} \right|$ and $1/K \sum_{k=1}^{K} \left| \sup_{t \in [0,\tau]} \mathcal{G}_{n,k} \{ \widehat{\phi}_{j,k}^{\mathrm{RMTL},h}(t,a;\widehat{\eta}_{j,k}^{\Lambda,h}) - \phi_{j,\infty}^{\mathrm{RMTL},h}(t,a;\eta_{j,\infty}^{\Lambda,h}) \} \right|$ by replacing Condition 1 and Condition 4 with Condition 5 and Condition 7. ∎

### D.7 PROOF OF THEOREM 2

**Condition 1** *There exist $\eta_\infty^{\Lambda,h}$ such that $\max_k \{ \mathcal{B}_{\iota,k}^{\mathrm{RMST},h}(\widehat{\eta}_k^{\Lambda,h}, \eta_\infty^{\Lambda,h}) \}^2 \xrightarrow{P} 0$ for all $a$, $\tau$, and $\iota = 1, \ldots, 14$, where $\mathcal{B}_{\iota,k}^{\mathrm{RMST},h}(\widehat{\eta}_k^{\Lambda,h}, \eta_\infty^{\Lambda,h})$ is defined in Lemma 3.*

**Condition 2** *There exists $\epsilon \in (1, \infty)$ such that $\{ \widehat{S}_k(t \mid a, X), S_\infty(t \mid a, X), \widehat{\pi}_k(a \mid X), \pi_\infty(a \mid X), \widehat{G}_k(t \mid a, X), G_\infty(t \mid a, X) \} \geq 1/\epsilon$, $h_{\min} \leq \{ \widehat{h}_k(X), h_\infty(X) \} \leq h_{\max}$, and $P_{n,k} \{ \widehat{h}_k(X) \} > 0$, $E_0 \{ h_\infty(X) \} > 0$ for all $a$, $X$ and $k$.*

**Condition 3** *One of the following conditions is satisfied for all $t \in [0,\tau], a \in \{0,1\}$ and almost all $x$: (1) $h_\infty(X) = h_0(X)$ and $S_\infty(t \mid a, X) = S_0(t \mid a, X)$; or (2) $h_\infty(X) = h_0(X)$ and $\pi_\infty(a \mid X) = \pi_0(a \mid X)$ and $G_\infty(t \mid a, X) = G_0(t \mid a, X)$.*

**Condition 4** $\max_k \{ \widetilde{\mathcal{B}}_{1,k}^{\mathrm{RMST},h}(\widehat{\eta}_k^{\Lambda,h}, \eta_\infty^{\Lambda,h}) \}^2 \xrightarrow{P} 0$ *for every $a$ and $\tau$.*

**Condition 5** *There exist $\eta_{j,\infty}^{\Lambda,h}$ such that $\max_k \{ \mathcal{B}_{j,\iota,k}^{\mathrm{RMTL},h}(\widehat{\eta}_{j,k}^{\Lambda,h}, \eta_{j,\infty}^{\Lambda,h}) \}^2 \xrightarrow{P} 0$ for cause $j$ of interest, and for all $a$, $\tau$, $\iota = 1, \ldots, 36$, where $\mathcal{B}_{j,\iota,k}^{\mathrm{RMTL},h}(\widehat{\eta}_{j,k}^{\Lambda,h}, \eta_{j,\infty}^{\Lambda,h})$ is defined in Lemma 3.*

**Condition 6** *One of the following conditions is satisfied for all $t \in [0,\tau], a \in \{0,1\}$, almost all $X$, and cause $j$ of interest only: (1) $h_\infty(X) = h_0(X)$ and $S_\infty(t \mid a, X) = S_0(t \mid a, X)$ and $F_{j,\infty}(t \mid a, X) = F_{j,0}(t \mid a, X)$; or (2) $h_\infty(X) = h_0(X)$ and $\pi_\infty(a \mid X) = \pi_0(a \mid X)$ and $G_\infty(t \mid a, X) = G_0(t \mid a, X)$; or (3) $h_\infty(X) = h_0(X)$ and $F_{j,\infty}(t \mid a, X) = F_{j,0}(t \mid a, X)$ and $G_\infty(t \mid a, X) = G_0(t \mid a, X)$.*

**Condition 7** $\max_k \{ \widetilde{\mathcal{B}}_{j,1,k}^{\mathrm{RMTL},h}(\widehat{\eta}_{j,k}^{\Lambda,h}, \eta_{j,\infty}^{\Lambda,h}) \}^2 \xrightarrow{P} 0$ *for cause $j$ of interest and every $a$, $\tau$.*

For Condition 3, if (1) $h_\infty(X) = h_0(X)$ and $S_\infty(t \mid a, X) = S_0(t \mid a, X)$ holds, then $\mathrm{RMST}_\infty(t \mid a, X) = \mathrm{RMST}_0(t \mid a, X)$, $\Lambda_\infty(t \mid a, X) = \Lambda_0(t \mid a, X)$, and $\mathrm{d}M_\infty(t \mid a, X) = \mathrm{d}M_0(t \mid a, X)$. We consider the event martingale representation of $\widehat{\psi}^{\mathrm{RMST},h}(\tau,a)$ since $E_0 \{ \int_0^{\tau \wedge \widetilde{T}} (\cdot) \mathrm{d}M_0(t \mid a, X) \mid A = a, X \} = 0$,

$$\widehat{\psi}^{\mathrm{RMST},h}(\tau, a) \xrightarrow{p}$$

$$[E_0\{h_0(X)\}]^{-1} E_0\{h_0(X)\mathrm{RMST}_0(\tau \mid a, X)\} + \left[E_0\left\{\frac{h_0(X)I(A=a)}{\pi_\infty(a \mid X)}\right\}\right]^{-1}$$

$$\times E_0\left[\left\{\frac{h_0(X)I(A=a)}{\pi_\infty(a \mid X)}\right\}\left\{-\mathrm{RMST}_0(\tau \mid a, X)\int_0^{\tau \wedge \widetilde{T}} \frac{\mathrm{d}M_0(t \mid a, X)}{S_0(t \mid a, X)G_\infty(t- \mid a, X)}\right.\right.$$

$$\left.\left. + \int_0^{\tau \wedge \widetilde{T}} \frac{\mathrm{RMST}_0(t \mid a, X)\mathrm{d}M_0(t \mid a, X)}{S_0(t \mid a, X)G_\infty(t- \mid a, X)}\right\}\right]$$

$$=[E_0\{h_0(X)\}]^{-1} E_0\{h_0(X)\mathrm{RMST}_0(\tau \mid a, X)\} + \left[E_0\left\{\frac{h_0(X)\pi_0(a \mid X)}{\pi_\infty(a \mid X)}\right\}\right]^{-1}$$

$$\times E_0\left(\left\{\frac{h_0(X)\pi_0(a \mid X)}{\pi_\infty(a \mid X)}\right\}\left[-\mathrm{RMST}_0(\tau \mid a, X)E\left\{\int_0^{\tau \wedge \widetilde{T}} \frac{\mathrm{d}M_0(t \mid a, X)}{S_0(t \mid a, X)G_\infty(t- \mid a, X)} \mid A = a, X\right\}\right.\right.$$

$$\left.\left. + E_0\left\{\int_0^{\tau \wedge \widetilde{T}} \frac{\mathrm{RMST}_0(t \mid a, X)\mathrm{d}M_0(t \mid a, X)}{S_0(t \mid a, X)G_\infty(t- \mid a, X)} \mid A = a, X\right\}\right]\right)$$

$$=[E_0\{h_0(X)\}]^{-1} E_0\{h_0(X)\mathrm{RMST}_0(\tau \mid a, X)\} = \psi_0^{\mathrm{RMST},h}(\tau, a)$$

If (2) $h_\infty(X) = h_0(X)$ and $\pi_\infty(a \mid X) = \pi_0(a \mid X)$ and $G_\infty(t \mid a, X) = G_0(t \mid a, X)$ holds, then $\Lambda_\infty^C(t \mid a, X) = \Lambda_0^C(t \mid a, X)$, and $\mathrm{d}M_\infty^C(t \mid a, X) = \mathrm{d}M_0^C(t \mid a, X)$. We consider the censoring martingale representation of $\widehat{\psi}^{\mathrm{RMST},h}(\tau, a)$ since $E\{\int_0^{\tau \wedge \widetilde{T}}(\cdot)\mathrm{d}M_0^C(t \mid a, X) \mid A = a, X\} = 0$,

$$\widehat{\psi}^{\mathrm{RMST},h}(\tau, a) \xrightarrow{P}$$

$$=[E_0\{h_0(X)\}]^{-1} E_0\{h_0(X)\mathrm{RMST}_\infty(\tau \mid a, X)\} + \left[E_0\left\{\frac{h_0(X)I(A=a)}{\pi_0(a \mid X)}\right\}\right]^{-1}$$

$$\times E_0\left[\left\{\frac{h_0(X)I(A=a)}{\pi_0(a \mid X)}\right\}\left\{\frac{\Delta(\tau)\min(\widetilde{T}, \tau)}{G_0(\tau \wedge \widetilde{T}- \mid a, X)} - \mathrm{RMST}_\infty(\tau \mid a, X)\right.\right.$$

$$\left.\left. + \int_0^{\tau \wedge \widetilde{T}} \frac{t\mathrm{d}M_0^C(t \mid a, X)}{G_0(t- \mid a, X)} + \int_0^{\tau \wedge \widetilde{T}} \frac{\{\mathrm{RMST}_\infty(\tau \mid a, X) - \mathrm{RMST}_\infty(t \mid a, X)\}\mathrm{d}M_0^C(t \mid a, X)}{S_\infty(t \mid a, X)G_0(t- \mid a, X)}\right\}\right]$$

$$=[E_0\{h_0(X)\}]^{-1} E_0\{h_0(X)\mathrm{RMST}_\infty(\tau \mid a, X)\} + [E_0\{h_0(X)\}]^{-1}$$

$$\times E_0\left(h_0(X)\left[E\left\{\frac{\Delta(\tau)\min(\widetilde{T}, \tau)}{G_0(\tau \wedge \widetilde{T}- \mid a, X)} \mid A = a, X\right\} - \mathrm{RMST}_\infty(\tau \mid a, X) + E_0\left\{\int_0^{\tau \wedge \widetilde{T}} \frac{t\mathrm{d}M_0^C(t \mid a, X)}{G_0(t- \mid a, X)} \mid A = a, X\right\}\right.\right.$$

$$\left.\left. + E_0\left\{\int_0^{\tau \wedge \widetilde{T}} \frac{\{\mathrm{RMST}_\infty(\tau \mid a, X) - \mathrm{RMST}_\infty(t \mid a, X)\}\mathrm{d}M_0^C(t \mid a, X)}{S_\infty(t \mid a, X)G_0(t- \mid a, X)} \mid A = a, X\right\}\right]\right)$$

$$=[E_0\{h_0(X)\}]^{-1} E_0\{h_0(X)\mathrm{RMST}_0(\tau \mid a, X)\} = \psi_0^{\mathrm{RMST},h}(\tau, a)$$

We used

$$E_0\left\{\frac{\Delta(\tau)\min(\widetilde{T}, \tau)}{G_0(\tau \wedge \widetilde{T}- \mid a, X)} \mid A = a, X\right\}$$

$$=E_0\left\{\frac{I(C > \tau \wedge \widetilde{T}-)\min(T, \tau)}{G_0(\tau \wedge \widetilde{T}- \mid a, X)} \mid A = a, X\right\} = \mathrm{RMST}_0(\tau \mid a, X)$$

The proof above also implies $E_0\{\phi_0^{\mathrm{RMST}}(\tau, a; \eta_0^\Lambda) \mid A = a, X\} = \mathrm{RMST}_0(\tau \mid a, X)$.

For Condition 6, if (1) $h_\infty(X) = h_0(X)$ and $S_\infty(t \mid a, X) = S_0(t \mid a, X)$ and $F_{j,\infty}(t \mid a, X) = F_{j,0}(t \mid a, X)$ holds, then $\mathrm{RMTL}_{j,\infty}(t \mid a, X) = \mathrm{RMTL}_{j,0}(t \mid a, X)$, $\Lambda_\infty(t \mid a, X) = \Lambda_0(t \mid a, X)$, $\Lambda_{j,\infty}(t \mid a, X) = \Lambda_{j,0}(t \mid a, X)$, $\mathrm{d}M_\infty(t \mid a, X) = \mathrm{d}M_0(t \mid a, X)$, and $\mathrm{d}M_{j,\infty}(t \mid a, X) = \mathrm{d}M_{j,0}(t \mid a, X)$. We consider the event martingale representation of $\widehat{\psi}_j^{\mathrm{RMTL},h}(\tau, a)$ since $E[\int_0^{\tau \wedge \widetilde{T}}(\cdot)\mathrm{d}M_0(t \mid a, X) \mid A =$

$a, X] = 0$, and $E[\int_0^{\tau \wedge \widetilde{T}} (\cdot) \mathrm{d}M_{j,0}(t \mid a, X) \mid A = a, X] = 0$,

$$\widehat{\psi}_j^{\mathrm{RMTL},h}(\tau, a) \xrightarrow{P}$$

$$[E_0\{h_0(X)\}]^{-1} E_0\{h_0(X)\mathrm{RMTL}_{j,0}(\tau \mid a, X)\} + \left[E_0\left\{\frac{h_0(X)I(A=a)}{\pi_\infty(a \mid X)}\right\}\right]^{-1}$$

$$\times E_0\left[\left\{\frac{h_0(X)I(A=a)}{\pi_\infty(a \mid X)}\right\}\left\{\int_0^{\tau \wedge \widetilde{T}} \frac{(\tau-t)\mathrm{d}M_{j,0}(t \mid a, X)}{G_\infty(t- \mid a, X)} + \int_0^{\tau \wedge \widetilde{T}} \frac{(\tau-t)F_{j,0}(t \mid a, X)\mathrm{d}M_0(t \mid a, X)}{S_0(t \mid a, X)G_\infty(t- \mid a, X)}\right.\right.$$

$$\left.\left. - \int_0^{\tau \wedge \widetilde{T}} \frac{\{\mathrm{RMST}_0(\tau \mid a, X) - \mathrm{RMST}_0(t \mid a, X)\}\mathrm{d}M_0(t \mid a, X)}{S_0(t \mid a, X)G_\infty(t- \mid a, X)}\right\}\right]$$

$$=[E_0\{h_0(X)\}]^{-1} E_0\{h_0(X)\mathrm{RMTL}_{j,0}(\tau \mid a, X)\} + \left[E_0\left\{\frac{h_0(X)\pi_0(a \mid X)}{\pi_\infty(a \mid X)}\right\}\right]^{-1}$$

$$\times E_0\left(\left\{\frac{h_0(X)\pi_0(a \mid X)}{\pi_\infty(a \mid X)}\right\}\left[E_0\left\{\int_0^{\tau \wedge \widetilde{T}} \frac{(\tau-t)\mathrm{d}M_{j,0}(t \mid a, X)}{G_\infty(t- \mid a, X)} \mid A = a, X\right\}\right.\right.$$

$$+ E_0\left\{\int_0^{\tau \wedge \widetilde{T}} \frac{(\tau-t)F_{j,0}(t \mid a, X)\mathrm{d}M_0(t \mid a, X)}{S_0(t \mid a, X)G_\infty(t- \mid a, X)} \mid A = a, X\right\}$$

$$\left.\left. - E_0\left\{\int_0^{\tau \wedge \widetilde{T}} \frac{\{\mathrm{RMST}_0(\tau \mid a, X) - \mathrm{RMST}_0(t \mid a, X)\}\mathrm{d}M_0(t \mid a, X)}{S_0(t \mid a, X)G_\infty(t- \mid a, X)} \mid A = a, X\right\}\right]\right)$$

$$=[E_0\{h_0(X)\}]^{-1} E_0\{h_0(X)\mathrm{RMTL}_{j,0}(\tau \mid a, X)\} = \psi_{j,0}^{\mathrm{RMTL},h}(\tau, a)$$

If (2) $h_\infty(X) = h_0(X)$ and $\pi_\infty(a \mid X) = \pi_0(a \mid X)$ and $G_\infty(t \mid a, X) = G_0(t \mid a, X)$ holds, then $\Lambda_\infty^C(t \mid a, X) = \Lambda_0^C(t \mid a, X)$, and $\mathrm{d}M_\infty^C(t \mid a, X) = \mathrm{d}M_0^C(t \mid a, X)$. We consider the censoring martingale representation of $\widehat{\psi}_j^{\mathrm{RMTL},h}(\tau, a)$ since $E\{\int_0^{\tau \wedge \widetilde{T}} (\cdot) \mathrm{d}M_0^C(t \mid a, X) \mid A = a, X\} = 0$,

$$\widehat{\psi}_j^{\mathrm{RMTL},h}(\tau, a) \xrightarrow{P}$$

$$[E_0\{h_0(X)\}]^{-1} E_0\{h_0(X)\mathrm{RMTL}_{j,\infty}(\tau \mid a, X)\} + \left[E_0\left\{\frac{h_0(X)I(A=a)}{\pi_0(a \mid X)}\right\}\right]^{-1} E_0\left[\left\{\frac{h_0(X)I(A=a)}{\pi_0(a \mid X)}\right\}\right.$$

$$\times \left\{\frac{(\tau - \min(\widetilde{T}, \tau))I(\widetilde{J} = j)}{G_0(\widetilde{T}- \mid a, X)} - \mathrm{RMTL}_{j,\infty}(\tau \mid a, X) - \int_0^{\tau \wedge \widetilde{T}} \frac{(\tau-t)F_{j,\infty}(t \mid a, X)\mathrm{d}M_0^C(t \mid a, X)}{S_\infty(t \mid a, X)G_0(t- \mid a, X)}\right.$$

$$\left.\left. + \int_0^{\tau \wedge \widetilde{T}} \frac{\{\mathrm{RMTL}_{j,\infty}(\tau \mid a, X) - \mathrm{RMTL}_{j,\infty}(t \mid a, X)\}\mathrm{d}M_0^C(t \mid a, X)}{S_\infty(t \mid a, X)G_0(t- \mid a, X)}\right\}\right]$$

$$=[E_0\{h_0(X)\}]^{-1} E_0\{h_0(X)\mathrm{RMTL}_{j,\infty}(\tau \mid a, X)\} + [E_0\{h_0(X)\}]^{-1}$$

$$\times E_0\left(h_0(X)\left[E_0\left\{\frac{(\tau - \min(\widetilde{T}, \tau))I(\widetilde{J} = j)}{G_0(\widetilde{T}- \mid a, X)} \mid A = a, X\right\} - \mathrm{RMTL}_{j,\infty}(\tau \mid a, X)\right.\right.$$

$$- E_0\left\{\int_0^{\tau \wedge \widetilde{T}} \frac{(\tau-t)F_{j,\infty}(t \mid a, X)\mathrm{d}M_0^C(t \mid a, X)}{S_\infty(t \mid a, X)G_0(t- \mid a, X)} \mid A = a, X\right\}$$

$$\left.\left. + E_0\left\{\int_0^{\tau \wedge \widetilde{T}} \frac{\{\mathrm{RMTL}_{j,\infty}(\tau \mid a, X) - \mathrm{RMTL}_{j,\infty}(t \mid a, X)\}\mathrm{d}M_0^C(t \mid a, X)}{S_\infty(t \mid a, X)G_0(t- \mid a, X)} \mid A = a, X\right\}\right]\right)$$

$$=[E_0\{h_0(X)\}]^{-1} E_0\{h_0(X)\mathrm{RMTL}_{j,0}(\tau \mid a, X)\} = \psi_{j,0}^{\mathrm{RMTL},h}(\tau, a)$$

We plugged in

$$E_0\left\{\frac{(\tau - \min(\widetilde{T}, \tau))I(\widetilde{J} = j)}{G_0(\widetilde{T}- \mid a, X)} \mid A = a, X\right\}$$

$$=E_0\left\{\frac{\Delta(\tau - \min(T, \tau))I(J = j)}{G_0(\widetilde{T}- \mid a, X)} \mid A = a, X\right\} = \mathrm{RMTL}_{j,0}(\tau \mid a, X)$$

If (3) $h_\infty(X) = h_0(X)$ and $F_{j,\infty}(t \mid a, X) = F_{j,0}(t \mid a, X)$ and $G_\infty(t \mid a, X) = G_0(t \mid a, X)$ holds, then $\mathrm{RMTL}_{j,\infty}(t \mid a, X) = \mathrm{RMTL}_{j,0}(t \mid a, X)$, $\Lambda_\infty^C(t \mid a, X) = \Lambda_0^C(t \mid a, X)$, and $\mathrm{d}M_\infty^C(t \mid a, X) = \mathrm{d}M_0^C(t \mid a, X)$. We consider the censoring martingale representation of $\widehat{\psi}_j^{\mathrm{RMTL},h}(\tau, a)$ since $E[\int_0^{\tau \wedge \widetilde{T}} (\cdot) \mathrm{d}M_0^C(t \mid a, X) \mid A = a, X] = 0$

$$\widehat{\psi}_j^{\mathrm{RMTL},h}(\tau, a) \xrightarrow{P}$$

$$[E_0\{h_0(X)\}]^{-1} E_0\{h_0(X)\mathrm{RMTL}_{j,0}(\tau \mid a, X)\} + \left[ E_0 \left\{ \frac{h_0(X)I(A=a)}{\pi_\infty(a \mid X)} \right\} \right]^{-1} E_0 \left[ \left\{ \frac{h_0(X)I(A=a)}{\pi_\infty(a \mid X)} \right\} \right.$$

$$\times \left\{ \frac{(\tau - \min(\widetilde{T}, \tau))I(\widetilde{J} = j)}{G_0(\widetilde{T}- \mid a, X)} - \mathrm{RMTL}_{j,0}(\tau \mid a, X) - \int_0^{\tau \wedge \widetilde{T}} \frac{(\tau - t)F_{j,0}(t \mid a, X)\mathrm{d}M_0^C(t \mid a, X)}{S_\infty(t \mid a, X)G_0(t- \mid a, X)} \right.$$

$$\left. \left. + \int_0^{\tau \wedge \widetilde{T}} \frac{\{\mathrm{RMTL}_{j,0}(\tau \mid a, X) - \mathrm{RMTL}_{j,0}(t \mid a, X)\}\mathrm{d}M_0^C(t \mid a, X)}{S_\infty(t \mid a, X)G_0(t- \mid a, X)} \right\} \right]$$

$$= [E_0\{h_0(X)\}]^{-1} E_0\{h_0(X)\mathrm{RMTL}_{j,0}(\tau \mid a, X)\} + \left[ E_0 \left\{ \frac{h_0(X)\pi_0(a \mid X)}{\pi_\infty(a \mid X)} \right\} \right]^{-1}$$

$$\times E_0 \left( \left[ E_0 \left\{ \frac{h_0(X)\pi_0(a \mid X)}{\pi_\infty(a \mid X)} \right\} \left\{ \frac{(\tau - \min(\widetilde{T}, \tau))I(\widetilde{J} = j)}{G_0(\widetilde{T}- \mid a, X)} \mid A = a, X \right\} - \mathrm{RMTL}_{j,0}(\tau \mid a, X) \right. \right.$$

$$- E_0 \left\{ \int_0^{\tau \wedge \widetilde{T}} \frac{(\tau - t)F_{j,0}(t \mid a, X)\mathrm{d}M_0^C(t \mid a, X)}{S_\infty(t \mid a, X)G_0(t- \mid a, X)} \mid A = a, X \right\}$$

$$\left. \left. + E_0 \left\{ \int_0^{\tau \wedge \widetilde{T}} \frac{\{\mathrm{RMTL}_{j,0}(\tau \mid a, X) - \mathrm{RMTL}_{j,0}(t \mid a, X)\}\mathrm{d}M_0^C(t \mid a, X)}{S_\infty(t \mid a, X)G_0(t- \mid a, X)} \mid A = a, X \right\} \right] \right)$$

$$= [E_0\{h_0(X)\}]^{-1} E_0\{h_0(X)\mathrm{RMTL}_{j,0}(\tau \mid a, X)\} = \psi_{j,0}^{\mathrm{RMTL},h}(\tau, a)$$

The proof above also implies $E_0\{\phi_{j,0}^{\mathrm{RMTL}}(\tau, a; \eta_{j,0}^\Lambda) \mid A = a, X\} = \mathrm{RMTL}_{j,0}(\tau \mid a, X)$.

Using Lemma 2 of Westling et al. (2023),

$$|\widehat{\psi}^{\mathrm{RMST},h}(\tau, a) - \psi_0^{\mathrm{RMST},h}(\tau, a)|$$

$$\leq |P_n\{\varphi_\infty^{\mathrm{RMST},h}(\tau, a; \eta_\infty^{\Lambda,h})\}| + \left| \frac{1}{K} \sum_{k=1}^K \frac{Kn_k^{1/2}}{n} \mathcal{G}_{n,k}\{\widehat{\phi}_k^{\mathrm{RMST},h}(\tau, a; \widehat{\eta}_k^{\Lambda,h}) - \phi_\infty^{\mathrm{RMST},h}(\tau, a; \eta_\infty^{\Lambda,h})\} \right|$$

$$+ \left| \frac{1}{K} \sum_{k=1}^K \frac{Kn_k}{n} P_0\{\widehat{\phi}_k^{\mathrm{RMST},h}(\tau, a; \widehat{\eta}_k^{\Lambda,h}) - \phi_\infty^{\mathrm{RMST},h}(\tau, a; \eta_\infty^{\Lambda,h})\} \right|$$

Under Condition 3 and weak law of large numbers, $P_n\{\varphi_\infty^{\mathrm{RMST},h}(t, a; \eta_\infty^{\Lambda,h})\} \xrightarrow{P} 0$. With Condition 1, 2, and Lemma 4, the second term is $o_P(n^{-1/2})$. The third term is bounded by $2[\max_k P_0\{\widehat{\phi}_k^{\mathrm{RMST},h}(\tau, a; \widehat{\eta}_k^{\Lambda,h}) - \phi_\infty^{\mathrm{RMST},h}(\tau, a; \eta_\infty^{\Lambda,h})\}^2]^{1/2}$, which is further bounded based on Lemma 3 and converges to zero in probability. Consequently, $|\widehat{\psi}^{\mathrm{RMST},h}(\tau, a) - \psi_0^{\mathrm{RMST},h}(\tau, a)| = o_P(1)$.

In terms of uniform consistency,

$$\sup_{t \in [0,\tau]} |\widehat{\psi}^{\mathrm{RMST},h}(t, a) - \psi_0^{\mathrm{RMST},h}(t, a)|$$

$$\leq \sup_{t \in [0,\tau]} |P_n\{\varphi_\infty^{\mathrm{RMST},h}(t, a; \eta_\infty^{\Lambda,h})\}| + \sup_{t \in [0,\tau]} \left| \frac{1}{K} \sum_{k=1}^K \frac{Kn_k^{1/2}}{n} \mathcal{G}_{n,k}\{\widehat{\phi}_k^{\mathrm{RMST},h}(t, a; \widehat{\eta}_k^{\Lambda,h}) - \phi_\infty^{\mathrm{RMST},h}(t, a; \eta_\infty^{\Lambda,h})\} \right|$$

$$+ \sup_{t \in [0,\tau]} \left| \frac{1}{K} \sum_{k=1}^K \frac{Kn_k}{n} P_0\{\widehat{\phi}_k^{\mathrm{RMST},h}(t, a; \widehat{\eta}_k^{\Lambda,h}) - \phi_\infty^{\mathrm{RMST},h}(t, a; \eta_\infty^{\Lambda,h})\} \right|$$

With a Donsker $\phi_\infty^{\mathrm{RMST},h}(t, a; \eta_\infty^{\Lambda,h})$, the first term on the RHS is $O_P(n^{-1/2})$. The second term is $o_P(n^{-1/2})$ when Condition 2 and 4 holds. By Lemma 3, the third term converges to zero in probability. Eventually, $\sup_{t \in [0,\tau]} |\widehat{\psi}^{\mathrm{RMST},h}(t, a) - \psi_0^{\mathrm{RMST},h}(t, a)| = o_P(1)$. Similar reasoning applies for $|\widehat{\psi}_j^{\mathrm{RMTL},h}(\tau, a) - \psi_{j,0}^{\mathrm{RMTL},h}(\tau, a)| = o_P(1)$ and $\sup_{t \in [0,\tau]} |\widehat{\psi}_j^{\mathrm{RMTL},h}(t, a) - \psi_{j,0}^{\mathrm{RMTL},h}(t, a)| = o_P(1)$.

Condition 1 and 5 require convergence of estimated nuisance functions to their limiting counterparts in probability, which are used to regulate the empirical process terms from the von Mises expansion of $\widehat{\psi}^{\mathrm{RMST},h}(\tau, a) - \psi_0^{\mathrm{RMST},h}(\tau, a)$ and $\widehat{\psi}_j^{\mathrm{RMTL},h}(\tau, a) - \psi_{j,0}^{\mathrm{RMTL},h}(\tau, a)$. Condition 2 bounds conditional distributions, appeared in the denominators after decomposing $P\{\widehat{\phi}_0^{\mathrm{RMST},h}(\tau, a; \widehat{\eta}^{\Lambda,h}) - \phi_0^{\mathrm{RMST},h}(\tau, a; \eta_0^{\Lambda,h})\}^2$ and $P\{\widehat{\phi}_j^{\mathrm{RMTL},h}(\tau, a; \widehat{\eta}_j^{\Lambda,h}) - \phi_{j,0}^{\mathrm{RMTL},h}(\tau, a; \eta_{j,0}^{\Lambda,h})\}^2$, away from zero in all strata. This condition can be assured empirically by Winsorizing extreme values. Condition 3 and 6 are often referred to as model double robustness where the consistency of the resulting estimators relies only on consistent estimation of a subset of nuisance functions. When $h_0(x)$ is a function of $\pi_0(a \mid x)$, Condition 3 becomes (1) $\pi_\infty(a \mid X) = \pi_0(a \mid X)$ and $S_\infty(t \mid a, X) = S_0(t \mid a, X)$ or (2) $\pi_\infty(a \mid X) = \pi_0(a \mid X)$ and $G_\infty(t \mid a, X) = G_0(t \mid a, X)$ while Condition 6 reduces to (1) $\pi_\infty(a \mid X) = \pi_0(a \mid X)$ and $S_\infty(t \mid a, X) = S_0(t \mid a, X)$ and $F_{j,\infty}(t \mid a, X) = F_{j,0}(t \mid a, X)$ or (2) $\pi_\infty(a \mid X) = \pi_0(a \mid X)$ and $G_\infty(t \mid a, X) = G_0(t \mid a, X)$. When $h_0(x) = 1$, Condition 3 reduce to either (1) $S_\infty(t \mid a, X) = S_0(t \mid a, X)$ or (2) $\pi_\infty(a \mid X) = \pi_0(a \mid X)$ and $G_\infty(t \mid a, X) = G_0(t \mid a, X)$. This condition is similar to consistency conditions of doubly-robust estimators for survival functions on ATE (Zhang & Schaubel, 2012; Bai et al., 2013) etc. When $h_0(x) = 1$, Condition 6 weakens to (1) $S_\infty(t \mid a, X) = S_0(t \mid a, X)$ and $F_{j,\infty}(t \mid a, X) = F_{j,0}(t \mid a, X)$ or (2) $\pi_\infty(a \mid X) = \pi_0(a \mid X)$ and $G_\infty(t \mid a, X) = G_0(t \mid a, X)$ or (3) $F_{j,\infty}(t \mid a, X) = F_{j,0}(t \mid a, X)$ and $G_\infty(t \mid a, X) = G_0(t \mid a, X)$, which corresponds to Theorem 1 for cause-specific CIFs in Ozenne et al. (2020). The Donsker conditions that require the nuisance models have bounded complexity, often a bounded entropy integral, are avoided by cross-fitting (Chernozhukov et al., 2018). ∎

## D.8 PROOF OF THEOREM 3

Additional conditions are required for pointwise and uniform asymptotic linearity.

**Condition 8** *It holds that* $\max_k |[P_{n,k}\{\widehat{w}_k^h(a, X)\}]^{-1} - [E_0\{h_0(X)\}]^{-1}| = o_P(n^{-1/2})$, $\max_k E_0|\widehat{h}_k(X) - h_0(X)| = o_P(n^{-1/2})$.

**Condition 9** *It holds that* $\widehat{r}_1(\tau, a) = o_P(n^{-1/2})$.

**Condition 10** *It holds that* $\widehat{r}_2^{\mathrm{RMST}}(\tau, a) = o_P(n^{-1/2})$.

**Condition 11** *Furthermore,* $\sup_{t \in [0,\tau]} \widehat{r}_1(t, a) = o_P(n^{-1/2})$ *and* $\sup_{t \in [0,\tau]} \widehat{r}_2^{\mathrm{RMST}}(t, a) = o_P(n^{-1/2})$.

**Condition 12** *It holds that* $\widehat{r}_{2,j}^{\mathrm{RMTL}}(\tau, a) = o_P(n^{-1/2})$ *and* $\widehat{r}_{3,j}^{\mathrm{RMTL}}(\tau, a) = o_P(n^{-1/2})$.

**Condition 13** *Furthermore,* $\sup_{t \in [0,\tau]} \widehat{r}_1(t, a) = o_P(n^{-1/2})$, $\sup_{t \in [0,\tau]} \widehat{r}_{2,j}^{\mathrm{RMTL}}(t, a) = o_P(n^{-1/2})$, *and* $\sup_{t \in [0,\tau]} \widehat{r}_{3,j}^{\mathrm{RMTL}}(t, a) = o_P(n^{-1/2})$.

Under Condition 1, 2, 3, 8, 9, 10,

$$\widehat{\psi}^{\mathrm{RMST},h}(\tau, a) = \psi_0^{\mathrm{RMST},h}(\tau, a) + P_n\{\varphi_0^{\mathrm{RMST},h}(\tau, a; \eta_0^{\Lambda,h})\} + o_P(n^{-1/2})$$
$$n^{1/2}\{\widehat{\psi}^{\mathrm{RMST},h}(\tau, a) - \psi_0^{\mathrm{RMST},h}(\tau, a)\} \xrightarrow{D} \mathcal{N}(0, P_0[\{\varphi_0^{\mathrm{RMST},h}(\tau, a; \eta_0^{\Lambda,h})\}^2])$$

If Condition 4 and 11 hold additionally,

$$\sup_{t \in [0,\tau]} \left|\widehat{\psi}^{\mathrm{RMST},h}(t, a) - \psi_0^{\mathrm{RMST},h}(t, a) - P_n\{\varphi_0^{\mathrm{RMST},h}(t, a; \eta_0^{\Lambda,h})\}\right| = o_P(n^{-1/2})$$

such that $n^{1/2}\{\widehat{\psi}^{\mathrm{RMST},h}(t, a) - \psi_0^{\mathrm{RMST},h}(t, a)\}$ ($t \in [0,\tau]$) converges weakly to a tight mean zero Gaussian process with covariance $\sigma_0^{\mathrm{RMST},h}(u, t, a) = P\{\varphi_0^{\mathrm{RMST},h}(u, a; \eta_0^{\Lambda,h})\varphi_0^{\mathrm{RMST},h}(t, a; \eta_0^{\Lambda,h})\}$.

Under Condition 2, 5, 6, 8, 9, 12,

$$\widehat{\psi}_j^{\mathrm{RMTL},h}(\tau, a) = \psi_{j,0}^{\mathrm{RMTL},h}(\tau, a) + P_n\{\varphi_{j,0}^{\mathrm{RMTL},h}(\tau, a; \eta_0^{\Lambda,h})\} + o_P(n^{-1/2})$$
$$n^{1/2}\{\widehat{\psi}_j^{\mathrm{RMTL},h}(\tau, a) - \psi_{j,0}^{\mathrm{RMTL},h}(\tau, a)\} \xrightarrow{D} \mathcal{N}(0, P_0[\{\varphi_{j,0}^{\mathrm{RMTL},h}(\tau, a; \eta_0^{\Lambda,h})\}^2])$$

If Condition 7 and 13 holds additionally,

$$\sup_{t \in [0,\tau]} \left|\widehat{\psi}_j^{\mathrm{RMTL},h}(t, a) - \psi_{j,0}^{\mathrm{RMTL},h}(t, a) - P_n\{\varphi_{j,0}^{\mathrm{RMTL},h}(t, a; \eta_{j,0}^{\Lambda,h})\}\right| = o_P(n^{-1/2})$$

such that $n^{1/2}\{\widehat{\psi}_j^{\mathrm{RMTL},h}(t,a) - \psi_{j,0}^{\mathrm{RMTL},h}(t,a)\}$ ($t \in [0,\tau]$) converges weakly to a tight Gaussian process with mean zero and covariance $\sigma_{j,0}^{\mathrm{RMTL},h}(u,t,a) = P\{\varphi_{j,0}^{\mathrm{RMTL},h}(u,a;\eta_{j,0}^{\Lambda,h})\varphi_{j,0}^{\mathrm{RMTL},h}(t,a;\eta_{j,0}^{\Lambda,h})\}$.

Following Proof of Theorem 2, and when $\eta_\infty^{\Lambda,h} = \eta_0^{\Lambda,h}$, we have

$$\widehat{\psi}^{\mathrm{RMST},h}(\tau,a) - \psi_0^{\mathrm{RMST},h}(\tau,a)$$

$$=P_n\{\varphi_0^{\mathrm{RMST},h}(\tau,a;\eta_0^{\Lambda,h})\} + \frac{1}{K}\sum_{k=1}^{K}\frac{Kn_k^{1/2}}{n}\mathcal{G}_{n,k}\{\widehat{\phi}_k^{\mathrm{RMST},h}(\tau,a;\widehat{\eta}_k^{\Lambda,h}) - \phi_0^{\mathrm{RMST},h}(\tau,a;\eta_0^{\Lambda,h})\}$$

$$+ \frac{1}{K}\sum_{k=1}^{K}\frac{Kn_k}{n}[P_0\{\widehat{\phi}_k^{\mathrm{RMST},h}(\tau,a;\widehat{\eta}_k^{\Lambda,h})\} - \psi_0^{\mathrm{RMST},h}(\tau,a)]$$

Under Condition 1, 2, 4, and Lemma 3,

$$\frac{1}{K}\sum_{k=1}^{K}\frac{Kn_k^{1/2}}{n}\sup_{t\in[0,\tau]}\left|\mathcal{G}_{n,k}\{\widehat{\phi}_k^{\mathrm{RMST},h}(\tau,a;\widehat{\eta}_k^{\Lambda,h}) - \phi_0^{\mathrm{RMST},h}(\tau,a;\eta_0^{\Lambda,h})\}\right| = o_P(n^{-1/2})$$

For the third term on the RHS, we substitute the Duhamel equation

$$\frac{S_0(t- \mid a,X)}{\widehat{S}(t \mid a,X)}\{\mathrm{d}\widehat{\Lambda}(t \mid a,X) - \mathrm{d}\Lambda_0(t \mid a,X)\} = \mathrm{d}\left\{\frac{S_0(t \mid a,X)}{\widehat{S}(t \mid a,X)} - 1\right\}$$

into Lemma 2,

$$P_0\{\widehat{\phi}_k^{\mathrm{RMST},h}(\tau,a;\widehat{\eta}_k^{\Lambda,h})\} - \psi_0^{\mathrm{RMST},h}(\tau,a)$$

$$=[P_{n,k}\{\widehat{w}_k^h(a,X)\}]^{-1}E_0\left[\frac{\widehat{h}_k(X)\pi_0(a \mid X)\widehat{\mathrm{RMST}}_k(\tau \mid a,X)}{\widehat{\pi}_k(a \mid X)}\int_0^\tau\left\{\frac{G_0(t- \mid a,X)}{\widehat{G}_k(t- \mid a,X)} - 1\right\}\mathrm{d}\left\{\frac{S_0(t \mid a,X)}{\widehat{S}_k(t \mid a,X)} - 1\right\}\right]$$

$$+ E_0\left[\widehat{h}_k(X)\left\{\frac{\pi_0(a \mid X)}{\widehat{\pi}_k(a \mid X)} - 1\right\}\widehat{\mathrm{RMST}}_k(\tau \mid a,X)\left\{\frac{S_0(\tau \mid a,X)}{\widehat{S}_k(\tau \mid a,X)} - 1\right\}\right]$$

$$- E_0\left[\frac{\widehat{h}_k(X)\pi_0(a \mid X)}{\widehat{\pi}_k(a \mid X)}\int_0^\tau\widehat{\mathrm{RMST}}_k(t \mid a,X)\left\{\frac{G_0(t- \mid a,X)}{\widehat{G}_k(t- \mid a,X)} - 1\right\}\mathrm{d}\left\{\frac{S_0(t \mid a,X)}{\widehat{S}_k(t \mid a,X)} - 1\right\}\right]$$

$$- E_0\left(\widehat{h}_k(X)\left\{\frac{\pi_0(a \mid X)}{\widehat{\pi}_k(a \mid X)} - 1\right\}\left[\widehat{\mathrm{RMST}}_k(\tau \mid a,X)\left\{\frac{S_0(\tau \mid a,X)}{\widehat{S}_k(\tau \mid a,X)}\right\} - \mathrm{RMST}_0(\tau \mid a,X)\right]\right)$$

$$+ ([P_{n,k}\{\widehat{h}_k(X)\}]^{-1} - [P_{n,k}\{\widehat{w}_k^h(a,X)\}]^{-1})E_0\{\widehat{h}_k(X)\widehat{\mathrm{RMST}}_k(\tau \mid a,X)\}$$

$$+ ([P_{n,k}\{\widehat{w}_k^h(a,X)\}]^{-1} - [E_0\{h_0(X)\}]^{-1})E_0\{\widehat{h}_k(X)\mathrm{RMST}_0(\tau \mid a,X)\}$$

$$+ [E_0\{h_0(X)\}]^{-1}E_0[\{\widehat{h}_k(X) - h_0(X)\}\mathrm{RMST}_0(\tau \mid a,X)]$$

such that

$$\left|P_0\{\widehat{\phi}_k^{\mathrm{RMST},h}(\tau,a;\widehat{\eta}_k^{\Lambda,h})\} - \psi_0^{\mathrm{RMST},h}(\tau,a)\right|$$

$$\leq 2h_{\min}^{-1}h_{\max}\epsilon\tau E_0\left|\int_0^\tau\left\{\frac{G_0(t- \mid a,X)}{\widehat{G}_k(t- \mid a,X)} - 1\right\}\mathrm{d}\left\{\frac{S_0(t \mid a,X)}{\widehat{S}_k(t \mid a,X)} - 1\right\}\right|$$

$$+ h_{\min}^{-1}h_{\max}\epsilon E_0\left|\{\widehat{\pi}_k(a \mid X) - \pi_0(a \mid X)\}\{\widehat{\mathrm{RMST}}_k(\tau \mid a,X) - \mathrm{RMST}_0(\tau \mid a,X)\}\right|$$

$$+ 2h_{\min}^{-1}h_{\max}\tau h_{\max}\tau\left|[P_{n,k}\{\widehat{w}_k^h(a,X)\}]^{-1} - [E_0\{h_0(X)\}]^{-1}\right| + h_{\min}^{-1}\tau E_0|\{\widehat{h}_k(X) - h_0(X)|$$

where we define

$$\widehat{r}_1(\tau,a) = \max_k E_0\left|\int_0^\tau\left\{\frac{G_0(t- \mid a,X)}{\widehat{G}_k(t- \mid a,X)} - 1\right\}\mathrm{d}\left\{\frac{S_0(t \mid a,X)}{\widehat{S}_k(t \mid a,X)} - 1\right\}\right|$$

$$\widehat{r}_2^{\mathrm{RMST}}(\tau,a) = \max_k E_0\left|\{\widehat{\pi}_k(a \mid X) - \pi_0(a \mid X)\}\{\widehat{\mathrm{RMST}}_k(\tau \mid a,X) - \mathrm{RMST}_0(\tau \mid a,X)\}\right|$$

Note that $1/K \sum_{k=1}^{K} Kn_k/n \leq 2$, such that

$$\left| \frac{1}{K} \sum_{k=1}^{K} \frac{Kn_k}{n} [P_0\{\widehat{\phi}_k^{\mathrm{RMST},h}(\tau,a;\widehat{\eta}_k^{\Lambda,h})\} - \psi_0^{\mathrm{RMST},h}(\tau,a)] \right|$$

$$\leq 4h_{\min}^{-1}h_{\max}\epsilon\tau\widehat{r}_1(\tau,a) + 2h_{\min}^{-1}h_{\max}\epsilon\widehat{r}_2^{\mathrm{RMST}}(\tau,a)$$

$$+ 4h_{\min}^{-1}h_{\max}\tau h_{\max}\tau \left| [P_{n,k}\{\widehat{w}_k^h(a,X)\}]^{-1} - [E_0\{h_0(X)\}]^{-1} \right| + 2h_{\min}^{-1}\tau E_0|\{\widehat{h}_k(X) - h_0(X)|$$

By Condition 8, 9, 10, the RHS is $o_P(n^{-1/2})$. Hence $\widehat{\psi}^{\mathrm{RMST},h}(\tau,a) = \psi_0^{\mathrm{RMST},h}(\tau,a) + P_n\{\varphi_0^{\mathrm{RMST},h}(\tau,a;\eta_0^{\Lambda,h})\} + o_P(n^{-1/2})$. Due to $P_0\{\varphi_0^{\mathrm{RMST},h}(\tau,a;\eta_0^{\Lambda,h})\} = 0$, $P_0[\{\varphi_0^{\mathrm{RMST},h}(\tau,a;\eta_0^{\Lambda,h})\}]^2 < \infty$, and its uniform boundness, $n^{1/2}P_n\{\varphi_0^{\mathrm{RMST},h}(\tau,a;\eta_0^{\Lambda,h})\} \xrightarrow{D} \mathcal{N}(0, P_0[\{\varphi_0^{\mathrm{RMST},h}(\tau,a;\eta_0^{\Lambda,h})\}^2])$.

When Condition 11 holds

$$\sup_{t\in[0,\tau]} \left| \frac{1}{K} \sum_{k=1}^{K} \frac{Kn_k}{n} [P_0\{\widehat{\phi}_k^{\mathrm{RMST},h}(t,a;\widehat{\eta}_k^{\Lambda,h})\} - \psi_0^{\mathrm{RMST},h}(t,a)] \right|$$

$$\leq 4h_{\min}^{-1}h_{\max}\epsilon\tau \sup_{t\in[0,\tau]} \widehat{r}_1(t,a) + 2h_{\min}^{-1}h_{\max}\epsilon \sup_{t\in[0,\tau]} \widehat{r}_2^{\mathrm{RMST}}(t,a)$$

$$+ 4h_{\min}^{-1}h_{\max}\tau h_{\max}\tau \left| [P_{n,k}\{\widehat{w}_k^h(a,X)\}]^{-1} - [E_0\{h_0(X)\}]^{-1} \right| + 2h_{\min}^{-1}\tau E_0|\{\widehat{h}_k(X) - h_0(X)| = o_P(n^{-1/2})$$

As a result,

$$\sup_{t\in[0,\tau]} \left| \widehat{\psi}^{\mathrm{RMST},h}(t,a) - \psi_0^{\mathrm{RMST},h}(t,a) - P_n\{\varphi_0^{\mathrm{RMST},h}(t,a;\eta_0^{\Lambda,h})\} \right| = o_P(n^{-1/2})$$

As $\varphi_0^{\mathrm{RMST},h}(t,a;\eta_0^{\Lambda,h})$ is uniformly bounded for $t \in [0,\tau]$, then $n^{1/2}\{P_n\varphi_0^{\mathrm{RMST},h}(t,a;\eta_0^{\Lambda,h})\}$ converges weakly to a tight Gaussian process with mean zero and covariance $\sigma_0^{\mathrm{RMST},h}(u,t,a) = P_0\{\varphi_0^{\mathrm{RMST},h}(u,a;\eta_0^{\Lambda,h})\varphi_0^{\mathrm{RMST},h}(t,a;\eta_0^{\Lambda,h})\}$.

We plug the Duhamel equation into the competing risks setting of Lemma 2,

$$P_0\{\widehat{\phi}_{j,k}^{\mathrm{RMTL},h}(\tau,a;\widehat{\eta}_{j,k}^{\Lambda,h})\} - \psi_{j,0}^{\mathrm{RMTL},h}(\tau,a)$$

$$=[P_{n,k}\{\widehat{w}_k^h(a,X)\}]^{-1}\left(-E_0\left[\frac{\widehat{h}_k(X)\pi_0(a\mid X)\tau}{\widehat{\pi}_k(a\mid X)}\int_0^\tau S_0(t\mid a,X)\left\{\frac{G_0(t-\mid a,X)}{\widehat{G}_k(t-\mid a,X)}-1\right\}\right.\right.$$

$$\times\{\mathrm{d}\widehat{\Lambda}_{j,k}(t\mid a,X)-\mathrm{d}\Lambda_{j,0}(t\mid a,X)\}\bigg] - E_0\left[\widehat{h}_k(X)\left\{\frac{\pi_0(a\mid X)}{\widehat{\pi}_k(a\mid X)}-1\right\}\right.$$

$$\times\tau\int_0^\tau S_0(t\mid a,X)\{\mathrm{d}\widehat{\Lambda}_{j,k}(t\mid a,X)-\mathrm{d}\Lambda_{j,0}(t\mid a,X)\}\bigg]$$

$$+ E_0\left[\frac{\widehat{h}_k(X)\pi_0(a\mid X)}{\widehat{\pi}_k(a\mid X)}\int_0^\tau tS_0(t\mid a,X)\left\{\frac{G_0(t-\mid a,X)}{\widehat{G}_k(t-\mid a,X)}-1\right\}\{\mathrm{d}\widehat{\Lambda}_{j,k}(t\mid a,X)-\mathrm{d}\Lambda_{j,0}(t\mid a,X)\}\right]$$

$$+ E_0\left[\widehat{h}_k(X)\left\{\frac{\pi_0(a\mid X)}{\widehat{\pi}_k(a\mid X)}-1\right\}\int_0^\tau tS_0(t\mid a,X)\{\mathrm{d}\widehat{\Lambda}_{j,k}(t\mid a,X)-\mathrm{d}\Lambda_{j,0}(t\mid a,X)\}\right]$$

$$+ E_0\left[\frac{\widehat{h}_k(X)\pi_0(a\mid X)\widehat{\mathrm{RMTL}}_{j,k}(\tau\mid a,X)}{\widehat{\pi}_k(a\mid X)}\int_0^\tau\left\{\frac{G_0(t-\mid a,X)}{\widehat{G}_k(t-\mid a,X)}-1\right\}\mathrm{d}\left\{\frac{S_0(t\mid a,X)}{\widehat{S}_k(t\mid a,X)}-1\right\}\right]$$

$$+ E_0\left[\widehat{h}_k(X)\left\{\frac{\pi_0(a\mid X)}{\widehat{\pi}_k(a\mid X)}-1\right\}\widehat{\mathrm{RMTL}}_{j,k}(\tau\mid a,X)\left\{\frac{S_0(\tau\mid a,X)}{\widehat{S}_k(\tau\mid a,X)}-1\right\}\right]$$

$$- E_0\left[\frac{\widehat{h}_k(X)\pi_0(a\mid X)}{\widehat{\pi}_k(a\mid X)}\int_0^\tau\widehat{\mathrm{RMTL}}_{j,k}(t\mid a,X)\left\{\frac{G_0(t-\mid a,X)}{\widehat{G}(t-\mid a,X)}-1\right\}\mathrm{d}\left\{\frac{S_0(t\mid a,X)}{\widehat{S}_k(t\mid a,X)}-1\right\}\right]$$

$$- E_0\left(\widehat{h}_k(X)\left\{\frac{\pi_0(a\mid X)}{\widehat{\pi}_k(a\mid X)}-1\right\}\left[\widehat{\mathrm{RMTL}}_{j,k}(\tau\mid a,X)\left\{\frac{S_0(\tau\mid a,X)}{\widehat{S}_k(\tau\mid a,X)}\right\}-\int_0^\tau\frac{S_0(t\mid a,X)}{\widehat{S}_k(t\mid a,X)}\widehat{F}_{j,k}(t\mid a,X)\mathrm{d}t\right]\right)$$

$$- E_0\left[\frac{\widehat{h}_k(X)\pi_0(a\mid X)\tau}{\widehat{\pi}_k(a\mid X)}\int_0^\tau\widehat{F}_{j,k}(t\mid a,X)\left\{\frac{G_0(t-\mid a,X)}{\widehat{G}_k(t-\mid a,X)}-1\right\}\mathrm{d}\left\{\frac{S_0(t\mid a,X)}{\widehat{S}_k(t\mid a,X)}-1\right\}\right]$$

$$- E_0\left[\widehat{h}_k(X)\left\{\frac{\pi_0(a\mid X)}{\widehat{\pi}_k(a\mid X)}-1\right\}\tau\left\{\frac{\widehat{F}_{j,k}(\tau\mid a,X)S_0(\tau\mid a,X)}{\widehat{S}_k(\tau\mid a,X)}-\int_0^\tau S_0(\tau\mid a,X)\mathrm{d}\widehat{\Lambda}_{j,k}(t\mid a,X)\right\}\right]$$

$$+ E_0\left[\frac{\widehat{h}_k(X)\pi_0(a\mid X)}{\widehat{\pi}_k(a\mid X)}\int_0^\tau t\widehat{F}_{j,k}(t\mid a,X)\left\{\frac{G_0(t-\mid a,X)}{\widehat{G}_k(t-\mid a,X)}-1\right\}\mathrm{d}\left\{\frac{S_0(t\mid a,X)}{\widehat{S}_k(t\mid a,X)}-1\right\}\right]$$

$$+ E_0\left[\widehat{h}_k(X)\left\{\frac{\pi_0(a\mid X)}{\widehat{\pi}_k(a\mid X)}-1\right\}\left\{\frac{\widehat{F}_{j,k}(\tau\mid a,X)S_0(\tau\mid a,X)\tau}{\widehat{S}_k(\tau\mid a,X)}-\int_0^\tau\frac{S_0(t\mid a,X)}{\widehat{S}_k(t\mid a,X)}\widehat{F}_{j,k}(t\mid a,X)\mathrm{d}t\right.\right.$$

$$\left.\left.-\int_0^\tau tS_0(\tau\mid a,X)\mathrm{d}\widehat{\Lambda}_{j,k}(t\mid a,X)\right\}\right]$$

$$+ ([P_{n,k}\{\widehat{h}_k(X)\}]^{-1}-[P_{n,k}\{\widehat{w}_k^h(a,X)\}]^{-1})E_0\{\widehat{h}_k(X)\widehat{\mathrm{RMTL}}_{j,k}(\tau\mid a,X)\}$$

$$+ ([P_{n,k}\{\widehat{w}_k^h(a,X)\}]^{-1}-[E_0\{h_0(X)\}]^{-1})E_0\{\widehat{h}_k(X)\mathrm{RMTL}_{j,0}(\tau\mid a,X)\}$$

$$+ [E_0\{h_0(X)\}]^{-1}E_0[\{\widehat{h}_k(X)-h_0(X)\}\mathrm{RMTL}_{j,0}(\tau\mid a,X)]$$

After canceling out common terms, we have

$$P_0\{\widehat{\phi}_{j,k}^{\mathrm{RMTL},h}(\tau,a;\widehat{\eta}_{j,k}^{\Lambda,h})\} - \psi_{j,0}^{\mathrm{RMTL},h}(\tau,a)$$

$$=[P_{n,k}\{\widehat{w}_k^h(a,X)\}]^{-1}\left(-E_0\left[\frac{\widehat{h}_k(X)\pi_0(a\mid X)}{\widehat{\pi}_k(a\mid X)}\int_0^\tau (\tau-t)S_0(t\mid a,X)\left\{\frac{G_0(t-\mid a,X)}{\widehat{G}_k(t-\mid a,X)}-1\right\}\right.$$

$$\times\{\mathrm{d}\widehat{\Lambda}_{j,k}(t\mid a,X)-\mathrm{d}\Lambda_{j,0}(t\mid a,X)\}\right] + E_0\left[\frac{\widehat{h}_k(X)\pi_0(a\mid X)}{\widehat{\pi}_k(a\mid X)}\int_0^\tau\{\widehat{\mathrm{RMTL}}_{j,k}(\tau\mid a,X)-\widehat{\mathrm{RMTL}}_{j,k}(t\mid a,X)\}\right.$$

$$\times\left\{\frac{G_0(t-\mid a,X)}{\widehat{G}_k(t-\mid a,X)}-1\right\}\mathrm{d}\left\{\frac{S_0(t\mid a,X)}{\widehat{S}_k(t\mid a,X)}-1\right\}\right] - E_0\left[\frac{\widehat{h}_k(X)\pi_0(a\mid X)}{\widehat{\pi}_k(a\mid X)}\int_0^\tau(\tau-t)\widehat{F}_{j,k}(t\mid a,X)\right.$$

$$\times\left\{\frac{G_0(t-\mid a,X)}{\widehat{G}_k(t-\mid a,X)}-1\right\}\mathrm{d}\left\{\frac{S_0(t\mid a,X)}{\widehat{S}_k(t\mid a,X)}-1\right\}\right] + E_0\left[\widehat{h}_k(X)\left\{\frac{\pi_0(a\mid X)}{\widehat{\pi}_k(a\mid X)}-1\right\}\right.$$

$$\times\{\widehat{\mathrm{RMTL}}_{j,k}(\tau\mid a,X)-\mathrm{RMTL}_{j,0}(\tau\mid a,X)\}\right]$$

$$+ ([P_n\{\widehat{h}_k(X)\}]^{-1}-[P_n\{\widehat{w}_k^h(a,X)\}]^{-1})E_0\{\widehat{h}_k(X)\mathrm{R}\widehat{\mathrm{MTL}}_{j,k}(\tau\mid a,X)\}$$

$$+ ([P_{n,k}\{\widehat{w}_k^h(a,X)\}]^{-1}-[E_0\{h_0(X)\}]^{-1})E_0\{\widehat{h}_k(X)\mathrm{RMTL}_j(\tau\mid a,X)\}$$

$$+ [E_0\{h_0(X)\}]^{-1}E_0[\{\widehat{h}_k(X)-h_0(X)\}\mathrm{RMTL}_{j,0}(\tau\mid a,X)]$$

Finally,

$$\left|P_0\{\widehat{\phi}_{j,k}^{\mathrm{RMTL},h}(\tau,a;\widehat{\eta}_{j,k}^{\Lambda,h})\} - \psi_{j,0}^{\mathrm{RMTL},h}(\tau,a)\right|$$

$$=2h_{\min}^{-1}h_{\max}\epsilon\tau E_0\left|\int_0^\tau\left\{\frac{G_0(t-\mid a,X)}{\widehat{G}_k(t-\mid a,X)}-1\right\}\{\mathrm{d}\widehat{\Lambda}_{j,k}(t\mid a,X)-\mathrm{d}\Lambda_{j,0}(t\mid a,X)\}\right|$$

$$+ 4h_{\min}^{-1}h_{\max}\epsilon\tau E_0\left|\int_0^\tau\left\{\frac{G_0(t-\mid a,X)}{\widehat{G}_k(t-\mid a,X)}-1\right\}\mathrm{d}\left\{\frac{S_0(t\mid a,X)}{\widehat{S}_k(t\mid a,X)}-1\right\}\right|$$

$$+ h_{\min}^{-1}h_{\max}\epsilon E_0\left|\{\widehat{\pi}_k(a\mid X)-\pi_0(a\mid X)\}\{\widehat{\mathrm{RMTL}}_{j,k}(\tau\mid a,X)-\mathrm{RMTL}_{j,0}(\tau\mid a,X)\}\right|$$

$$+ 2h_{\min}^{-1}h_{\max}\tau h_{\max}\tau\left|[P_{n,k}\{\widehat{w}_k^h(a,X)\}]^{-1}-[E_0\{h_0(X)\}]^{-1}\right| + h_{\min}^{-1}\tau E_0|\{\widehat{h}_k(X)-h_0(X)|$$

where we define

$$\widehat{r}_{2,j}^{\mathrm{RMTL}}(\tau,a) = \max_k E_0\left|\{\widehat{\pi}_k(a\mid X)-\pi_0(a\mid X)\}\{\widehat{\mathrm{RMTL}}_{j,k}(\tau\mid a,X)-\mathrm{RMTL}_{j,0}(\tau\mid a,X)\}\right|$$

$$\widehat{r}_{3,j}^{\mathrm{RMTL}}(\tau,a) = \max_k E_0\left|\int_0^\tau\left\{\frac{G_0(t-\mid a,X)}{\widehat{G}_k(t-\mid a,X)}-1\right\}\{\mathrm{d}\widehat{\Lambda}_{j,k}(t\mid a,X)-\mathrm{d}\Lambda_{j,0}(t\mid a,X)\}\right|$$

Remaining proofs are similar to that of $P_0\{\widehat{\phi}_k^{\mathrm{RMST},h}(\tau,a;\widehat{\eta}_k^{\Lambda,h})\} - \psi_0^{\mathrm{RMST},h}(\tau,a)$.

Regarding efficiency bounds of the proposed estimators, the conditional variances of $\widehat{\psi}^{\mathrm{RMST},h}(\tau)$ is

$$\mathrm{var}\{\widehat{\psi}^{\mathrm{RMST},h}(\tau)\mid O_n\}$$

$$=\frac{1}{n}[P_n\{\widehat{h}(X)\}]^{-2}P_n[\widehat{h}^2(X)\,\mathrm{var}\{\widehat{\mathrm{RMST}}(\tau\mid 1,X)\mid O_n\}] + \frac{1}{n}[P_n\{\widehat{w}^h(1,X)\}]^{-2}$$

$$\times P_n\left[\frac{\widehat{h}^2(X)I(A=1)}{\widehat{\pi}^2(1\mid X)}\,\mathrm{var}\left\{-\int_0^{\tau\wedge\widetilde{T}}\frac{\{\widehat{\mathrm{RMST}}(\tau\mid 1,X)-\widehat{\mathrm{RMST}}(t\mid 1,X)\}\mathrm{d}\widehat{M}(t\mid 1,X)}{\widehat{S}(t\mid 1,X)\widehat{G}(t-\mid 1,X)}\mid O_n\right\}\right]$$

$$+\frac{1}{n}[P_n\{\widehat{h}(X)\}]^{-2}P_n[\widehat{h}^2(X)\,\mathrm{var}\{\widehat{\mathrm{RMST}}(\tau\mid 0,X)\mid O_n\}] + \frac{1}{n}[P_n\{\widehat{w}^h(0,X)\}]^{-2}$$

$$\times P_n\left[\frac{\widehat{h}^2(X)I(A=0)}{\widehat{\pi}^2(0\mid X)}\,\mathrm{var}\left\{-\int_0^{\tau\wedge\widetilde{T}}\frac{\{\widehat{\mathrm{RMST}}(\tau\mid 0,X)-\widehat{\mathrm{RMST}}(t\mid 0,X)\}\mathrm{d}\widehat{M}(t\mid 0,X)}{\widehat{S}(t\mid 0,X)\widehat{G}(t-\mid 0,X)}\mid O_n\right\}\right]$$

By the Slutsky's theorem,

$$n \times \text{var}\{\widehat{\psi}^{\text{RMST},h}(\tau) \mid O_n\}$$

$$\to \left\{\int h_0(x)p_0(x)\mathcal{F}(dx)\right\}^{-2} \int \left[\frac{\text{var}\{\phi_0^{\text{RMST}}(\tau,1;\eta_0^{\Lambda}) \mid x\}}{\pi_0(1 \mid x)} + \frac{\text{var}\{\phi_0^{\text{RMST}}(\tau,0;\eta_0^{\Lambda}) \mid x\}}{\pi_0(0 \mid x)}\right] h_0^2(x)p_0(x)\mathcal{F}(dx)$$

where $p(x)$ is the probability density function of $X$ with respect to a measure $\mathcal{F}(\cdot)$. Assuming untilted uncentered true IFs are conditionally homoskedastic, i.e., $\text{var}\{\phi_0^{\text{RMST}}(\tau,1;\eta_0^{\Lambda}) \mid x\} = \text{var}\{\phi_0^{\text{RMST}}(\tau,0;\eta_0^{\Lambda}) \mid x\} = C_{\text{var}}^{\text{RMST}}$, we have

$$n \times \text{var}\{\widehat{\psi}^{\text{RMST},h}(\tau) \mid O_n\} \to C_{\text{var}}^{\text{RMST}} \left\{\int h_0(x)p_0(x)\mathcal{F}(dx)\right\}^{-2} \int \left[\frac{h_0^2(x)p_0(x)\mathcal{F}(dx)}{\pi_0(1 \mid x)\pi_0(0 \mid x)}\right]$$

Following the Cauchy-Schwarz inequality,

$$\left\{\int h_0(x)p_0(x)\mathcal{F}(dx)\right\}^2 = \left\{\int \frac{h_0(x)}{\sqrt{\pi_0(1 \mid x)\pi_0(0 \mid x)}} \sqrt{\pi_0(1 \mid x)\pi_0(0 \mid x)}p_0(x)\mathcal{F}(dx)\right\}^2$$

$$\leq \int \frac{h_0^2(x)}{\pi_0(1 \mid x)\pi_0(0 \mid x)}p_0(x)\mathcal{F}(dx) \times \int \pi_0(1 \mid x)\pi_0(0 \mid x)p_0(x)\mathcal{F}(dx)$$

The equal sign holds if and only if $h_0(x)/\{\sqrt{\pi_0(1 \mid x)\pi_0(0 \mid x)}\} \propto \sqrt{\pi_0(1 \mid x)\pi_0(0 \mid x)}$, identically $h_0(x) \propto \pi_0(1 \mid x)\pi_0(0 \mid x) \propto \text{var}(a \mid x)$. This is when the proposed estimators reach the semiparametric efficiency bounds for the WATE under conditional homoscedasticity of untilted uncentered IFs. The proof for $\text{var}\{\widehat{\psi}_j^{\text{RMTL},h}(\tau) \mid O_n\}$ is similar. ■

Note that when $h_0(x) = \pi(1 \mid x)\pi(0 \mid x)$, we have

$$\psi_0^{\text{RMST,ATO}}(\tau) = \frac{E_0[\text{cov}\{A, \min(T,\tau) \mid X\}]}{E_0\{\text{var}(A \mid X)\}}$$

and

$$\psi_{j,0}^{\text{RMTL,ATO}}(\tau) = \frac{E_0(\text{cov}[A, \{\tau - \min(T,\tau)\}I(J=j) \mid X])}{E_0\{\text{var}(A \mid X)\}}$$

Vansteelandt & Dukes (2022) discussed adaptive estimation of a partially linear model for continuous and binary outcomes. However, as the survival time is subject to censoring, the structural nested model requires additional semiparametric assumptions on the restricted survival time (Hagiwara et al., 2020) and hence is not pursued in this paper.