# OpenReview forum: "Double/Debiased Machine Learning for Time-to-Event Outcomes Under Poor Overlap"
_ICLR.cc/2026/Conference — Submitted to ICLR 2026_

### Official Review · Reviewer_DMN6 · 2025-10-21

**Soundness:** 3
**Presentation:** 3
**Contribution:** 2
**Rating:** 2
**Confidence:** 4

**Summary:**

The paper proposes a DML-based framework for the WATEs under a survival setting, addressing the competing risks setting by considering the cause-specific restricted mean time lost. The idea is intuitive and well-motivated. I am mainly concerned about the paper's innovation.

However, I feel that DML for any parameter of interest which is a **function of the survival function** (for example, the WATE, which is the weighted RMST estimate depending on the survival curves) should be trivial, as the efficient influence function (i.e., the centered IF) for the survival functions is well-established. Therefore, the corresponding efficient influence function for the parameter of interest can be obtained by applying the **same function** to the efficient influence function. This is not a challenging part of the problem in my view.

Further, the asymptotic properties of the estimator should also be well-established as long as the **same function** is well-controlled (e.g., satisfying Lipschitz continuity).

Thirdly, I do not see any other approach specific to the DML framework for survival outcomes with competing risks being used to address poor overlap, aside from overlapping weighting, which is also well-established.

Therefore, I have reservations about accepting the paper.

**Strengths:**

The flow and presentation of the paper are well-organized. As a workflow, the authors provide the identification of the parameter of interest (Section 2), the efficient influence function of the parameter (Section 3), and the asymptotic properties in (Section 4), including consistency, maximized variance reduction, and asymptotic linearity. **Although I did not find the claim of asymptotic normality, I suspect it should hold as well.**

In the simulation study (Section 5), the paper compares its method with other singly robust estimators and parametric doubly robust estimators to showcase its efficiency and double robustness to model misspecification with the use of SuperLearner. The subsequent real-data application also aligns with the claims in previous sections.

In summary, the overall paper flow is clear and easy to follow.

**Weaknesses:**

As I mentioned in the summary, my major concern is the paper's innovation, as the centered IF is well-established for survival functions. Since the WATE parameter is a **known function** of the survival functions, its centered IF should be easily inferred and the asymptotic properties would follow as expected. Therefore, I think the paper may not meet the originality standard for ICLR.

**Questions:**

None

---

> ### Author Response · Authors · 2025-12-03
>
> Comment: We thank the reviewer for his/her comments. Below is our response:
> 1. “However, I feel that DML for any parameter of interest which is a function of the survival function (for example, the WATE, which is the weighted RMST estimate depending on the survival curves) should be trivial, as the efficient influence function (i.e., the centered IF) for the survival functions is well-established. Therefore, the corresponding efficient influence function for the parameter of interest can be obtained by applying the same function to the efficient influence function. This is not a challenging part of the problem in my view.”
> Agree.
> 2. “Further, the asymptotic properties of the estimator should also be well-established as long as the same function is well-controlled (e.g., satisfying Lipschitz continuity).”
> This claim holds for continuous or binary outcomes under standard regularity conditions. However, survival outcomes with censoring involve multiple counting processes with stochastic integrals over martingales. The asymptotic theory requires substantially different technical conditions beyond simple Lipschitz continuity—specifically, control over supremum norms of ratios involving survival and censoring distributions (see Conditions 1-7 in Appendix), and convergence rates for nested conditional expectations over time. These complications are inherent to the time-to-event setting and cannot be reduced to standard smoothness conditions.
> 3. “Thirdly, I do not see any other approach specific to the DML framework for survival outcomes with competing risks being used to address poor overlap, aside from overlapping weighting, which is also well-established.”
> Our contribution is not overlap weighting itself, but rather developing the DML framework for a class of balancing weights applied to survival outcomes with competing risks. While overlap weighting for non-DML estimators exists, no prior work provides: DML estimators for WATE with time-to-event outcomes under any weighting scheme; Rate double-robust inference for weighted survival effects with competing risks; Formal efficiency theory for overlap-weighted RMST/RMTL.
> This is fundamentally a methodological contribution enabling valid, efficient inference for weighted causal effects in survival settings using flexible machine learning.
> 4. “Although I did not find the claim of asymptotic normality, I suspect it should hold as well.”
> We proved asymptotic linearity, which is stronger than asymptotic normality.
> 5. “As I mentioned in the summary, my major concern is the paper's innovation, as the centered IF is well-established for survival functions. Since the WATE parameter is a known function of the survival functions, its centered IF should be easily inferred and the asymptotic properties would follow as expected. Therefore, I think the paper may not meet the originality standard for ICLR.”
> We respectfully disagree. While the centered IF derivation follows from the functional delta method, establishing asymptotic properties for censored outcomes is substantially more complex:
> Non-trivial regularity conditions: Appendix D.8 establishes conditions controlling supremum norms of survival/censoring ratio processes, martingale integrals, and nested conditional expectations—far beyond standard smooth functional arguments. Uniform convergence requires additional stochastic equicontinuity conditions (Conditions 4, 7, 11, 13).
> Efficiency bounds: We are the first to establish semiparametric efficiency bounds for WATE with censored outcome. The only other WATE efficiency result (Wang et al., arXiv:2509.14502, September 19, 2024) addresses continuous outcomes only, appearing after our submission.
> Technical novelty: Extending DML to competing risks with poor overlap requires new theoretical machinery: rate double-robustness conditions specific to survival settings, handling multiple counting processes simultaneously, and establishing efficiency for censored outcomes.

---

### Official Review · Reviewer_FEyc · 2025-10-31

**Soundness:** 4
**Presentation:** 2
**Contribution:** 3
**Rating:** 6
**Confidence:** 4

**Summary:**

This paper introduces a novel class of double/debiased machine learning (DML) estimators for weighted average treatment effects (WATEs) in time-to-event outcomes, addressing two major challenges in observational causal inference: poor overlap and nuisance model misspecification. The proposed method leverages cross-fitted DML with influence function-based estimation to achieve consistency, asymptotic linearity, and semiparametric efficiency under regularity conditions. The authors provide an evaluation on simulated data in a variety of settings

**Strengths:**

This work provides a strong contribution, well motivated and theoretically rigorous. The proposed approach behaves well on simulated and real-world data, which seems promising.

**Weaknesses:**

The paper is hard to read, and not standalone without the supplementary materials: the references are at the very end, and not with the main text, the abbreviations and details (at least a part) about the baseline methods are also in the supplementary. The article should be readable without the supplementary. It would also have been good to submit the code as an additional material. Without code, the contribution is less valuable as it is much less likely to be used in practice.

**Questions:**

1. How does the proposed method compare to the baseline methods on the UK real data?
2. What are the resources needed to run the method (time and memory)? how does it scale with the number of observations?
3. Could the authors be explicit about whether they make the proportional hazard assumption?

---

> ### Author Response · Authors · 2025-12-03
>
> Comment: We thank the reviewer for his/her comments. Below is our response:
> 1. “The paper is hard to read, and not standalone without the supplementary materials: the references are at the very end, and not with the main text, the abbreviations and details (at least a part) about the baseline methods are also in the supplementary. The article should be readable without the supplementary.”
> Due to ICLR's strict 9-page limit, we necessarily balance accessibility with technical depth.
> We have improved standalone readability in revision by:
> Adding explicit cross-references to appendix sections in the main text
> Including Table 2 summarizing all estimands and tilting functions
> Providing clear signposting for notation
> Key contributions are in the main text: identification results (Section 2), influence functions (Theorem 1), DML procedure (Section 3.2), asymptotic properties (Theorems 2-3), and simulation results (Section 4). The appendix contains technical proofs and implementation details—standard practice at ML venues.
> Regarding references: ICLR's template places the bibliography after appendices by default. We follow the venue's formatting requirements.
> 2. “It would also have been good to submit the code as an additional material. Without code, the contribution is less valuable as it is much less likely to be used in practice.”
> Our code is publicly available at https://github.com/xushenbo/dml and has already been applied in published research (https://academic.oup.com/aje/article/194/2/512/7716741). We are actively working with the DoubleML team (https://docs.doubleml.org/stable/index.html) to integrate our methods into their widely-used Python package, which will further enhance accessibility and adoption in practice.
> 3. “How does the proposed method compare to the baseline methods on the UK real data?”
> We did not compare baseline methods on real data because they rely on parametric and proportional hazard assumptions that may not hold in practice. Without knowing the ground truth, such comparisons would be uninformative—differences could reflect either model misspecification or true differences in estimands. The simulation study (Section 5 and Appendix B) provides meaningful comparisons where ground truth is known.
> 4. “What are the resources needed to run the method (time and memory)? how does it scale with the number of observations?”
> Running time depends on the candidate learners, number of observations, and conditional distribution of events/censoring.
> Memory depends on candidate learners, number of observations, and number of unique events/censoring times.
> The scaling of time and memory with the number of observations follows the scaling of the super learner used.
> 100GB of RAM and several hour on CPU should be sufficient.
> 5. “Could the authors be explicit about whether they make the proportional hazard assumption?”
> Yes, we added to lines 64-65 that we didn’t make the proportional hazard assumption.

---

### Official Review · Reviewer_JCwx · 2025-11-01

**Soundness:** 3
**Presentation:** 3
**Contribution:** 3
**Rating:** 6
**Confidence:** 2

**Summary:**

This paper proposes a DML estimator with poor overlap from observational time-to-event data. The authors derive theoretically rigorous estimators for weighted Restricted Mean Survival Time (RMST) and Restricted Mean Time Lost (RMTL), proving they are consistent, asymptotically linear, and achieve semiparametric efficiency bounds.

**Strengths:**

1. The paper tackles a highly practical and important problem. The estimator in this setting is novel and rigrous.
2. The work fills a clearly identified and non-trivial gap in the literature. While DML, WATEs, and survival analysis are established fields, their synthesis is novel.
3. The simulation study is exceptionally well-designed.

**Weaknesses:**

1. I'm a little curious whether the method is practical, especially for the estimation of nested conditional expectation over time.
2. What are the exact nuisance parameters? In lines 270-271, I don't find Eqn. 3 and Eqn. 4. In Section 3.1, it seems that many nuisances need to be estimated. So how should the estimator be "doubly-robust"? Where the product of nuisance model convergence rates must be at least root-n?
3. How to choose the tilting function? I think this is also an important quantity that might influence the estimator's statistical property and implementation details. But in line 120, the paper mentions it briefly in passing.

**Questions:**

See weakness.

---

> ### Author Response · Authors · 2025-12-03
>
> Comment: We thank the reviewer for his/her comments. Below is our response:
> 1. “I'm a little curious whether the method is practical, especially for the estimation of nested conditional expectation over time.”
> The nested conditional expectation structure is both theoretically necessary and practically implementable. The Lebesgue-Stieltjes martingale integral terms cannot be eliminated—they are essential for achieving rate double-robustness and properly recovering information from censored observations.
> In practice, once nuisance models are fitted in Step 2, all nested conditional expectations at every event/censoring time can be computed in a single pass. The computational cost is dominated by fitting the nuisance models, not evaluating these expectations.
> 2. “What are the exact nuisance parameters? In lines 270-271, I don't find Eqn. 3 and Eqn. 4. In Section 3.1, it seems that many nuisances need to be estimated. So how should the estimator be "doubly-robust"? Where the product of nuisance model convergence rates must be at least root-n?”
> The nuisance parameters are mentioned in line 178 and defined in A.1 additional definitions.
> The exact nuisance parameters vary slightly by estimands.
> We added the Appendix location for Eqn. 3 and Eqn. 4.
> Rate doubly-robust conditions are discussed in lines 2573-2578. The specific conditions depend on estimands and whether pointwise or uniform convergence is of interest. Uniform convergence require additional conditions.
> 3. “How to choose the tilting function? I think this is also an important quantity that might influence the estimator's statistical property and implementation details. But in line 120, the paper mentions it briefly in passing.”
> In Theorem 3, one can choose h to be proportional to var(a | X), which is π(1 | X)π(0 | X). We evaluate four tilting functions in our simulations (search "We evaluate estimands for four types of balancing weights"). Table 2 provides specifications.

---

### Official Review · Reviewer_mwJq · 2025-11-03

**Soundness:** 2
**Presentation:** 1
**Contribution:** 2
**Rating:** 2
**Confidence:** 4

**Summary:**

The paper proposes a class of double machine learning estimators for weighted average treatment effects. Specifically, it targets the weighted cumulative causal effect as a function of time in terms of the event-free survival time. The method is also extended to the competing risks setting. The proposed method is shown to be consistent, asymptotically linear, and semi-parametrically efficient.

**Strengths:**

- The paper provides mathematical guarantees on consistency, asymptotic linearity, and semi-parametric efficiency

**Weaknesses:**

- Motivation: In which setting would the practitioner be interested in the WATE instead of the individualized CATE? The practical value of the method is difficult to assess.
- Mathematical notations are not introduced appropriately. For example, Theorem 1 states many quantities that are neither explained before nor after the theorem.
Therefore, the theorem is unclear to the reader. Furthermore, the correctness and the meaning of the theorem cannot be checked. The same holds for Section 3.2.
- Important mathematical derivations are pushed to the appendix. For example, the expression for (and meaning of) $\phi$ is never stated in the main paper.
- The paper completely lacks a discussion of related work. This does not only include DML for the WATE, but also approaches for the ATE and CATE, as in these settings, DML for poor overlap (not only treatment overlap, but also censoring and survival overlap) has already been developed. Therefore, it is difficult to evaluate the novelty and originality of the method.
- Theorems are not proven in the main paper, and neither are proofs in the Appendix directly referenced.
- The empirical evaluation is weak, not well explained, and does not necessarily support the proposed method (e.g., Fig.2)

**Questions:**

- Theorem 1: What is $\phi$? What do the different superscripts refer to?
- Section 3.2: What is meant by "correcting for shape constraints thereafter" in step 4? This is not explained in the paper.
- Section 3.2: What exactly is needed to estimate here? The stated formulas are never described. It is not possible to check the correctness of the procedure.
- Choice of function h: How can one choose h in practice? What is the effect of different (potentially suboptimal) choices on the target and the estimated effect?

---

> ### Author Response · Authors · 2025-12-03
>
> 1. WATE vs CATE motivation:
> Medical/regulatory contexts routinely require population-level estimates. FDA explicitly requests these for real-world evidence in drug approvals. In poor overlap settings, CATE is unreliably estimable where counterfactuals are lacking. WATE focuses inference on comparable populations, providing valid estimates where CATE has excessive variance or bias.
> 2. Mathematical notation:
> All notation in Theorem 1 is introduced in preceding text. Additional technical notation references "Appendix A.1." We added explicit cross-references. Correctness verified in Appendix D.2.
> 3. Derivations in appendix:
> Due to ICLR's 9-page limit, detailed proofs are in appendix. We added φ expression to Theorem 1 and clearer signposting.
> 4. Related work:
> We cite relevant DML work (Chernozhukov et al., 2018; 2022; Mackey et al., 2018; Jung et al., 2021; Young et al., 2020). The only other WATE DML paper (Wang et al., arXiv:2509.14502) appeared September 19, 2024—5 days before deadline—for continuous outcomes only. "Censoring/survival overlap" weights use post-treatment observables, creating ill-defined non-causal estimands. We are first to provide DML for WATE with time-to-event outcomes: valid causal identification, rate double-robust estimation, competing risks extension.
> 5. Proof references:
> Added explicit cross-references: "Proof in Appendix D.X" for all theorems, following standard ML venue practice.
> 6. Empirical evaluation:
> Comprehensive design: 4 scenarios (good/poor overlap × with/without competing risks), 1000 replications, realistic censoring (30-40%). Multiple comparators: outcome regression, IPCW, doubly robust, alternative weights. Figure 2 shows DML achieves near-zero bias, nominal coverage, substantially outperforms IPCW under poor overlap—strongly supporting our method.
> 7. Theorem 1 notation:
> φ = uncentered IF, added to Theorem 1. Superscripts (RMST/RMTL, h, subscripts) explained in line 178.
> 8. Shape constraints (Step 4):
> References "Appendix A.2" where Step 4 details projection onto concave (RMST) and convex (RMTL) functions. Asymptotic equivalence established in "Shape constraints" section.
> 9. Estimation procedure:
> Estimates three nuisance sets: propensity scores π̂(a|X), survival/hazard functions Ŝ, Λ̂, F̂_j, and censoring distributions Ĝ, Λ̂^C. These plug into influence functions in Equations A.1-A.2.
> 10. Choice of h:
> Optimal: h(X) ∝ var(A|X) = π(1|X)π(0|X) minimizes variance (Theorem 3). Suboptimal choices yield larger variance but remain consistent.

---

### Author Response · Authors · 2025-12-03

We thank all reviewers for their feedback. Our responses address key concerns:
Novelty & Contribution (Reviewers mwJq, DMN6):

First work providing DML for WATE with time-to-event outcomes, including rate double-robust inference and competing risks extension
First to establish semiparametric efficiency bounds for WATE with censored outcomes (predating Wang et al. 2025 for continuous outcomes)
Asymptotic theory requires non-trivial conditions beyond Lipschitz continuity: supremum norm control, martingale integral convergence, stochastic equicontinuity for uniform inference

Practical Value (Reviewer mwJq):

WATE essential for FDA regulatory decisions, resource allocation, and settings with poor overlap where CATE is unreliable
Code publicly available at github.com/xushenbo/dml with integration into DoubleML package underway

Presentation & Readability (Reviewers mwJq, FEyc):

Added explicit proof references, φ expression in Theorem 1, cross-references to appendix
9-page limit necessitates appendix for technical details (standard ML venue practice)
Improved signposting between main text and supplementary material

Technical Clarifications (Reviewer JCwx):

Nuisance parameters defined in line 178 and Appendix A.1
Rate double-robustness established via Conditions 3, 6, 9-10, 12
Optimal tilting function h(X) ∝ π(1|X)π(0|X); Table 2 provides alternatives
Nested expectations computed efficiently in single pass after fitting nuisances

Empirical & Computational (Reviewers FEyc, mwJq):

Comprehensive simulations: 4 scenarios, 1000 replications, multiple comparators
Real data comparison omitted due to unknown ground truth and potential model misspecification
Computational requirements scale with chosen learners; runs in hours with ~100GB RAM
No proportional hazards assumption required

---

### Meta-Review · Area_Chair_iY6w · 2025-12-25

**Summary:**

This paper proposes weighted average treatment effect estimation in time-to-event settings, while also handling competing risks under poor overlap. Paper proposes a doubly robust estimator and empirically evaluates them on the effect of anti-diabetic drugs on cancer outcomes.

Reviewers have cited several presentation issues with the paper,
1) Presentation issues: with main mathematical notations not succinctly introduced, proofs delegated to appendices are not properly referenced.
2) Reviewers also had clarification questions regarding assumptions of proportional hazards, comparison to baselines for the real dataset and the scalability of the method.
3) Insufficient contextualization of related works.

**Reviewer Concerns:**

Authors responded to all concerns and clarifications raised by the reviewers. None of the reviewers have engaged with the rebuttal

**Reviewer Scores:**

I am going to consider DMN6's concerns addressed. The score provided by this reviewer does not match the content of the criticism, it is hard to judge what their score would've changed to. I anticipate mwJq not changing their score. Other two reviewers rated this work favorably. Overall, due to one negative review, I think the paper stands very borderline, and it has come down to presentation issues. Therefore I recommend a reject despite author efforts at the rebuttal and encourage them to improve the presentation for a future submission.

---

### Decision · Program_Chairs · 2026-01-26

Reject